# Homomorphism Expressivity of Spectral Invariant Graph Neural Networks

**Jingchu Gai** [1]   **Yiheng Du** [1]   **Bohang Zhang**[1]*  **Haggai Maron** [2,3]   **Liwei Wang** [1]

[1]Peking University   [2]Technion   [3]NVIDIA Research

`gaijingchu@stu.pku.edu.cn`,  `zhangbohang@pku.edu.cn`,  `duyiheng@stu.pku.edu.cn`
`hmaron@nvidia.com`,  `wanglw@pku.edu.cn`

## Abstract

Graph spectra are an important class of structural features on graphs that have shown promising results in enhancing Graph Neural Networks (GNNs). Despite their widespread practical use, the theoretical understanding of the power of spectral invariants — particularly their contribution to GNNs — remains incomplete. In this paper, we address this fundamental question through the lens of homomorphism expressivity, providing a comprehensive and quantitative analysis of the expressive power of spectral invariants. Specifically, we prove that spectral invariant GNNs can homomorphism-count exactly a class of specific tree-like graphs which we refer to as *parallel trees*. We highlight the significance of this result in various contexts, including establishing a quantitative expressiveness hierarchy across different architectural variants, offering insights into the impact of GNN depth, and understanding the subgraph counting capabilities of spectral invariant GNNs. In particular, our results significantly extend Arvind et al. (2024) and settle their open questions. Finally, we generalize our analysis to higher-order GNNs and answer an open question raised by Zhang et al. (2024b).

## 1 Introduction

The graph spectrum, defined as the eigenvalues of a graph matrix, is an important class of graph invariants. It encapsulates rich graph structural information including the graph connectivity, bipartiteness, node clustering patterns, diameter, and more (Brouwer & Haemers, 2011). Besides eigenvalues, generalized spectral information may also include projection matrices, which further encodes node relations such as distances and random walk properties, enabling the definition of more fine-grained graph invariants (Fürer, 2010). These spectral invariants possesses strong *expressive power*. For example, a well-known conjecture raised by Van Dam & Haemers (2003); Haemers & Spence (2004) claimed that almost all graphs can be uniquely determined by their spectra up to isomorphism. The rare exceptions, known as cospectral graphs, tend to be highly similar in their structure and continue to be an active area of research in graph theory (Lorenzen, 2022).

In the machine learning community, spectral invariants have recently gained increasing popularity in designing Graph Neural Networks (GNNs) (Bruna et al., 2013; Defferrard et al., 2016; Lim et al., 2023; Huang et al., 2024; Feldman et al., 2023; Zhang et al., 2024b; Black et al., 2024), owing to several reasons. From a practical perspective, graph spectra have been shown to be closely related to certain practical applications such as molecular property prediction (Bonchev, 2018). Moreover, a recent line of works (Xu et al., 2019; Morris et al., 2019; Li et al., 2020; Chen et al., 2020; Zhang et al., 2023b) has pointed out that the expressive power of classic message-passing GNNs (MPNNs) are inherently limited, and cannot encode important graph structure like connectivity or distance. Incorporating spectral invariants into the design of MPNNs can naturally alleviate the limitations.

Therefore, from both theoretical and practical perspectives, it is beneficial to give a systematic understanding of the power of spectral invariants and their corresponding GNNs. The earliest study in this area may be traced back to Fürer (2010), who first linked the power of several spectral invariants to the classic Weisfeiler-Lehman test (Weisfeiler & Lehman, 1968) by proving that these invariants are upper bounded by 2-FWL. More recently, Rattan & Seppelt (2023) further revealed a strict

---

*Project lead.

expressivity gap between Fürer's spectral invariants and 2-FWL. Zhang et al. (2024b) and Arvind et al. (2024) analyzed *refinement-based* spectral invariants, which offer insights into the power of real GNN architectures. Yet, all of these works study expressiveness through the lens of Weisfeiler-Lehman tests, which has inherent limitations. So far, there remains a lack of *comprehensive* understanding of the *practical* power of spectral invariants and their corresponding GNN architectures.

**Current work.** In this paper, we investigate the aforementioned questions via a novel perspective called *graph homomorphism*. Specifically, Zhang et al. (2024a) recently proposed homomorphism expressivity as a quantitative framework to better understand the expressive power of various GNN architectures. As homomorphism expressivity is a fine-grained and practical measure, it naturally addresses several limitations of the WL test. However, extending this framework to other architectures, such as spectral invariant GNNs, poses significant challenges. In fact, whether homomorphism expressivity exists for a given architecture remains an open research direction (see Zhang et al. (2024a)). In our context, this problem becomes even challenging since homomorphism and spectral invariants correspond to two orthogonal branches in graph theory. Here, we provide affirmative answers to all these questions by formally proving that the homomorphism expressivity for spectral invariant GNNs exists and can be elegantly characterized as a special class of *parallel trees* (Theorem 3.3). This offers deep insights into a series of previous studies, extending their results and answering several open questions. We summarize our results below:

- **Separation power of spectral invariants/GNNs**. We offer a new proof that projection-based spectral invariants and corresponding GNNs are strictly bounded by 2-FWL (Corollary 3.4). Moreover, we establish a *quantitative hierarchy* among raw spectra information, projection, refinement-based spectral invariant, and various combinatorial variants of WL tests (see Figure 4). This (i) recovers and extends results in Rattan & Seppelt (2023), and (ii) provides clear insights into the hierarchy established in Zhang et al. (2024b).

- **The power of refinement**. We offer a systematic understanding of the role of refinement in spectral invariant GNNs. We show increasing the number of iterations always leads to a strict improvement in expressive power (Corollary 3.11), thus settling a key open question raised in Arvind et al. (2024). Moreover, our counterexamples establish a tight lower bound on the number of iterations required to achieve maximal expressivity, which is in the same order of graph size. This advances a line of research regarding iteration numbers in WL tests (Fürer, 2001; Kiefer & Schweitzer, 2016; Lichter et al., 2019).

- **Substructure counting power of spectral invariants/GNNs**. On the practical side, we precisely characterize the power of spectral invariants/GNNs in counting certain subgraphs as well as the required iterations. For example, they can count all cycles within 7 vertices, while using 1 iteration already suffices to count all cycles within 6 vertices (Corollary 3.15).

Empirically, a set of experiments on both synthetic and real-world tasks validate our theoretical results, showing that the homomorphism expressivity of spectral invariant GNNs well reflects their performance in down-stream tasks.

## 2 PRELIMINARIES

**Notations.** We use $\{\ \}$ and $\{\{\ \}\}$ to denote sets and multisets, respectively. The cardinality of a given (multi)set $S$ is denoted as $|S|$. In this paper, we consider finite, undirected, simple graphs with no self-loops or repeated edges, and without loss of generality we only consider connected graphs. Let $G = (V_G, E_G)$ be a graph with vertex set $V_G$ and edge set $E_G$, where each edge in $E_G$ is a set $\{u, v\} \subset V_G$ of cardinality two. The *neighbors* of vertex $u$ is denoted as $N_G(u) := \{v \in V_G | \{u, v\} \in E_G\}$. A *walk* of length $k$ is a sequence of vertices $u_0, \cdots, u_k \in V_G$ such that $\{u_{i-1}, u_i\} \in E_G$ for all $i \in [k]$. It is further called a *path* if $u_i \neq u_j$ for all $i < j$, and it is called a *cycle* if $u_0, \cdots, u_{k-1}$ is a path and $u_0 = u_k$. The shortest path distance between two nodes $u, v \in V_G$, denoted as $\mathsf{dis}_G(u, v)$, is the minimum length of walk from $u$ to $v$. A graph $F = (V_F, E_F)$ is a *subgraph* of $G$ if $V_F \subset V_G$ and $E_F \subset E_G$. We use $P_n$ (resp. $C_n$) to denote a graph corresponding to a path (resp. cycle) of $n$ vertices. A graph is called a tree if it is connected and contains no cycle as a subgraph. We denote by $T^r$ the rooted tree $T$ with root $r$. The depth of a rooted tree $T^r$ is defined as $\mathsf{dep}(T^r) = \max_{u \in V_T} \mathsf{dis}_T(r, u)$, and the depth of $T$ is defined as $\mathsf{dep}(T) = \min_{r \in V_T} \mathsf{dep}(T^r)$.

## 2.1 SPECTRAL INVARIANT GNNs

Let $G$ be a graph of $n$ vertices where $V_G = [n]$, and denote by $\boldsymbol{A} \in \{0,1\}^{n \times n}$ the adjacency matrix of $G$. The *spectrum* of $G$ is defined as the multiset of all eigenvalues of $\boldsymbol{A}$. In addition to eigenvalues, eigenspaces also provide important spectral information. Formally, the eigenspace associated with some eigenvalue $\lambda$ can be characterized by its projection matrix $\boldsymbol{P}_\lambda$. It follows that there exist a unique set of orthogonal projection matrices $\{\boldsymbol{P}_\lambda\}_{\lambda \in \Lambda}$, where $\Lambda$ is the set of all distinct eigenvalues of $\boldsymbol{A}$, such that $\boldsymbol{A} = \sum_{\lambda \in \Lambda} \lambda \boldsymbol{P}_\lambda$, and the following conditions hold: $\sum_\lambda \boldsymbol{P}_\lambda = \boldsymbol{I}$, $\boldsymbol{P}_\lambda \boldsymbol{P}_{\lambda'} = 0$ for $\lambda \neq \lambda'$, and $\boldsymbol{A} \boldsymbol{P}_\lambda = \boldsymbol{P}_\lambda \boldsymbol{A}$ for all $\lambda \in \Lambda$. Combining the projection matrices with the associated eigenvalues naturally define an invariant between node pairs, which we denote by $\mathcal{P}$:

$$\mathcal{P}(u,v) := \{\!\{(\lambda, \boldsymbol{P}_\lambda(u,v)) | \lambda \in \Lambda\}\!\} \quad \text{for } u,v \in V_G.$$

Then, one can define the so-called "spectral invariant" of a graph as follows. Consider the following color refinement process by treating $\mathcal{P}(u,v)$ as the edge feature between vertices $u$ and $v$:

$$\chi_G^{\mathsf{Spec},(d+1)}(u) = \mathsf{hash}\left(\chi_G^{\mathsf{Spec},(d)}(u), \{\!\{(\chi_G^{\mathsf{Spec},(d)}(v), \mathcal{P}(u,v)) | v \in V_G\}\!\}\right) \quad \text{for } u \in V_G, d \in N_+,$$

where all colors $\chi_G^{\mathsf{Spec},(0)}(u)$ ($u \in V_G$) are constant in initialization, and $\mathsf{hash}$ is a perfect hash function. For each iteration $d$, the mapping $\chi_G^{\mathsf{Spec},(d)}$ induces an equivalence relation over vertex set $V_G$, and the relation gets *refined* with the increase of $d$. Therefore, with a sufficiently large number of iterations $d \leq |V_G|$, the relations get *stable*. The spectral invariant $\chi_G^{\mathsf{Spec},(\infty)}(G)$ is then defined to be the multiset of stable node colors. We can similarly define $\chi_G^{\mathsf{Spec},(d)}(G)$ to be the multiset of node colors after $d$ iterations (Arvind et al., 2024). We remark that $\chi_G^{\mathsf{Spec},(1)}(G)$ is exactly the Fürer's (weak) spectral invariant proposed in Fürer (2010).

Owing to the relation between GNNs and color refinement algorithms, one can easily transform the above refinement process into a GNN architecture by replacing hash function with a continuous, non-linear, parameterized function, while maintaining the same expressive power (Xu et al., 2019; Morris et al., 2019). We call the resulting architecture Spectral Invariant GNNs (see Zhang et al. (2024b) for concrete implementations of spectral invariant GNN layer). Without ambiguity, we may also refer to $\chi_G^{\mathsf{Spec},(d)}(G)$ as the graph representation computed by a $d$-layer spectral invariant GNN.

## 2.2 HOMOMORPHISM EXPRESSIVITY

Given two graphs $F$ and $G$, a homomorphism from $F$ to $G$ is a mapping $f : V_F \to V_G$ that preserves edge relations, i.e., $\{f(u), f(v)\} \in E_G$ for all $\{u,v\} \in E_F$. We denote by $\mathsf{Hom}(F,G)$ the set of all homomorphisms from $F$ to $G$ and define $\mathsf{hom}(F,G) = |\mathsf{Hom}(F,G)|$, which counts the number of homomorphisms. If $f$ is further surjective on both vertices and edges of $G$, we call $G$ a *homomorphic image* of $F$. A mapping $f : V_F \to V_G$ is called an isomorphism if $f$ is a bijection and both $f$ and its inverse $f^{-1}$ are homomorphisms. We denote by $\mathsf{sub}(F,G)$ the number of subgraphs of $G$ that is isomorphic to $F$.

In Zhang et al. (2024a), the authors introduced the concept the homomorphism expressivity to quantify the expressive power of a color refinement algorithm (or GNN). It is formally defined as follows:

**Definition 2.1.** Let $M$ be a color refinement algorithm (or GNN) that outputs a graph invariant $\chi_G^M(G)$ given graph $G$. The homomorphism expressivity of $M$, denoted by $\mathcal{F}^M$, is a family of connected graphs[1] satisfying the following conditions:

   a) For any two graphs $G, H$, $\chi_G^M(G) = \chi_H^M(H)$ *iff* $\mathsf{hom}(F,G) = \mathsf{hom}(F,H)$ for all $F \in \mathcal{F}^M$;

   b) $\mathcal{F}^M$ is maximal, i.e., for any connected graph $F \notin \mathcal{F}^M$, there exists a pair of graphs $G, H$ such that $\chi_G^M(G) = \chi_H^M(H)$ and $\mathsf{hom}(F,G) \neq \mathsf{hom}(F,H)$.

By characterizing the set $\mathcal{F}^M$ for different GNN models $M$, one can quantitatively understand the expressivity gap between two models by simply computing their set inclusion relation and set difference. Zhang et al. (2024a) examines several representative GNNs under this framework, including

---

[1] For simplicity, we focus on *connected* graphs in this paper. The results can be easily generalized to disconnected graphs following Seppelt (2024).

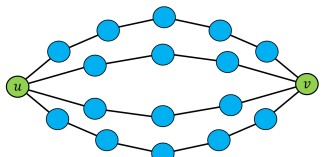 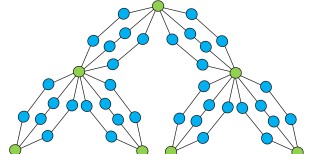 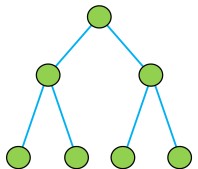

(a) A parallel edge with endpoints $(u, v)$     (b) An example of parallel tree and its tree skeleton

Figure 1: Illustration of a parallel edge with endpoints $(u, v)$ in (a) and a parallel tree with its skeleton on the right in (b).

the standard MPNNs and Folklore GNNs (Maron et al., 2019; Azizian & Lelarge, 2021), and recent architectures such as Subgraph GNN (Bevilacqua et al., 2022; Qian et al., 2022; Cotta et al., 2021) and Local GNN (Morris et al., 2020; Zhang et al., 2023a). However, one implicit challenge not reflected in Definition 2.1(a) is that the set $\mathcal{F}^M$ may not even exist for a general GNN $M$. Proving the existence corresponds to an involved research topic known as homomorphism distinguishing closedness (Roberson, 2022; Seppelt, 2024; Neuen, 2023), which is highly non-trivial. In the next section, we will give affirmative results showing that the homomorphism expressivity of spectral invariant GNNs does exist and give an elegant description of the graph family.

# 3 HOMOMORPHISM EXPRESSIVITY OF SPECTRAL INVARIANT GNNS

In this section, we investigate the homomorphism expressivity of spectral invariants and the corresponding GNNs. We will provide a complete characterization of the set $\mathcal{F}^{\mathsf{Spec},(d)}$ for arbitrary model depth $d \in \mathbb{N} \cup \{\infty\}$. This allows us to analyze spectral invariants in a novel perspective, significantly extending prior research and resolving previously unanswered questions.

## 3.1 MAIN RESULTS

Our idea is motivated by the previous finding that the homomorphism expressivity of MPNNs is exactly the family of all trees (Zhang et al., 2024a). Note that in the definition of spectral invariant GNN, if one replaces $\mathcal{P}(u, v)$ by the standard adjacency $\boldsymbol{A}_{uv}$, the resulting architecture is just an MPNN. Such a relationship perhaps implies that the homomorphism expressivity of spectral invariant GNNs also comprises "tree-like" graphs. We will show this is indeed true. To present our results, let us define a special class of graphs, referred to as *parallel trees*:

**Definition 3.1** (**Parallel Edge**). A graph $G$ is called a *parallel edge* if there exist two different vertices $u, v \in V_G$ such that the edge set $E_G$ can be partitioned into a sequence of simple paths $P_1, \ldots, P_m$, where all paths share endpoints $(u, v)$. We refer to $(u, v)$ as the endpoints of $G$.

**Definition 3.2** (**Parallel Tree**). A graph $F$ is called a *parallel tree* if there exists a tree $T$ such that $F$ can be obtained from $T$ by replacing each edge $\{u, v\} \in E_T$ with a parallel edge that has endpoints $\{u, v\}$. We refer to $T$ as the *parallel tree skeleton* of graph $F$. Given a parallel tree $F$, define the *parallel tree depth* of $F$ as the minimum depth of any parallel tree skeleton of $F$.

We give an illustration of parallel edge and parallel tree in Figure 1. With the above definitions, we are ready to state our main theorem:

**Theorem 3.3.** *For any $d \in \mathbb{N}$, the homomorphism expressivity of spectral invariant GNNs with $d$ iterations exists and can be characterized as follows:*

$$\mathcal{F}^{\mathsf{Spec},(d)} = \{F \mid F \text{ has parallel tree depth at most } d\}.$$

*Specifically, the following properties hold:*

- *Given any graphs $G$ and $H$, $\chi_G^{\mathsf{Spec},(d)}(G) = \chi_H^{\mathsf{Spec},(d)}(H)$ if and only if, for all connected graphs $F$ with parallel tree depth at most $d$, $\mathsf{hom}(F, G) = \mathsf{hom}(F, H)$.*

- *$\mathcal{F}^{\mathsf{Spec},(d)}$ is maximal; that is, for any connected graph $F \notin \mathcal{F}^{\mathsf{Spec},(d)}$, there exist graphs $G$ and $H$ such that $\chi_G^{\mathsf{Spec},(d)}(G) = \chi_H^{\mathsf{Spec},(d)}(H)$ and $\mathsf{hom}(F, G) \neq \mathsf{hom}(F, H)$.*

We will present a concise proof sketch of Theorem 3.3 in Section 3.3. Next, in Section 3.2, we will interpret this result in the context of GNNs and discuss its significance, including how it extends previous findings and addresses open problems identified in earlier studies.

## 3.2 IMPLICATIONS

Our theory has a wide range of applications, which will be separately discussed in detail below.

### 3.2.1 COMPARISON WITH 2-FWL

Firstly, we compare the expressive power of spectral invariant GNNs with the expressive power of the standard Weisfeiler-Lehman (WL) test. It immediately follows that the expressive power of spectral invariant GNNs strictly lies between the expressive power of 1-WL and 2-FWL test.

**Corollary 3.4.** *The expressive power of spectral invariant GNNs is strictly stronger than* 1-*WL and strictly weaker than* 2-*FWL.*

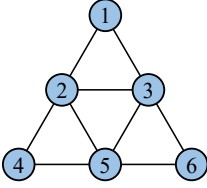

Figure 2: A counterexample graph in $\mathcal{F}^{2-\mathsf{FWL}} \backslash \mathcal{F}^{\mathsf{Spec},(\infty)}$.

*Proof.* According to Zhang et al. (2024a), the homomorphism expressivity of 2-FWL encompasses the set of all graphs with treewidth at most 2. A classical result in graph theory states that any subgraph of any series-parallel graph has treewidth at most 2 (Diestel, 2017). Since any parallel tree is clearly a subgraph of some series-parallel graph, its treewidth is at most 2. It follows that the homomorphism expressivity of parallel trees is contained within that of the 2-FWL. To show the gap, we give a counterexample graph in Figure 2. This implies that the expressive power of spectral invariant GNNs is strictly weaker than that of the 2-FWL. The proof for the case of 1-WL is similar and we omit it for clarity. □

### 3.2.2 HIERARCHY

Theorem 3.3 not only provides insights into the relationship between the expressive power of spectral invariant GNNs and 2-FWL, but also allows for a comparison with a wide range of graph invariants and the corresponding GNNs. Specifically, similar to the analysis in Corollary 3.4, for any GNN models $A$ and $B$ such that their homomorphism expressivity exists, if $\mathcal{F}^A \subsetneq \mathcal{F}^B$, then $A$ is strictly weaker than $B$ in expressive power. We now use this property to establish a comprehensive hierarchy by linking spectral invariant GNNs to other fundamental graph invariants and GNNs.

**Corollary 3.5.** *Spectral invariant GNN with* 1 *iteration is strictly weaker than subgraph GNN (also referred to as* $(1, 1)$-*WL in* Rattan & Seppelt (2023)).

*Proof.* According to Zhang et al. (2024a), the homomorphism expressivity of subgraph GNNs contains all graphs that become a forest upon the deletion of a specific vertex. On the other hand, Theorem 3.3 states that the homomorphism expressivity of spectral invariant GNNs with one iteration contains all parallel trees of depth 1. Since any parallel tree of depth 1 becomes a forest when deleting the root vertex, we have proved that $\mathcal{F}^{\mathsf{Spec},(1)}$ is a subset of that of subgraph GNNs. Finally, one can easily construct a counterexample graph to prove the strict separation. □

**Remark 3.6.** Our result recovers and strengthens the main result in Rattan & Seppelt (2023), which only studied spectral invariants with 1 iteration (Fürer's weak spectral invariant). We will next show this result actually does *not* hold in case of more than 1 iterations.

**Corollary 3.7.** *Spectral invariant GNNs with* 2 *iterations are incomparable to subgraph GNNs.*

We provide a counterexample in Figure 3. Nevertheless, we can still bound the expressive power of spectral invariant GNNs with multiple iterations to that of Local 2-GNN, as stated in the following:

**Corollary 3.8.** *For any* $d \in \mathbb{N}_+ \cup \{\infty\}$, *spectral invariant GNNs with* $d$ *iterations are strictly weaker than Local* 2-*GNN* (Morris et al., 2020; Zhang et al., 2024a).

*Proof.* According to Zhang et al. (2024a), the homomorphism expressivity of Local 2-GNNs contains all graphs that admit a strong nested ear decomposition. Since any parallel edge can be partitioned into ears with the same endpoints, one can easily construct a nested ear decomposition for any parallel tree. This shows $\mathcal{F}^{\mathsf{Spec},(d)}$ is a subset of that of Local 2-GNN. The expressivity gap can be seen using the same counterexample graph in Figure 2. □

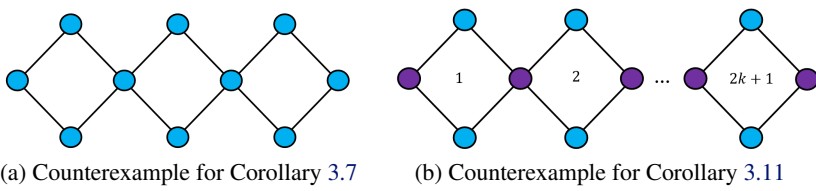

(a) Counterexample for Corollary 3.7    (b) Counterexample for Corollary 3.11

Figure 3: Counterexample for Corollary 3.7 and Corollary 3.11

**Remark 3.9.** Corollaries 3.7 and 3.8 significantly extend the findings of Arvind et al. (2024, Theorem 17) and provide additional insights into Zhang et al. (2024b, Theorem 4.3).

**The power of projection.** We next conduct a fine-grained analysis by separating eigenvalues and projections to better understand their individual contributions to enhancing the expressive power of GNN models. We first prove the following theorem:

**Theorem 3.10.** *The homomorphism expressivity of graph spectra is the set of all cycles $C_n$ ($n \geq 3$) plus paths $P_1$ and $P_2$, i.e., $\{C_n | n \geq 3\} \cup \{P_1, P_2\}$.*

The proof of Theorem 3.10 is provided in Appendix C, which has the same structure as that of Theorem 3.3. Previously, Van Dam & Haemers (2003); Dell et al. (2018) have proved that the spectra of two graphs $G$ and $H$ are identical if and only if for every cycle $F$, $\mathrm{hom}(F, G) = \mathrm{hom}(F, H)$. We extend their result by further proving the maximal property (Definition 2.1(b)), which only adds two trivial graphs $P_1$ and $P_2$ to the homomorphism expressivity. From this result, one can easily see that using eigenvalues alone can already improve the expressive power of an MPNN since the homomorphism expressivity of MPNN contains only trees (but not cycles).

To understand the role of projection, one can compare the set $\{C_n | n \geq 3\} \cup \{P_1, P_2\}$ with $\mathcal{F}^{\mathsf{Spec},(1)}$ (the homomorphism expressivity of Fürer's spectral invariant). Clearly, the set of all parallel trees of depth 1 is strictly larger than $\{C_n | n \geq 3\} \cup \{P_1, P_2\}$, confirming that adding projection information significantly enhances the expressive power beyond graph spectra.

**The power of refinement.** We finally investigate the power of iterations $d$ (or number of GNN layers) in enhancing the model's expressive power. We have the following result:

**Corollary 3.11.** *For any $d \in \mathbb{N}$, spectral invariant GNNs with $d + 1$ iterations are strictly more powerful than spectral invariant GNNs with $d$ iterations.*

*Proof.* For any $k \in \mathbb{N}$, we can construct a counterexample formed by replacing each edge in the path graph $P_{2k+2}$ with a parallel edge. We illustrate the construction in Figure 3(b). One can easily see that the resulting graph is in $\mathcal{F}^{\mathsf{Spec},(k+1)}$ but not $\mathcal{F}^{\mathsf{Spec},(k)}$. $\square$

**Remark 3.12.** Corollary 3.11 addresses the key open question posed in Arvind et al. (2024), who conjectured that spectral invariant GNNs converge within *constant* iterations. Specifically, the authors questioned whether, for $d \geq 4$, spectral invariant GNNs with $d + 1$ iterations are as powerful as those with $d$ iterations. We disproved this conjecture by providing a family of example graphs that cannot be distinguished in $d$ iterations but can be distinguished in $d + 1$ iterations.

Our counterexamples further leads to the following result:

**Corollary 3.13.** *For any $d \in \mathbb{N}_+$, There exist two graphs with $\mathcal{O}(d)$ vertices such that spectral invariant GNNs require at least $d$ iterations to distinguish between them.*

Corollary 3.13 establishes a tight bound on the number of layers needed for spectral invariant GNNs to reach maximal expressivity, showing that it scales with the order of graph size. This advances an important research topic that aims to study the relation between expressiveness and iteration number of color refinement algorithms (Fürer, 2001; Kiefer & Schweitzer, 2016; Lichter et al., 2019).

To summarize all the above results, we illustrate the hierarchy established for spectral invariant GNNs and other mainstream GNNs in Figure 4.

### 3.2.3   SUBGRAPH COUNT

In fact, our results can go beyond the WL framework and reveal the expressive power of spectral invariant GNNs in a more practical perspective. As an example, we will show below how Theorem 3.3 can be used to understand the subgraph counting capabilities of spectral invariant GNNs.

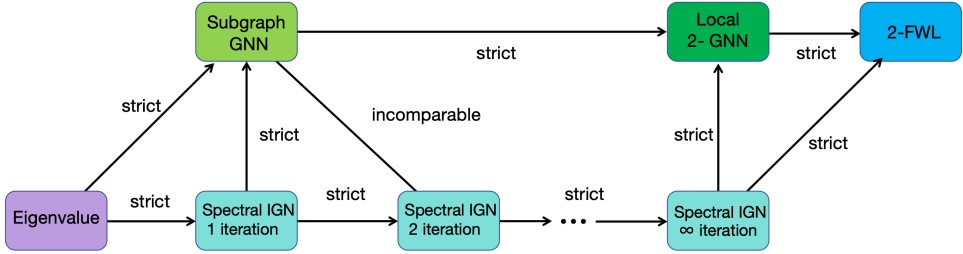

Figure 4: Hierarchy of spectral invariant GNN (abbreviated as Spectral IGN) and other mainstream GNNs. Each arrow points to the strictly stronger architecture.

Given any graph $F$, we say a GNN model $M$ can subgraph-count substructure $F$ if for any graphs $G$ and $H$, the condition $\chi_G^M(G) = \chi_H^M(H)$ implies $\mathsf{sub}(F, G) = \mathsf{sub}(F, H)$. Denote by $\mathsf{Spasm}(F)$ the set of all homomorphic images of $F$. Previous results have proved that, if the homomorphism expressivity $\mathcal{F}^M$ exists for model $M$, then $M$ can subgraph-count $F$ if and only if $\mathsf{Spasm}(F) \subset \mathcal{F}^M$ (Seppelt, 2023; Zhang et al., 2024a). This allows us to precisely analyze which substructure can be subgraph-counted by spectral invariant GNNs.

**Corollary 3.14.** *Spectral invariant GNN can count cycles and paths with up to 7 vertices.*

*Proof.* For cycles or paths with at most 7 vertices, one can check by enumeration that their homomorphic images are all parallel trees. For cycles or paths with at least 8 vertices, the 4-clique is a valid homomorphic image but is not a parallel tree. ☐

We can further strengthen the above results by studying the number of iterations needed to count substructures. We have the following results:

**Corollary 3.15.** *The following holds:*

1. *Spectral invariant GNNs can subgraph-count all cycles up to 7 vertices within 2 iterations.*

2. *The above upper bound is tight: spectral invariant GNNs with only 1 iteration (i.e., Fürer's weak spectral invariant) cannot subgraph-count 7-cycle.*

3. *Spectral invariant GNNs with 1 iteration suffice to subgraph-count all cycles up to 6 vertices.*

**Remark 3.16.** The subgraph counting power of spectral invariant has long been studied in the literature. Cvetkovic et al. (1997) proved that the graph angles (which can be determined by projection) can subgraph-count all cycles of length no more than 5. In comparison, our results significantly extend their findings, which even match the cycle counting power of 2-FWL (Arvind et al., 2020). Moreover, we show that Fürer's weak spectral invariant can already count 6-cycles, thus extending the work of Fürer (2017).

### 3.3 PROOF SKETCH

In this section, we provide a proof sketch of Theorem 3.3, with the complete proof presented in the Appendix. We begin by demonstrating that the information encoded by spectral invariants is closely related to encoding *walk information* in the aggregation process of GNNs. This corresponds to the following lemma (proved in Appendix B.2, see also Arvind et al. (2024)):

**Lemma 3.17.** *(Equivalence of encoding walk and encoding spectral information)* *Let $G = (V_G, E_G)$ be a graph, with its adjacency matrix denoted by $\boldsymbol{A}$. For vertices $x, y \in V_G$, define $\omega_G^k(x, y) = \boldsymbol{A}_{x,y}^k$ for all $k \in \{0, 1, 2, \ldots, |V_G|\}$, which represents the number of $k$-walks from vertex $x$ to vertex $y$. Define the tuple $\omega_G^*(x, y) = (\omega_G^0(x, y), \omega_G^1(x, y), \ldots, \omega_G^{n-1}(x, y))$, where $n = |V_G|$. Define the walk-encoding GNN with the following update rule:*

$$\chi_G^{\mathsf{Walk}, (d+1)}(x) = \mathsf{hash}(\chi_G^{\mathsf{Walk}, (d)}(x), \{\!\{(\omega_G^*(x, y), \chi_G^{\mathsf{Walk}, (d)}(y)) \mid y \in V_G\}\!\}).$$

*The walk-encoding GNN outputs a representation $\chi_G^{\mathsf{Walk}, (d)}(G) = \{\!\{\chi_G^{\mathsf{Walk}, (d)}(u) \mid u \in V_G\}\!\}$. For any graphs $G$, $H$, we have $\chi_G^{\mathsf{Walk}, (d)}(G) = \chi_H^{\mathsf{Walk}, (d)}(H)$ if and only if $\chi_G^{\mathsf{Spec}, (d)}(G) = \chi_H^{\mathsf{Spec}, (d)}(H)$.*

Our next step aims to prove that for graphs $G$ and $H$, $\chi_G^{\mathsf{Walk}, (d)}(G) = \chi_H^{\mathsf{Walk}, (d)}(H)$ iff, for all graphs $F$ with parallel tree depth at most $d$, $\mathsf{hom}(F, G) = \mathsf{hom}(F, H)$. This will yield the first property outlined in Theorem 3.3. The proof has a similar structure to that in Zhang et al. (2024a), which

is based on the tools of tree-decomposed graphs and algebraic graph theory (see Theorems B.14 and B.20 and Lemma B.17). This part corresponds to Appendix B.3.

Now, it remains to prove that the set $\mathcal{F}^{\mathsf{Spec},(d)}$ is maximal (the second property in Theorem 3.3). To achieve this, we leverage the technique known as pebble game (Cai et al., 1992), which was originally used to construct counterexample graphs that cannot be distinguished by the $k$-FWL test. We extend the framework and define the pebble game for spectral invariant GNNs as follows:

**Definition 3.18.** (**Pebble game for spectral invariant GNNs**) The pebble game is conducted on two graphs $G = (V_G, E_G)$ and $H = (V_H, E_H)$. Without loss of generality, we assume $V_G = V_H$. Initially, each graph is equipped with two distinct pebbles denoted as $u$ and $v$, which initially lie outside the graphs. The game involves two players: the *spoiler* and the *duplicator*. The game process is described as follows:

- *Initialization:* The spoiler first selects a non-empty subset $V^{\mathsf{S}}$ from either $V_G$ or $V_H$, and the duplicator responds with a subset $V^{\mathsf{D}}$ from the other graph, ensuring that $|V^{\mathsf{D}}| = |V^{\mathsf{S}}|$. Then, the spoiler places the pebble $u$ on some vertex in $V^{\mathsf{D}}$, and the duplicator places the corresponding pebble $u$ on some vertex in $V^{\mathsf{S}}$. Similarly, the spoiler and duplicator repeat the process to place two pebbles $v$. After the initialization, all pebbles will lie on the two graphs.

- *Main Process:* The game iteratively repeats the following steps, where in each iteration the spoiler may choose freely between the following two actions:

  1. Action 1 (moving pebble $v$). The spoiler first selects a non-empty subset $V^{\mathsf{S}}$ from either $V_G$ or $V_H$, and the duplicator responds with a subset $V^{\mathsf{D}}$ from the other graph, ensuring that $|V^{\mathsf{D}}| = |V^{\mathsf{S}}|$. The spoiler then moves pebble $v$ to some vertex in $V^{\mathsf{D}}$, and the duplicator moves the corresponding pebble $v$ to some vertex in $V^{\mathsf{S}}$.
  2. Action 2 (moving pebble $u$). This action is similar to the above one except that both players move pebble $u$ instead of pebble $v$.

- *Termination:* The spoiler wins if, after a certain number of rounds, $\omega_G^{\star}(u, v)$ for graph $G$ differs from $\omega_H^{\star}(u, v)$ for graph $H$. Conversely, the duplicator wins if the spoiler is unable to win after any number of rounds.

With the above definition, we can now prove the equivalence between the outcome of a pebble game and the ability to distinguish non-isomorphic graphs using spectral invariant GNNs:

**Lemma 3.19.** *(Equivalence of pebble game and spectral invariant GNNs)  Given graphs $G$ and $H$ and the number of steps $d \in \mathbb{N}$, the spoiler cannot win the pebble game in $d$ steps iff $\chi_G^{\mathsf{Spec},(d+1)}(G) = \chi_H^{\mathsf{Spec},(d+1)}(H)$.*

We give a proof in Appendix B.4. Next, to identify counterexamples $G$ and $H$ for any $F \notin \mathcal{F}^{\mathsf{Spec},(d)}$ such that $\chi_G^{\mathsf{Spec},(d)}(G) = \chi_H^{\mathsf{Spec},(d)}(H)$ and $\mathrm{hom}(F, G) \neq \mathrm{hom}(F, H)$, we draw inspiration from a special class of graphs called Fürer graphs (Fürer, 2001), which is a principled approach to constructing pairs of non-isomorphic but structurally similar graphs. If graphs $G$ and $H$ are the Fürer graph and twisted Fürer graph constructed from the same base graph $F$, we show that the pebble game can be significantly simplified. Importantly, the simplified pebble game will be played on the base graph $F$ instead of the complex Fürer graphs, making the subsequent analysis much easier. Due to space constraints, a detailed description of the simplified pebble game is provided in Appendix B.5. We then establish the following lemma, which relates the simplified pebble game to spectral invariant GNNs:

**Lemma 3.20.** *(Equivalence of pebble game on Fürer graphs and spectral invariant GNNs) Given a base graph $F$, let $G(F)$ and $H(F)$ be the Fürer graph and twisted Fürer graph of $F$, respectively. Then, the spoiler cannot win the simplified pebble game on $F$ in $d$ steps iff $\chi_G^{\mathsf{Spec},(d+1)}(G(F)) = \chi_H^{\mathsf{Spec},(d+1)}(H(F))$.*

Note that for any connected graph $F$, $\mathrm{hom}(F, G(F)) \neq \mathrm{hom}(F, H(F))$ (Roberson, 2022; Zhang et al., 2024a). Furthermore, we demonstrate that the spoiler has a winning strategy on $F$ in $d$ steps if and only if $F$ is a parallel tree with parallel tree depth at most $d + 1$ (see Appendix B.6). By combining these results with Lemma 3.20, we establish the following lemma:

**Lemma 3.21.** *For any $F \notin \mathcal{F}^{\mathsf{Spec},(d)}$, the spoiler cannot win the simplified pebble game on $F$. Consequently, $\chi_G^{\mathsf{Spec},(d)}(G(F)) = \chi_H^{\mathsf{Spec},(d)}(H(F))$.*

This yields the second property in Theorem 3.3 and concludes the proof.

## 3.4 EXTENSIONS

So far, this paper mainly analyzes the standard spectral invariant GNNs, which refines *node features* based on projection information. In this subsection, we will show the flexibility of our proposed homomorphism expressivity framework, which can also be used to analyze other spectral-based GNN models such as higher-order spectral invariant GNNs.

### 3.4.1 HIGHER ORDER

Let us consider generalizing Section 2.1 to higher order spectral invariant GNNs. A natural update rule of higher order spectral invariant GNN can be defined as follows:

**Definition 3.22** (**Higher-Order Spectral Invariant GNN**). For any $k \in \mathbb{N}_+$, the $k$-order spectral invariant GNN maintains a color $\chi_G^{k\text{-Spec}}(\boldsymbol{u})$ for each vertex $k$-tuple $\boldsymbol{u} = (u_1, \ldots, u_k) \in V_G^k$. Initially, $\chi_G^{k\text{-Spec},(0)}(\boldsymbol{u}) = (\mathcal{P}(u_1, u_2), \ldots, \mathcal{P}(u_1, u_k), \ldots, \mathcal{P}(u_{k-1}, u_k))$. In each iteration $t + 1$, the color is updated as follows:

$$\chi_G^{k\text{-Spec},(t+1)}(\boldsymbol{u}) = \mathsf{hash}(\chi_G^{k\text{-Spec},(t)}(\boldsymbol{u}), \{\!\!\{(\chi_G^{k\text{-Spec},(t)}(v, u_2, \ldots, u_k), \mathcal{P}(u_1, v)) : v \in V_G\}\!\!\}, \cdots,$$
$$\{\!\!\{(\chi_G^{k\text{-Spec},(t)}(u_1, u_2, \ldots, u_{k-1}, v), \mathcal{P}(u_k, v)) : v \in V_G\}\!\!\}).$$

Denote the stable color of vertex tuple $\boldsymbol{u} \in V_G^k$ as $\chi_G^{k\text{-Spec}}(\boldsymbol{u})$. The graph representation is defined as $\chi_G^{k\text{-Spec}}(G) := \{\!\!\{\chi_G^{k\text{-Spec}}(\boldsymbol{u}) : \boldsymbol{u} \in V_G^k\}\!\!\}$.

One can see that when $k = 1$, the above definition degenerates to the standard spectral invariant GNN defined in Section 2.1. To illustrate the homomorphism expressivity of higher-order spectral invariant GNNs, we extend the concept of strong nested ear decomposition (NED) introduced by Zhang et al. (2024a) and define the parallel strong NED. Our main result is stated below:

**Theorem 3.23** (informal). *A graph $F$ is said to have a parallel $k$-order strong nested ear decomposition (NED) if there exists a graph $G$ such that $G$ admits a strong NED and $F$ can be obtained from $G$ by replacing each edge $\{u, v\} \in E_G$ with a parallel edge that has endpoints $(u, v)$. Then, the homomorphism expressivity of $k$-order spectral invariant GNNs is the set of all graphs that admit a parallel $k$-order strong NED.*

Due to space constraints, we leave the formal definition of $k$-order strong NED and the technical proof of Theorem 3.23 to the Appendix.

### 3.4.2 SYMMETRIC POWER

To generalize spectrum and projection to higher order, another classic approach in the literature is to use the symmetric power of a graph (also called the *token graph*). Audenaert et al. (2005) first introduced the graph symmetric power to generalize eigenvalues into higher-order graph invariants. The formal definition of the symmetric $k$-th power is presented as follows:

**Definition 3.24** (**Symmetric Power**). For any $k \in \mathbb{N}_+$ and graph $G$, the symmetric $k$-th power of $G$, denoted by $G^{\{k\}}$, is a graph where its vertices are $k$-subsets of $V_G$, and two subsets are adjacent if and only if their symmetric difference is an edge in $G$.

Our homomorphism expressivity framework can be used to study the ability of mainstream GNNs to encode the symmetric power of graphs. Our main result is stated as follows:

**Theorem 3.25.** *The Local $2k$-GNN defined in Morris et al. (2020); Zhang et al. (2024a) can encode the symmetric $k$-th power. Specifically, for given graphs $G$ and $H$, if $G$ and $H$ have the same representation under Local $2k$-GNN, then $G^{\{k\}}$ and $H^{\{k\}}$ have the same representation under the spectral invariant GNN defined in Section 2.1.*

**Discussions with prior work.** Regarding the expressive power of symmetric power, Alzaga et al. (2008); Barghi & Ponomarenko (2009) gave the first upper bound, showing that if $2k$-FWL fails to distinguish between two non-isomorphic graphs, then their symmetric $k$-th powers are cospectral.

Table 1: Experimental results on homomorphism counting, real-world tasks and substructure count.

| Task Model | Homomorphism Count | | | | | ZINC | | Substructure Count | | | | | |
|---|---|---|---|---|---|---|---|---|---|---|---|---|---|
| | | | | | | Subset | Full | | | | | | |
| MPNN | .300 | .261 | .276 | .233 | .341 | $.138 \pm .006$ | $.030 \pm .002$ | .358 | .208 | .188 | .146 | .261 | .205 |
| Spectral Invariant GNN | .045 | .046 | .053 | .048 | .303 | $.103 \pm .006$ | $.028 \pm .003$ | .072 | .072 | .089 | .089 | .060 | .099 |
| Subgraph GNN | .011 | .013 | .010 | .015 | .260 | $.110 \pm .007$ | $.028 \pm .002$ | .010 | .020 | .024 | .046 | .007 | .027 |
| Local 2-GNN | .008 | .006 | .008 | .008 | .112 | $.069 \pm .001$ | $.024 \pm .002$ | .008 | .011 | .017 | .034 | .007 | .016 |

However, it remains unclear whether the conclusion extends to the more powerful projection information (beyond eigenvalues), or if the stated upper bound is tight. These open questions are further highlighted in Zhang et al. (2024b). Our result answers both questions by bounding the stronger refinement-based spectral invariant for the $k$-th symmetric power graphs to Local $2k$-GNN, which is strictly weaker than $2k$-FWL (Zhang et al., 2024a). This offers a deeper understanding of the capability of mainstream GNNs in encoding higher-order spectral information.

## 4 EXPERIMENT

In this section, we validate our theoretical findings through empirical experiments. We evaluate the performance of GNN models on both synthetic and real-world tasks. For the synthetic tasks, we assess the homomorphic counting power and subgraph counting power of the GNN models. These experiments serve to confirm our theoretical results, including Theorem 3.3 and Corollary 3.14. In addition, for the real-world task, we focus on molecular reaction prediction, specifically evaluating GNN performance on the ZINC dataset (Dwivedi et al., 2020). Our primary objective is not to achieve SOTA results but to validate our theoretical findings. We compare the performance of spectral invariant GNNs to both MPNNs and subgraph GNNs on the ZINC dataset. Details about model architectures are in Appendix D.

**Homomorphism Count** We use the benchmark dataset from Zhao et al. (2022) to evaluate the homomorphism expressivity of four mainstream GNN models. The reported performance is measured by the normalized Mean Absolute Error (MAE) on the test set. The empirical results are presented in Table 1. We can see that concerning homomorphism: (i) MPNN is unable to encode any of the five substructures, and none of the five substructures is a tree; (ii) Spectral invariant GNN can only encode the 1st and 2nd substructures; (iii) Subgraph GNN can encode the 1st, 2nd, and 3rd substructures; and (iv) Local 2-GNN can encode the 1st, 2nd, 3rd, and 4th substructures. The empirical results basically align with our theoretical findings.

**Subgraph Count** Cycle counting is a fundamental problem in chemical and biological tasks. Following the settings in Frasca et al. (2022); Zhang et al. (2023a); Huang et al. (2023), we evaluate the cycle counting power of four GNNs. The empirical results in Table 1 demonstrate that the spectral invariant GNN can accurately count 3-, 4-, 5-, and 6-cycles, indicating its strong performance in cycle counting tasks. This empirical result is also consistent with our theoretical predictions.

**Real-World Task** We evaluate our GNN models on the ZINC-subset and ZINC-full dataset (Dwivedi et al., 2020). Following the standard configuration, all models are constrained to a 500K parameter budget. The results show that the spectral invariant GNN outperforms MPNN while demonstrating comparable performance to the subgraph GNN on the real-world task. These findings are consistent with our theoretical predictions.

## 5 CONCLUSION

In this work, we investigate the expressive power of spectral invariant graph neural networks (GNNs). By leveraging the framework of homomorphism expressivity, we give a precise characterization the homomorphism expressivity of these networks. We then establish a comprehensive hierarchy of spectral invariant GNNs relative to other mainstream GNNs based on their homomorphism expressivity. Additionally, we analyze the subgraph counting capabilities of spectral invariant GNNs, with a focus on their ability to count essential substructures. Our results are extended to higher-order contexts and address additional problems related to spectral structures using our homomorphism framework. We demonstrate the significance of our findings by showing how our results extend previous work and address open problems identified in the literature. Finally, we conduct experiments to validate our theoretical results.

## ACKNOWLEDGEMENTS

This work is supported by National Science and Technology Major Project (2022ZD0114902)and National Science Foundation of China (NSFC92470123, NSFC62276005).

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

# Appendix

## Table of Contents

# A  ADDITIONAL RELATED WORK

**Spectral Based Graph Neural Network.**    Spectral invariants refer to eigenvalues, projection matrices, and other generalized spectral information. In recent studies, spectral invariants have gained significant attention in the fields of graph learning and graph theory (Fürer, 2010; Van Dam & Haemers, 2003; Haemers & Spence, 2004). For instance, a well-known conjecture proposed by Van Dam & Haemers (2003); Haemers & Spence (2004) posits that almost all graphs can be uniquely determined by their spectra, up to isomorphism. Given the importance and widespread application of graph spectral information (Bonchev, 2018), the machine learning community has also focused on analyzing the ability of graph neural networks (GNNs) to encode spectral information and on designing GNN models that incorporate more spectral features. As a result, several recent works have concentrated on the spectral-based design of GNNs (Bruna et al., 2013; Defferrard et al., 2016; Lim et al., 2023; Huang et al., 2024; Feldman et al., 2023; Zhang et al., 2024b). Specifically, Dwivedi et al. (2023; 2021); Kreuzer et al. (2021); Rampášek et al. (2022) have designed spectral GNNs by encoding Laplacian eigenvectors as absolute positional encodings. A key drawback of using Laplacian eigenvectors is the ambiguity in choosing eigenvectors; thus, follow-up works have sought to design GNNs that are invariant to the choice of eigenvectors. Lim et al. (2023) introduced BasisNet, which achieves spectral invariance for the first time using projection matrices. Huang et al. (2024) further generalized BasisNet by proposing the Spectral Projection Encoding (SPE), which performs soft aggregation across different eigenspaces, as opposed to the hard separation implemented in BasisNet.

In addition to the design of spectral-based GNNs, several recent works have also focused on analyzing the expressive power of spectral GNNs and comparing them with other mainstream GNN models. Balcilar et al. (2021) investigate the relationship between ChebNet (Defferrard et al., 2016) and the 1-WL test, demonstrating that for graphs with similar maximum eigenvalues, ChebNet is as expressive as 1-WL. Geerts & Reutter (2022) revisit this analysis and prove that CaleyNet (Levie et al., 2018) is bounded by the 2-WL test.

Black et al. (2024) introduced several new WL algorithms based on absolute and relative positional encodings (PE). The authors further established a bunch of equivalence relationships among these algorithms. Notably, there exists a strong connection between the proposed "stack of power of matrices" PE and Spectral Invariant GNNs. We can prove that the proposed $(I, L, \cdots, L^{2n-1})$-WL (see Theorem 4.6 in Black et al. (2024)) is as expressive as spectral invariant GNNs with matrix $L$, and similarly, $(I, A, \cdots, A^{2n-1})$-WL is as expressive as spectral invariant GNNs with the ordinary adjacency matrix. Therefore, all results in our paper can be used to understand the power of these WL variants. Since Zhang et al. (2024b) has shown that the expressive power of RD-WL is bounded by Spectral Invariant GNNs, it follows that the proposed $L^\dagger$-WL (see Theorem 4.6 in Black et al. (2024)) is also bounded in expressive power by Spectral Invariant GNNs. This conclusion reproduces their key result (Theorem 4.4 in Black et al. (2024)).

**Homomorphism Count and Subgraph Count.**    Subgraph counting is a fundamental problem in chemical and biological tasks, as the ability to count subgraphs is strongly correlated with the performance of GNN in molecular prediction tasks. Kanatsoulis & Ribeiro studies subgraph counting power for a novel GNN framework, where classic message-passing GNNs are enhanced with random node features, and the GNN output is computed by taking the expectation over the introduced randomness. The paper demonstrates that such GNNs can learn to count various substructures, including cycles and cliques. These findings share similarities with our work, as both studies characterize the cycle-counting power of certain GNN models. Notably, the GNN framework proposed in Kanatsoulis & Ribeiro can count more complex substructures, such as 4-cliques and 8-cycles, which exceed the expressive power of 2-FWL.

Moreover, based on the foundational theory of Lovász (2012); Curticapean et al. (2017), it follows that the subgraph counting power of a GNN can be inferred from its ability to count homomorphisms (Seppelt, 2023; Zhang et al., 2024a). Consequently, recent research has also focused on the homomorphism counting power of GNNs. Dell et al. (2018) demonstrates that two graphs have the same representation under the $k$-WL algorithm if and only if the number of homomorphisms to the two graphs from any substructure with bounded tree width $k$ is equal. Additionally, Zhang et al. (2024a) introduce the concept of homomorphism expressivity as a quantitative framework for assessing the expressive power of GNNs. This paper specifically focuses on the subgraph counting

power of spectral invariant GNNs. Related works in this area include Cvetkovic et al. (1997), which shows that the graph angles (which can be determined through projection) are capable of counting all cycles of length up to 5, and Lim et al. (2023), which demonstrates that GNNs can count cycles with up to 5 vertices. A detailed comparison of our results with these previous studies is provided in the main text.

## B  PROOF OF THEOREM 3.3

### B.1  PREPARATION: PARALLEL TREE AND UNFOLDING TREE

#### B.1.1  ADDITIONAL EXPLANATION FOR PARALLEL TREE

For the reader's convenience, we begin by restating the definition of the parallel tree, as introduced in the main paper.

**Definition B.1** (**Parallel Edge:**). We denote a graph $G$ as a *parallel edge* if there exist vertices $u, v \in V_G$ such that the edge set $E_G$ can be partitioned into a sequence of simple paths $P_1, \ldots, P_m$, where each path has endpoints $(u, v)$. We refer to $(u, v)$ as the endpoints of the parallel edge $G$.

**Definition B.2** (**Parallel Tree:**). We define a graph $F$ as a *parallel tree* if there exists a tree $T$ such that we can obtain a graph isomorphic to $F$ by replacing each edge $(u, v) \in E_T$ with a parallel edge having endpoints $(u, v)$. We refer to $T$ as the *parallel tree skeleton* of the graph $F$. Additionally, we denote the minimum depth of any parallel tree skeleton of $F$ as the *parallel tree depth* of $F$.

We further define parallel tree decomposition for any parallel tree as follows:

**Definition B.3** (**Parallel tree decomposition**). For a parallel tree $F = (V_F, E_F)$, its parallel tree decomposition involves constructing a rooted tree $T^r = (V_{T^r}, E_{T^r})$ along with mapping functions $\beta_{T^r}$ and $\gamma_{T^r}$ that satisfy the following conditions:

1. The label function for nodes, $\beta_{T^r} : V_{T^r} \to V_F$, maps each node in $T^r$ to a unique vertex in $F$.

2. Let $\mathcal{E}_F$ denote the union of all paths in the graph $F$. The edge label function, $\gamma_{T^r} : E_{T^r} \to 2^{\mathcal{E}_F}$, satisfies the condition that for all $(t_1, t_2) \in E_{T^r}$, each $P \in \gamma_{T^r}(t_1, t_2)$ is a path connecting $\beta_{T^r}(t_1)$ and $\beta_{T^r}(t_2)$. Moreover, for each edge $e \in E_F$, there exists a unique tuple $(t_1, t_2, P)$, where $(t_1, t_2) \in V_T \times V_T$ and $P \in \gamma_T(t_1, t_2)$, such that $e$ lies on the path $P$.

We denote $T^r = (V_{T^r}, E_{T^r}, \beta_{T^r}, \gamma_{T^r})$ as the decomposition skeleton of graph $F$, and the ordered pair $(F, T^r)$ as a parallel-tree decomposed graph.

Let $\mathcal{S}^{pt}$ denote the set of all parallel trees, and we use $\mathcal{S}_d^{pt}$ to denote the set of all parallel trees whose parallel tree skeleton has depth at most $d$.

#### B.1.2  UNFOLDING TREE OF SPECTRAL INVARIANT GNN

We now introduce a process of constructing a parallel tree from any vertex of a given graph.

**Definition B.4** (**Constructing an unfolding tree of spectral invariant GNN**). Given a graph $G$, vertex $u \in V(G)$ and a non-negative integer $d$, the depth-$d$ spectral GNN unfolding tree of graph $G$ at vertex $u$, denoted as $(F_G^{(d)}(u), T_G^{(d)}(u))$, is a parallel-tree decomposed graph constructed as follows: At the beginning, $F = \{u\}$, and T only has a root node $r$ with $\beta_{T^r}(r) = \{u\}$. We can define a mapping $\pi : V_F \to V_G$ as $\pi(u) = u$.

For each leaf node $t$ in $T^r$, do the following procedure: Let $\beta_{T^r}(t) = x$. For each $w \in V_G$, add a fresh node $t_w$ to $T^r$ and designate $t$ as its parent. Then, consider the following case:

1. If $w \neq \pi(x)$, add $x_w$ to $F$ and extend $\pi$ with $\pi(x_w) = w$. We define $\beta_{T^r}(t_w) = x_w$. For every walk $w = v_1, v_2, \ldots, v_n = \pi(x)$ with $n \leq |V_G|$, where $v_1 = \pi(x), v_n = w$, we introduce a path $x_{v_1}, x_{v_2}, \ldots, x_{v_n}$ linking $x_w$ and $x$ to graph $F$, where $x_{v_1} = x, x_{v_n} = x_w$. We can also extend mapping $\pi$ with $\pi(x_{v_1}) = v_1, \pi(x_{v_2}) = v_2, \ldots, \pi(x_{v_n}) = v_n$. We define $\gamma_{T^r}(t, t_w)$ to be the set of all path $x_{v_1}, x_{v_2}, \ldots, x_{v_n}$ connecting $x$ and $x_w$ introduced in this step.

2. If $w = \pi(x)$, we define $\beta_{T^r}(t_w) = x$. Similarly, for every walk $w = v_1, v_2, \ldots, v_n = \pi(x)$ with $n \leq |V_G|$, we introduce a loop $x_{v_1}, x_{v_2}, \ldots, x_{v_n}$ to graph $F$, where $x_{v_1} = x = x_{v_n}$. We can also extend mapping $\pi$ with $\pi(x_{v_1}) = v_1, \pi(x_{v_2}) = v_2, \ldots, \pi(x_{v_n}) = v_n$. We define

$\gamma_{T^r}(t, t_w)$ to be the set of all path $x_{v_1}, x_{v_2}, \ldots, x_{v_n}$ connecting $x$ and $x_w$ introduced in this step.

We terminate the process once $T^r$ becomes a complete tree of depth $d$.

The following fact is straightforward from the construction of the unfolding tree:

**Fact B.5.** *For any graph $G$, any vertex $u \in V_G$, and any non-negative integer $D$, there is a homomorphism from $F_G^{(D)}(u)$ to $G$.*

With additional Explanation for parallel tree and construction of unfolding tree, we are now ready to prove Theorem 3.3 step by step.

### B.2 STEP 1: EQUIVALENCE OF ENCODING WALK INFORMATION AND SPECTRAL INFORMATION

In this section, we aim to prove Lemma 3.17. The key idea is to use the Cayley-Hamilton theorem to demonstrate that the walk-encoding GNN, as defined in Lemma 3.17, is equivalent to the spectral invariant GNN.

#### B.2.1 PROOF OF LEMMA 3.17

**Lemma B.6.** *Let $G = (V_G, E_G)$ be a graph, with its adjacency matrix denoted by $\boldsymbol{A}$. For vertices $x, y \in V_G$, define $\omega_G^k(x, y) = \boldsymbol{A}_{x,y}^k$ for all $k \in \{0, 1, 2, \ldots, |V_G|\}$, which represents the number of $k$-walks from vertex $x$ to vertex $y$. Define the tuple $\omega_G^*(x, y) = (\omega_G^0(x, y), \omega_G^1(x, y), \ldots, \omega_G^{n-1}(x, y))$, where $n = |V_G|$. Define the walk-encoding GNN with the following update rule:*

$$\chi_G^{\mathsf{Walk}, (d+1)}(x) = \mathsf{hash}(\chi_G^{\mathsf{Walk}, (d)}(x), \{\!\{(\omega_G^*(x, y), \chi_G^{\mathsf{Walk}, (d)}(y)) \mid y \in V_G\}\!\}).$$

*The walk-encoding GNN outputs a graph invariant $\chi_G^{\mathsf{Walk}, (d)}(G) = \{\!\{\chi_G^{\mathsf{Walk}, (d)}(u) | u \in V_G\}\!\}$. For any graphs $G$ and $H$, we have $\chi_G^{\mathsf{Walk}, (d)}(G) = \chi_H^{\mathsf{Walk}, (d)}(H)$ if and only if $\chi_G^{\mathsf{Spec}, (d)}(G) = \chi_H^{\mathsf{Spec}, (d)}(H)$.*

*Proof.* We begin by proving the following statement: If the spectra of graph $G$ and graph $H$ are identical (denoted as $(\lambda_1, \lambda_2, \ldots, \lambda_m)$), then for $x, u \in V_G$ and $y, v \in V_H$, $\mathcal{P}(x, u) = \mathcal{P}(y, v)$ if and only if $\omega_G^*(x, u) = \omega_H^*(y, v)$.

1. First, we prove that if $\mathcal{P}(x, u) = \mathcal{P}(y, v)$, then $\omega_G^*(x, u) = \omega_H^*(y, v)$.

   By the properties of diagonalizable matrices, for any $k \in \{1, 2, \ldots, |V_G|\}$, we have:

   $$\omega_G^k(x, u) = \lambda_1^k \boldsymbol{P}_{\lambda_1}(x, u) + \lambda_2^k \boldsymbol{P}_2(x, u) + \cdots + \lambda_m^k \boldsymbol{P}_{\lambda_m}(x, u).$$

   Therefore, if

   $$\boldsymbol{P}_{\lambda_r}(x, u) = \boldsymbol{P}_{\lambda_r}(y, v), \quad \forall r \in [m],$$

   it follows that:

   $$\omega_G^k(x, u) = \sum_{r=1}^{m} \lambda_r^k \boldsymbol{P}_{\lambda_r}(x, u) = \sum_{r=1}^{m} \lambda_r^k \boldsymbol{P}_{\lambda_r}(y, v) = \omega_H^k(y, v).$$

   Thus, we have proven the first direction of the statement.

2. Now, we prove that if $\omega_G^*(x, u) = \omega_H^*(y, v)$, then $\mathcal{P}(x, u) = \mathcal{P}(y, v)$.

   Let $\boldsymbol{A}_G$ and $\boldsymbol{A}_H$ denote the adjacency matrices of graphs $G$ and $H$, respectively. By the Cayley-Hamilton theorem, the minimal annihilating polynomial of matrix $\boldsymbol{A}_G$ is given by:

   $$f(\lambda) = (\lambda - \lambda_1)(\lambda - \lambda_2) \cdots (\lambda - \lambda_m).$$

   For each $r \in \{1, 2, \ldots, m\}$, the eigenspace corresponding to eigenvalue $\lambda_r$ is $\mathbf{Ker}(\lambda_r I - \boldsymbol{A}_G)$. Since:

   $$\mathbb{R}^n = \mathbf{Ker}(\lambda_1 I - \boldsymbol{A}_G) \oplus \mathbf{Ker}(\lambda_2 I - \boldsymbol{A}_G) \oplus \cdots \oplus \mathbf{Ker}(\lambda_m I - \boldsymbol{A}_G),$$

for each $r \in \{1, 2, \ldots, m\}$, the projection matrix onto the kernel space $\mathbf{Ker}(\lambda_r I - \mathbf{A}_G)$ is:

$$f_r(\mathbf{A}_G) = \prod_{j \neq r} (\lambda_j I - \mathbf{A}_G) = \mathbf{P}_{\lambda_r}.$$

Therefore, there exist coefficients $c_0^r, \ldots, c_{m-1}^r$ such that:

$$\mathbf{P}_{\lambda_r}(x, u) = c_0^r \cdot \omega_G^0(x, u) + c_1^r \cdot \omega_G^1(x, u) + \cdots + c_{m-1}^r \cdot \omega_G^{m-1}(x, u),$$
$$\mathbf{P}_{\lambda_r}(y, v) = c_0^r \cdot \omega_H^0(y, v) + c_1^r \cdot \omega_H^1(y, v) + \cdots + c_{m-1}^r \cdot \omega_H^{m-1}(y, v).$$

Finally, we conclude that if $\omega_G^\star(x, u) = \omega_H^\star(y, v)$, then $\mathcal{P}(x, u) = \mathcal{P}(y, v)$ for all $x, u \in V_G$ and $y, v \in V_H$.

Armed with the statement proven above, we are now prepared to prove Lemma 3.17. We will prove the two directions of the lemma separately as follows:

1. First, we prove that if $\chi_G^{\mathsf{Spec}}(G) = \chi_H^{\mathsf{Spec}}(H)$, then $\chi_G^{\mathsf{Walk}}(G) = \chi_H^{\mathsf{Walk}}(H)$. To do so, it suffices to show that for all $t \in \mathbb{N}$, if $\chi_G^{\mathsf{Spec},(t)}(u) = \chi_H^{\mathsf{Spec},(t)}(v)$ for all $(u, v) \in V_G \times V_H$, then $\chi_G^{\mathsf{Walk},(t)}(u) = \chi_H^{\mathsf{Walk},(t)}(v)$.

   We prove this by induction. Initially, the statement holds trivially for $t = 0$. We then assume the statement holds for $t = d$ and aim to prove it for $t = d + 1$. If $\chi_G^{\mathsf{Spec},(d+1)}(u) = \chi_H^{\mathsf{Spec},(d+1)}(v)$, then the following conditions are satisfied:

   $$
   \begin{aligned}
   &\chi_G^{\mathsf{Spec},(d)}(u) = \chi_H^{\mathsf{Spec},(d)}(v), \\
   &\{\!\{(\mathcal{P}(u, x), \chi_G^{\mathsf{Spec},(d)}(x)) \mid x \in V_G\}\!\} = \{\!\{(\mathcal{P}(v, y), \chi_H^{\mathsf{Spec},(d)}(y)) \mid y \in V_H\}\!\}.
   \end{aligned}
   \tag{1}
   $$

   For any $x \in V_G$ and $y \in V_H$, if $(\mathcal{P}(u, x), \chi_G^{\mathsf{Spec},(d)}(x)) = (\mathcal{P}(v, y), \chi_H^{\mathsf{Spec},(d)}(y))$, then by our previous result and the induction hypothesis, we have:

   $$(\omega_G^\star(u, x), \chi_G^{\mathsf{Walk},(d)}(x)) = (\omega_H^\star(v, y), \chi_H^{\mathsf{Walk},(d)}(y)). \tag{2}$$

   By combining equation 1 and equation 2, we conclude:

   $$
   \begin{aligned}
   &\chi_G^{\mathsf{Walk},(d)}(u) = \chi_H^{\mathsf{Walk},(d)}(v), \\
   &\{\!\{(\omega_G^\star(u, x), \chi_G^{\mathsf{Walk},(d)}(x)) \mid x \in V_G\}\!\} = \{\!\{(\omega_H^\star(v, y), \chi_H^{\mathsf{Walk},(d)}(y)) \mid y \in V_H\}\!\}.
   \end{aligned}
   $$

   Thus, we conclude that $\chi_G^{\mathsf{Walk},(d+1)}(u) = \chi_H^{\mathsf{Walk},(d+1)}(v)$. Therefore, we have proven that $\chi_G^{\mathsf{Spec}}(G) = \chi_H^{\mathsf{Spec}}(H)$ implies $\chi_G^{\mathsf{Walk}}(G) = \chi_H^{\mathsf{Walk}}(H)$.

2. Now, we prove the converse: if $\chi_G^{\mathsf{Walk}}(G) = \chi_H^{\mathsf{Walk}}(H)$, then $\chi_G^{\mathsf{Spec}}(G) = \chi_H^{\mathsf{Spec}}(H)$. Initially, $\chi_G^{\mathsf{Walk}}(G) = \chi_H^{\mathsf{Walk}}(H)$ implies $\{\!\{\chi_G^{\mathsf{Walk},(1)}(u) \mid u \in V_G\}\!\} = \{\!\{\chi_H^{\mathsf{Walk},(1)}(v) \mid v \in V_H\}\!\}$. If $\chi_G^{\mathsf{Walk},(1)}(u) = \chi_H^{\mathsf{Walk},(1)}(v)$, then $\omega_G^\star(u, u) = \omega_H^\star(v, v)$. This leads to:

   $$\{\!\{\omega_G^\star(u, u) \mid u \in V_G\}\!\} = \{\!\{\omega_H^\star(v, v) \mid v \in V_H\}\!\}.$$

   Hence, we derive that for all $k \in [n]$:

   $$\mathrm{tr}\left(\mathbf{A}_G^k\right) = \sum_{u \in V_G} \mathbf{A}_G^k(u, u) = \sum_{u \in V_G} \omega_G^k(u, u) = \sum_{v \in V_H} \omega_H^k(v, v) = \sum_{v \in V_H} \mathbf{A}_H^k(v, v) = \mathrm{tr}\left(\mathbf{A}_H^k\right).$$

   By standard results from linear algebra, the spectra of graphs $G$ and $H$ must be identical.

   Similar to the first direction, we now prove that for all $t \in \mathbb{N}$, if $\chi_G^{\mathsf{Walk},(t)}(u) = \chi_H^{\mathsf{Walk},(t)}(v)$ for all $(u, v) \in V_G \times V_H$, then $\chi_G^{\mathsf{Spec},(t)}(u) = \chi_H^{\mathsf{Spec},(t)}(v)$.

   We again proceed by induction. Initially, the statement holds trivially for $t = 0$. Assuming the statement holds for $t = d$, we aim to prove it for $t = d + 1$. If $\chi_G^{\mathsf{Walk},(d+1)}(u) = \chi_H^{\mathsf{Walk},(d+1)}(v)$, we have:

   $$
   \begin{aligned}
   &\chi_G^{\mathsf{Walk},(d)}(u) = \chi_H^{\mathsf{Walk},(d)}(v), \\
   &\{\!\{(\omega_G^\star(u, x), \chi_G^{\mathsf{Walk},(d)}(x)) \mid x \in V_G\}\!\} = \{\!\{(\omega_H^\star(v, y), \chi_H^{\mathsf{Walk},(d)}(y)) \mid y \in V_H\}\!\}.
   \end{aligned}
   $$

According to the statement proven earlier, for any $x \in V_G$ and $y \in V_H$, $\omega_G^\star(u, x) = \omega_H^\star(v, y)$ implies that $\mathcal{P}(u, x) = \mathcal{P}(v, y)$. Thus, we obtain:

$$\chi_G^{\mathsf{Spec},(d)}(u) = \chi_H^{\mathsf{Spec},(d)}(v),$$
$$\{\!\{(\mathcal{P}(u, x), \chi_G^{\mathsf{Spec},(d)}(x)) \mid x \in V_G\}\!\} = \{\!\{(\mathcal{P}(v, y), \chi_H^{\mathsf{Spec},(d)}(y)) \mid y \in V_H\}\!\}.$$

Therefore, we conclude that $\chi_G^{\mathsf{Spec},(d+1)}(u) = \chi_H^{\mathsf{Spec},(d+1)}(v)$. Finally, we have proven that $\chi_G^{\mathsf{Walk}}(G) = \chi_H^{\mathsf{Walk}}(H)$ implies $\chi_G^{\mathsf{Spec}}(G) = \chi_H^{\mathsf{Spec}}(H)$.

By combining both directions, we conclude that for any two graphs $G$ and $H$, $\chi_G^{\mathsf{Walk}}(G) = \chi_H^{\mathsf{Walk}}(H)$ if and only if $\chi_G^{\mathsf{Spec}}(G) = \chi_H^{\mathsf{Spec}}(H)$. Hence, the walk-encoding GNN is as expressive as the spectral-invariant GNN.

$\square$

### B.3   STEP 2: FINDING THE HOMOMORPHIC EXPRESSIVITY

We first define the isomorphism between parallel-tree decomposed graphs.

**Definition B.7.** Given two parallel-tree decomposed graphs $(F, T^r)$ and $(\tilde{F}, \tilde{T}^r)$, a pair of mappings $(\rho, \tau)$ is called an isomorphism from $(F, T^r)$ to $(\tilde{F}, \tilde{T}^r)$, denoted by $(F, T^r) \cong (\tilde{F}, \tilde{T}^r)$, if the following hold:

1. $\rho$ is an isomorphism from $F$ to $\tilde{F}$, while $\tau$ is an isomorphism from $T^r$ to $\tilde{T}^r$ (ignoring labels $\beta$ and $\gamma$).

2. For any $t \in V_{T^r}$, $\rho(\beta_{T^r}(t)) = \beta_{\tilde{T}^r}(\tau(t))$. Moreover, for any $(t_1, t_2) \in E_{T^r}$, $\rho(\gamma_{T^r}(t_1, t_2)) = \gamma_{T^r}(\tau(t_1), \tau(t_2))$

**Theorem B.8.** *For any two graphs $G, H$, any vertices $u \in V_G$, $x \in V_H$, and any non-negative integer $D$, $\chi_G^{\mathsf{Walk},(D)}(u) = \chi_H^{\mathsf{Walk},(D)}(x)$ iff there exists an isomorphism $(\rho, \tau)$ from $(F_G^{(D)}(u), T_G^{(D)}(u))$ to $(F_H^{(D)}(x), T_H^{(D)}(x))$ such that $\rho(u) = x$.*

*Proof.* The proof proceeds by induction on $D$. The base case is straightforward: for $D = 0$, the theorem holds trivially. Now assume the theorem holds for all $D \leq d$, and we will prove it for $D = d + 1$.

We first prove that $\chi_G^{\mathsf{Walk},(d+1)}(u) = \chi_H^{\mathsf{Walk},(d+1)}(x)$ implies the existence of an isomorphism $(\rho, \tau)$ from $(F_G^{(d+1)}(u), T_G^{(d+1)}(u))$ to $(F_H^{(d+1)}(x), T_H^{(d+1)}(x))$ such that $\rho(u) = x$. Given that $\chi_G^{(d+1)}(u) = \chi_H^{(d+1)}(x)$, it follows that:

$$\{\!\{\omega_G^*(u, v), \chi_G^{\mathsf{Walk},(d)}(v)\}\!\}_{v \in V_G} = \{\!\{\omega_H^*(x, y), \chi_H^{\mathsf{Walk},(d)}(y)\}\!\}_{y \in V_H}.$$

Let $n = |V_G| = |V_H|$, and denote $V_G = \{v_1, v_2, \ldots, v_n\}$, $V_H = \{y_1, y_2, \ldots, y_n\}$ such that:

$$\omega_G^*(u, v_i) = \omega_H^*(x, y_i), \quad \chi_G^{\mathsf{Walk},(d)}(v_i) = \chi_H^{\mathsf{Walk},(d)}(y_i) \quad \text{for all } i \in [n].$$

By the definition of tree unfolding, we have:

$$F_G^{(d+1)}(u) = \left( \bigcup_{v_i} F_G^{(d)}(v_i) \right) \cup F_G^{(1)}(u), \quad F_H^{(d+1)}(x) = \left( \bigcup_{y_i} F_H^{(d)}(y_i) \right) \cup F_H^{(1)}(x),$$

where we use $\cup$ to represent graph union. By the inductive hypothesis, there exists an isomorphism $(\rho_i, \tau_i)$ from $(F_G^{(d)}(v_i), T_G^{(d)}(v_i))$ to $(F_H^{(d)}(y_i), T_H^{(d)}(y_i))$ such that $\rho_i(v_i) = y_i$. Additionally, since $\omega_G^*(u, v_i) = \omega_H^*(x, y_i)$, $F_G^{(1)}(u)$ is isomorphic to $F_H^{(1)}(x)$. Therefore, by merging all $\rho_i$ and $\tau_i$ into $\tilde{\rho}$ and $\tilde{\tau}$, and constructing an approximate mapping between tree nodes at depth no more than 1 in $T_G^{(d+1)}(u)$ and $T_H^{(d+1)}(x)$, it follows that $(\tilde{\rho}, \tilde{\tau})$ is a well-defined isomorphism from $(F_G^{(d+1)}(u), T_G^{(d+1)}(u))$ to $(F_H^{(d+1)}(x), T_H^{(d+1)}(x))$, satisfying $\tilde{\rho}(u) = x$.

Next, we prove that if there exists an isomorphism $(\rho, \tau)$ between the parallel-tree decomposed graphs $(F_G^{(d+1)}(u), T_G^{(d+1)}(u))$ and $(F_H^{(d+1)}(x), T_H^{(d+1)}(x))$ such that $\rho(u) = x$, then

$\chi_G^{\mathsf{Walk},(d+1)}(u) = \chi_H^{\mathsf{Walk},(d+1)}(x)$. Since $\tau$ is an isomorphism from $T_G^{(d+1)}(u)$ to $T_H^{(d+1)}(x)$, it maps all depth-1 nodes in $T_G^{(d+1)}(u)$ to depth-1 nodes in $T_H^{(d+1)}(x)$. Let $s_1, s_2, \ldots, s_n$ be the depth-1 nodes in $T_G^{(d+1)}(u)$, and $t_1, t_2, \ldots, t_n$ be the corresponding nodes in $T_H^{(d+1)}(x)$. For $i \in [n]$, we denote the subtree induced by $s_i$ and its descendants as $T_{G,s_i}^{(d+1)}(u)$, and similarly, the subtree induced by $t_i$ and its descendants as $T_{G,t_i}^{(d+1)}(x)$. Additionally, we define the subgraph of $F_G^{(d+1)}(u)$ induced by $T_{G,s_i}^{(d+1)}(u)$ as $F_{G,s_i}^{(d+1)}(u)$. Likewise, we define the subgraph of $F_H^{(d+1)}(u)$ induced by $T_{H,t_i}^{(d+1)}(u)$ as $F_{H,t_i}^{(d+1)}(u)$. Without loss of generality, we assume the following:

- $\tau$ is an isomorphism from the subtree $T_{G,s_i}^{(d+1)}(u)$ to $T_{H,t_i}^{(d+1)}(x)$.
- For all $s \in V_{T_{G,s_i}^{(d+1)}(u)}$, $\rho(\beta_{T_G^{(d+1)}(u)}(s)) = \beta_{T_H^{(d+1)}(x)}(\tau(s))$.
- For all $e \in E_{T_{G,s_i}^{(d+1)}(u)}$, $\rho(\gamma_{T_G^{(d+1)}(u)}(e)) = \gamma_{T_H^{(d+1)}(x)}(\tau(e))$.
- $\rho$ is an isomorphism between the subgraphs $F_{G,s_i}^{(d+1)}(u)$ and $F_{H,t_i}^{(d+1)}(x)$.

According to our assumption, $(F_{G,s_i}^{(d+1)}(u), T_{G,s_i}^{(d+1)}(u))$ is isomorphic to $(F_{H,t_i}^{(d+1)}(x), T_{H,t_i}^{(d+1)}(x))$. Additionally, by the definition of the unfolding tree, $(F_{G,s_i}^{(d+1)}(u), T_{G,s_i}^{(d+1)}(u))$ is isomorphic to the depth-$d$ unfolding tree $(F_G^{(d)}(v_i), T_G^{(d)}(v_i))$ for some $v_i \in V_G$. Similarly, $(F_{H,t_i}^{(d+1)}(x), T_{H,t_i}^{(d+1)}(x))$ is isomorphic to $(F_H^{(d)}(y_i), T_H^{(d)}(y_i))$ for some $y_i \in V_H$. By induction, we know that $\chi_G^{\mathsf{Walk},(d)}(v_i) = \chi_H^{\mathsf{Walk},(d)}(y_i)$ and $\omega_G^*(u, v_i) = \omega_H^*(x, y_i)$. Therefore, we conclude:

$$\left( \omega_G^*(u, v_i), \chi_G^{\mathsf{Walk},(d)}(v_i) \right) = \left( \omega_H^*(x, y_i), \chi_H^{\mathsf{Walk},(d)}(y_i) \right)$$

for all $i \in [n]$, implying that:

$$\{\!\!\{ \left( \omega_G^*(u, v_i), \chi_G^{\mathsf{Walk},(d)}(v_i) \right) \}\!\!\}_{v_i \in V_G} = \{\!\!\{ \left( \omega_H^*(x, y_i), \chi_H^{(d)}(y_i) \right) \}\!\!\}_{y_i \in V_H}. \qquad (3)$$

It remains to prove that $\chi_G^{\mathsf{Walk},(d)}(u) = \chi_H^{\mathsf{Walk},(d)}(x)$. To prove this, note that equation 3 implies that

$$\{\!\!\{ \left( \omega_G^*(u, v_i), \chi_G^{\mathsf{Walk},(d')}(v_i) \right) \}\!\!\}_{y_i \in V_G} = \{\!\!\{ \left( \omega_H^*(x, y_i), \chi_H^{\mathsf{Walk},(d')}(y_i) \right) \}\!\!\}_{y_i \in V_H}.$$

holds for all $0 \le d' \le d$. Combined this with the fact that $\chi_G^{\mathsf{Walk},(0)}(u) = \chi_H^{\mathsf{Walk},(0)}(x)$, we can incrementally prove that $\chi_G^{\mathsf{Walk},(d')}(u) = \chi_H^{\mathsf{Walk},(d')}(x)$ for all $d' \le d + 1$. We have thus concluded the proof. Thus, the proof is complete. $\qquad \square$

**Definition B.9.** Given a graph $G$ and a parallel-tree decomposed graph $(F, T^r)$, we define the function $\mathtt{treeCount}((F, T^r), G)$ as the number of ordered pairs $(u, d) \in V_G \times \mathbb{N}$ such that the depth-$d$ unfolding tree $(F_G^{(d)}(u), T_G^{(d)}(u))$ at vertex $u$ is isomorphic to $(F, T^r)$.

**Corollary B.10.** *For any graph $G, H$, $\chi_G^{\mathsf{Walk}}(G) = \chi_H^{\mathsf{Walk}}(H)$ iff $\mathtt{treeCount}((F, T^r), G) = \mathtt{treeCount}((F, T^r), H)$ holds for all parallel-tree decomposed graph $(F, T^r)$.*

*Proof.* We first prove one direction of the corollary. We aim to prove that if $\chi_G^{\mathsf{Walk}}(G) = \chi_H^{\mathsf{Walk}}(H)$, then $\mathtt{treeCount}((F, T^r), G) = \mathtt{treeCount}((F, T^r), H)$. If $\chi_G^{\mathsf{Walk}}(G) = \chi_H^{\mathsf{Walk}}(H)$, then $\{\!\!\{ \chi_G^{\mathsf{Walk}}(u) : u \in V_G \}\!\!\} = \{\!\!\{ \chi_H^{\mathsf{Walk}}(x) : x \in V_H \}\!\!\}$. For each color $c$ in the above multiset, pick $u \in V_G$ with $\chi_G^{\mathsf{Walk}}(u) = c$. It follows that if $(F, T^r) \cong (F_G^{(D)}(u), T_G^{(D)}(u))$ for some $D$, then $\mathtt{treeCount}((F, T^r), G) = |\{\!\!\{ u \in V_G : \chi_G^{\mathsf{Walk}}(u) = c \}\!\!\}| = |\{\!\!\{ x \in V_H : \chi_H(x) = c \}\!\!\}| = \mathtt{treeCount}((F, T^r), H)$ by Theorem B.8. On the other hand, if $(F, T^r) \not\cong (F_G^{(D)}(u), T_G^{(D)}(u))$ for all $u \in V_G$ and all $D$, then clearly $\mathtt{treeCount}((F, T^r), G) = \mathtt{treeCount}((F, T^r), H) = 0$.

We then aim to prove the second direction of the corollary. If $\mathtt{treeCount}((F, T^r), G) = \mathtt{treeCount}((F, T^r), H)$ holds for all parallel-tree decomposed graph $(F, T^r)$, it clearly holds for all $(F_G^{(D)}(u), T_G^{(D)}(u))$ with $u \in V_G$ and a sufficiently large $D$. This guarantees that for all color $c$, $|\{\!\!\{ u \in V_G : \chi_G^{\mathsf{Walk}}(u) = c \}\!\!\}| = |\{\!\!\{ x \in V_H : \chi_H^{\mathsf{Walk}}(x) = c \}\!\!\}|$ by Theorem B.8. Therefore, $\{\!\!\{ \chi_G^{\mathsf{Walk}}(u) : u \in V_G \}\!\!\} = \{\!\!\{ \chi_H^{\mathsf{Walk}}(x) : x \in V_H \}\!\!\}$, concluding the proof. $\qquad \square$

**Definition B.11.** For parallel-tree decomposed graph $(F, T^r)$, we use $\mathsf{Dep}(T^r)$ to denote the depth of tree $T$. For any tree note $t \in V_T$, we use $\mathsf{dep}_T(t)$ to denote the depth of node $t$ in $T^r$.

Using techniques similar to those in Corollary B.10, we can derive a finite-iteration version of Corollary B.10 as follows:

**Corollary B.12.** *For any graphs $G$ and $H$, $\chi_G^{\mathsf{Walk},(d)}(G) = \chi_H^{\mathsf{Walk},(d)}(H)$ if and only if* $\mathsf{treeCount}\left((F, T^r), G\right) = \mathsf{treeCount}\left((F, T^r), H\right)$ *holds for all parallel-tree decomposed graphs* $(F, T^r)$ *with* $\mathsf{Dep}(T^r) \leq d$.

In the following theorem, we will bridge homomorphic count with unfolding tree count. Before presenting the result, we first introduce some notations used to present the theorem.

**Definition B.13.** Given two parallel-tree decomposed graphs $(F, T^r)$ and $(\tilde{F}, \tilde{T}^r)$, a pair of mappings $(\rho, \tau)$ is called a *strong homomorphism* from $(F, T^r)$ to $(\tilde{F}, \tilde{T}^s)$ if it satisfies the following conditions: First, $\tau$ is a homomorphism from $T$ to $\tilde{T}$, ignoring the labels $\beta$ and $\gamma$, and is depth-preserving, i.e., $\mathsf{dep}_{T^r}(t) = \mathsf{dep}_{\tilde{T}^s}(\tau(t))$ for all $t \in V_T$. Additionally, $\rho$ is a homomorphism from $F[\gamma_T(t_1, t_2)]$ to $\tilde{F}[\gamma_{\tilde{T}}(\tau(t_1), \tau(t_2))]$. Finally, the depth of $T^r$ is equal to the depth of $\tilde{T}^s$.

We use $\mathsf{strHom}((F, T^r), (\tilde{F}, \tilde{T}^r))$ to denote the set of all strong homomorphism from $(F, T^r)$ to $(\tilde{F}, \tilde{T}^r)$, and let $\mathsf{strhom}((F, T^r), (\tilde{F}, \tilde{T}^r)) = |\mathsf{strHom}((F, T^r), (\tilde{F}, \tilde{T}^r))|$.

**Theorem B.14.** *Let $(F, T^r)$ be parallel-tree decomposed graph and let $G$ be a graph. We have*

$$\mathsf{hom}(F, G) = \sum_{\left(\tilde{F}, \tilde{T}^r\right) \in \mathcal{S}^{pt}} \mathsf{strhom}\left((F, T^r), \left(\tilde{F}, \tilde{T}^r\right)\right) \cdot \mathsf{treeCount}(\left(\tilde{F}, \tilde{T}^r\right), G)$$

*Proof.* We assume that $\beta_{T^r}(r) = u$ for $(F, T^r)$, and the depth of $(F, T^r)$ is $d$. Let $x \in V_G$ be any vertex in $G$, and denote $(F_G^{(d)}(x), T_G^{(d)}(x))$ as the depth-$d$ unfolding tree at $x$. We define $S_1(x)$ as the set of all homomorphisms from $F$ to $G$ that map the vertex $u \in V_F$ to $x \in V_G$. Furthermore, we define $S_2(x)$ as the set of strong homomorphisms $(\rho, \tau)$ from $(F, T^r)$ to $(F_G^{(d)}(x), T_G^{(d)}(x))$, such that $\rho(u) = x$. Then Theorem B.14 is equivalent to the following equation: $\sum_{x \in V_G} |S_1(x)| = \sum_{x \in V_G} |S_2(x)|$. We will prove that $|S_1(x)| = |S_2(x)|$ for all $x \in V_G$. Given $x \in V_G$, according to Fact B.5, there exists a homomorphism $\pi$ from $F_G^{(d)}(x)$ to graph $G$. Define a mapping $\sigma$ such that $\sigma(\rho, \tau) = \pi \circ \rho$ for all $(\rho, \tau) \in S_2(x)$. It suffices to prove that $\sigma$ is a bijection from $S_2(x)$ to $S_1(x)$.

We first prove that $\sigma$ is a mapping from $S_2(x)$ to $S_1(x)$. Since $\rho$ is a homomorphism from $F$ to $F_G^{(d)}(x)$, and $\pi$ is a homomorphism from $F_G^{(d)}(x)$ to $G$. The composition of homomorphism is still a homomorphism. Therefore, $\pi \circ \rho$ is a homomorphism from $F$ to graph $G$.

We then prove that $\sigma$ is a surjection. For all $g \in S_1(x)$, we define a mapping $(\rho, \tau)$ from $(F, T^r)$ to $(F_G^{(d)}(x), T_G^{(d)}(x))$ as follows. First define $\rho(u) = x$ and set $\tau(r)$ to be the root of $(F_G^{(d)}(x), T_G^{(d)}(x))$. Let $v_1, v_2, \ldots, v_m \in V_{T^r}$ be the tree nodes of depth 1. Similarly, by definition of the unfolding tree, let $y_1, y_2, \ldots, y_n \in V_{F_G^{(d)}(x)}$ be tree nodes of depth 1. For all $i \in [m]$, we denote $\{P_{i1}, P_{i2}, \ldots, P_{ia_i}\} = \gamma_{T^r}(u, v_i)$, to be the paths associated with edge $(u, v_i) \in E_{T^r}$. Similarly, for $i \in [n]$ we denote $\{\tilde{P}_{i1}, \tilde{P}_{i2}, \ldots, \tilde{P}_{ib_i}\} = \gamma_{\tilde{T}^r}(x, y_i)$ to be the paths associated with edge $(x, y_i) \in E_{T_G^{(d)}(x)}$. Since $g$ and $\pi$ are both homomorphism, we have:

- For every $v_i (i \in [m])$, there exists $y_j$ $(j \in [n])$, such that $g(\beta_{T^r}(v_i)) = \pi(\beta_{T_G^{(d)}(x)}(y_j)) = \tilde{z}_j$ for some $\tilde{z}_j \in V_G$.

- For every path $P_{ik} \in \gamma_{T^r}(u, v_i)$ $(k \in [a_i])$ linking $u$ and $\beta_{T^r}(v_i)$, there exists $\tilde{P}_{jl}$ $(l \in [b_j])$ linking $x$ and $\beta_{T_G^{(d)}(x)}(y_j)$ such that $g(P_{ik}) = \pi(\tilde{P}_{jl})$.

We then define $\rho(\beta_{T^r}(v_i)) = \beta_{T_G^{(d)}(x)}(y_j)$ and $\rho(P_{ik}) = \tilde{P}_{jl}$ for each $i \in [m]$ and $k \in [a_i]$. Based on the above two items, one can easily define $\tau$ such that each node $s$ in $T^r$ of depth 1

is mapped by $\tau$ to a node $t$ in $T_G^{(d)}(x)$ of the same depth, such that $\rho(\beta_{T^r}(s)) = \beta_{T_G^{(d)}(x)}(t)$ and $\rho(\gamma_{T^r}(r,s)) = \gamma_{T_G^{(d)}(x)}(x,t)$. Continuing, we denote the subtree of $T^r$ induced by $s$ and all its descendants as $T_s^r$, and the subgraph of $F$ induced by $T_s^r$ as $F_s$. Similarly, we denote the subtree of $T_G^{(d)}(x)$ induced by $\tau(s)$ and its descendants as $T_{G,\tau(s)}^{(d)}(x)$, and the subgraph of $F_G^{(d)}(x)$ induced by $T_{G,\tau(s)}^{(d)}(x)$ as $F_{G,\tau(s)}^{(d)}(x)$. We can recursively define the image of $\rho$ on $F_s$ for each tree node of depth 1, following the same construction described above. This recursive definition holds because $g$ remains a homomorphism from $(F_s, T_s^r)$ to $G$, and $\pi$ remains a homomorphism from $(F_{G,\tau(s)}^{(d)}(x), T_{G,\tau(s)}^{(d)}(x))$ to $G$, with $g(\beta_{T^r}(s)) = \pi(\beta_{T_G^{(d)}(x)}(\tau(s)))$. By recursively applying this procedure, we can construct $(\rho, \tau)$ such that it becomes a strong homomorphism (denoted strHom) from $(F, T^r)$ to $(F_G^{(d)}(x), T_G^{(d)}(x))$. Therefore, we have shown that for any $g \in S_1(x)$, there exists a preimage $(\rho, \tau) \in S_2(x)$ such that $\sigma(\rho, \tau) = g$.

Finally, we prove that $\sigma$ is an injection.
Let $(\rho_1, \tau_1), (\rho_2, \tau_2) \in S_2(x)$ such that $\pi \circ \rho_1 = \pi \circ \rho_2$. Similar to previous item, we define $v_1, v_2, \ldots, v_m \in V_{T^r}$ to be the tree nodes of depth 1. Similarly, by definition of the unfolding tree, let $y_1, y_2, \ldots, y_n \in V_{T_G^{(d)}(x)}$ be tree nodes of depth 1.

- For all $i \in [m]$, we denote $\{P_{i1}, P_{i2}, \ldots, P_{ia_i}\} = \gamma_{T^r}(u, v_i)$, to be the paths associated with edge $(u, v_i) \in E_{T^r}$. Similarly, for $i \in [n]$ we denote $\{\tilde{P}_{i1}, \tilde{P}_{i2}, \ldots, \tilde{P}_{ib_i}\} = \gamma_{T^r}(x, y_i)$ to be the paths associated with edge $(x, y_i) \in E_{T_G^{(d)}(x)}$. For each $i \in [m]$, let $j_1(i)$ and $j_2(i)$ be indices satisfying $\rho_1(w_i) = x_{j_1(i)}$ and $\rho_2(w_i) = x_{j_2(i)}$. It follows that $\pi(x_{j_1(i)}) = \pi(x_{j_2(i)})$. By the definition of unfolding tree, we must have $x_{j_1(i)} = x_{j_2(i)}$, and thus $\rho_1(w_i) = \rho_2(w_i)$.

- For each $k \in [a_i]$, $P_{ik} \in \gamma_{T^r}(u, v_i)$, let $l_1(k)$ and $l_2(k)$ be indices satisfying $\rho_1(P_{ik}) = \tilde{P}_{jl_1(j)}$ and $\rho_2(P_{ik}) = \tilde{P}_{jl_2(j)}$, where we use $j$ to denote $j = j_1(i) = j_2(i)$. With similar analysis as the previous item, we have $\pi(\tilde{P}_{jl_1(j)}) = \pi(\tilde{P}_{jl_2(j)})$. By the definition of the unfolding tree, we must have $\tilde{P}_{jl_1(j)} = \tilde{P}_{jl_2(j)}$, and thus $\rho_1(P_{ik}) = \rho_2(P_{ik})$.

Next, we recursively apply the previously described procedure to the subtree induced by the tree node $s$ at depth 1 and its descendants, following the same steps outlined earlier. Through this process, we can ultimately demonstrate that $\rho_1 = \rho_2$. Consequently, $\sigma$ is injective.

Combining the above three parts completes the proof. □

**Theorem B.15.** *Let $(F, T^r)$ be parallel-tree decomposed graph with $\mathsf{Dep}(T^r) \leq d$ and let $G$ be a graph. We have*

$$\mathsf{hom}(F, G) = \sum_{(\tilde{F}, \tilde{T}^r) \in \mathcal{S}_d^{pt}} \mathsf{strhom}\left((F, T^r), \left(\tilde{F}, \tilde{T}^r\right)\right) \cdot \mathsf{treeCount}\left(\left(\tilde{F}, \tilde{T}^r\right), G\right)$$

*Proof.* According to the third condition in Definition B.13, for $(F, T^r) \in \mathcal{S}_d^{pt}$ and $(\tilde{F}, \tilde{T}^r) \in \mathcal{S}^{pt}$, if $\mathsf{strhom}((F, T^r), (\tilde{F}, \tilde{T}^r)) \neq 0$, then $\mathsf{Dep}(T^r) = \mathsf{Dep}(\tilde{T}^r)$. Therefore, we have $(\tilde{F}, \tilde{T}^r) \in \mathcal{S}_d^{pt}$. Thus, the conclusion of the lemma follows. □

**Definition B.16.** Given two parallel-tree decomposed graphs $(F, T^r)$ and $(\tilde{F}, \tilde{T}^r)$, along with a strong homomorphism $(\rho, \tau)$, we define $(\rho, \tau)$ as a *surjective strong homomorphism* if both $\rho$ and $\tau$ are surjective mappings, and as an *injective strong homomorphism* if both $\rho$ and $\tau$ are injective mappings. We denote the set of all surjective strong homomorphisms from $(F, T^r)$ to $(\tilde{F}, \tilde{T}^r)$ by $\mathsf{strSurj}((F, T^r), (\tilde{F}, \tilde{T}^r))$, and further define $\mathsf{strsurj}((F, T^r), (\tilde{F}, \tilde{T}^r)) = |\mathsf{strSurj}((F, T^r), (\tilde{F}, \tilde{T}^r))|$. Similarly, we denote the set of all injective strong homomorphisms from $(F, T^r)$ to $(\tilde{F}, \tilde{T}^r)$ by $\mathsf{strInj}((F, T^r), (\tilde{F}, \tilde{T}^r))$, and further define $\mathsf{strinj}((F, T^r), (\tilde{F}, \tilde{T}^r)) = |\mathsf{strInj}((F, T^r), (\tilde{F}, \tilde{T}^r))|$.

We now present the following lemma regarding the relationships between strong homomorphisms, surjective strong homomorphisms, and injective strong homomorphisms.

**Lemma B.17.** *For any parallel-tree decomposed graph $(F, T^r)$ and $(\tilde{F}, \tilde{T}^s)$, we have*

$$\mathsf{strhom}\left((F, T^r), \left(\tilde{F}, \tilde{T}^s\right)\right) = \sum_{(\hat{F}, \hat{T}^t) \in \mathcal{S}^{\mathsf{pt}}} \frac{\mathsf{strsurj}\left((F, T^r), \left(\hat{F}, \hat{T}^t\right)\right) \cdot \mathsf{strinj}\left(\left(\hat{F}, \hat{T}^t\right), \left(\tilde{F}, \tilde{T}^s\right)\right)}{\mathsf{aut}\left(\hat{F}, \hat{T}^t\right)},$$

*where $\mathsf{aut}(\hat{F}, \hat{T}^t)$ denotes the number of automorphism of $(\hat{F}, \hat{T}^r)$. Here, the summation ranges over all non-isomorphic (parallel-tree decomposed) graphs in $\mathcal{S}^{\mathsf{pt}}$ and is well-defined as there are only a finite number of graphs making the value in the summation non-zero.*

*Proof.* We initially define the set $S$ as the set of triples $((\hat{F}, \hat{T}^t), (\rho, \tau), (\phi, \psi))$ that satisfy $(\hat{F}, \hat{T}^t) \in \mathcal{S}^{\mathsf{pt}}$, $(\rho, \tau) \in \mathsf{strSurj}((F, T^r), (\hat{F}, \hat{T}^t))$, and $(\phi, \psi) \in \mathsf{strInj}((\hat{F}, \hat{T}^t), (\tilde{F}, \tilde{T}^s))$. We define a mapping $\sigma$ such that $\sigma((\hat{F}, \hat{T}^t), (\rho, \tau), (\phi, \psi)) = (\phi \circ \rho, \psi \circ \tau)$ for all $((\hat{F}, \hat{T}^t), (\rho, \tau), (\phi, \psi)) \in S$. Our goal is to prove that $\sigma$ is a mapping from $S$ to $\mathsf{strHom}((F, T^r), (\tilde{F}, \tilde{T}^s))$. Moreover, we aim to show that $\sigma((\hat{F}_1, \hat{T}_1^{t_1}), (\rho_1, \tau_1), (\phi_1, \psi_1)) = \sigma((\hat{F}_2, \hat{T}_2^{t_2}), (\rho_2, \tau_2), (\phi_2, \psi_2))$ if and only if there exists an isomorphism $(\hat{\rho}, \hat{\tau})$ from $(\hat{F}_1, \hat{T}_1^{t_1})$ to $(\hat{F}_2, \hat{T}_2^{t_2})$ such that $\hat{\rho} \circ \rho_1 = \rho_2$, $\hat{\tau} \circ \tau_1 = \tau_2$, $\phi_1 = \phi_2 \circ \hat{\rho}$, and $\psi_1 = \psi_2 \circ \hat{\tau}$.

We will prove these statements one by one. We first prove that $\sigma$ is a mapping from $S$ to $\mathsf{strHom}((F, T^r), (\tilde{F}, \tilde{T}^s))$. This simply follows from the fact that $\mathsf{strSurj}$ and $\mathsf{strInj}$ are both $\mathsf{strHom}$, and the composition of two $\mathsf{strHom}$s are still a $\mathsf{strHom}$.

Next, we will prove that $\sigma$ is surjective. Given $(\tilde{\rho}, \tilde{\tau}) \in \mathsf{strHom}((F, T^r), (\tilde{F}, \tilde{T}^s))$, we define $(\hat{F}, \hat{T}^r)$, $(\rho, \tau)$, and $(\phi, \psi)$ as follows:

1. We define $\hat{F}$ as the subgraph of $\tilde{F}$ induced by $\tilde{\rho}(V_F)$, and we define $\hat{T}^t$ as the subgraph of $\tilde{T}^s$ induced by $\tau^{\mathsf{H}}(V_T)$. We clearly have $(\hat{F}, \hat{T}^t) \in \mathcal{S}^{\mathsf{pt}}$.

2. Let $\rho = \tilde{\rho}$ and $\tau = \tilde{\tau}$. Obviously, $(\rho, \tau)$ is a $\mathsf{strSurj}$ from $(F, T^r)$ to $(\hat{F}, \hat{T}^t)$.

3. Define identity mappings $\phi(u) = u$ for all $u \in V_{\hat{F}}$ for $\psi(t) = t$ for all $t \in V_{\hat{T}}$. Obviously, $(\phi, \psi)$ is a $\mathsf{strInj}$ from $(\hat{F}, \hat{T}^t)$ to $(\tilde{F}, \tilde{T}^s)$.

We clearly have $\tilde{\rho} = \phi \circ \rho$ and $\tilde{\tau} = \psi \circ \tau$. Thus, $\sigma$ is a surjection.

We will now prove that $\sigma((\hat{F}_1, \hat{T}_1^{t_1}), (\rho_1, \tau_1), (\phi_1, \psi_1)) = \sigma((\hat{F}_2, \hat{T}_2^{t_2}), (\rho_2, \tau_2), (\phi_2, \psi_2))$ iff there exist an isomorphism $(\hat{\rho}, \hat{\tau})$ from $(\hat{F}_1, \hat{T}_1^{t_1})$ to $(\hat{F}_2, \hat{T}_2^{t_2})$ such that $\hat{\rho} \circ \rho_1 = \rho_2$, $\hat{\tau} \circ \tau_1 = \tau_2$, $\phi_1 = \phi_2 \circ \hat{\rho}$, $\psi_1 = \psi_2 \circ \hat{\tau}$. It suffices to prove only one direction, namely, $\sigma((\hat{F}_1, \hat{T}_1^{t_1}), (\rho_1, \tau_1), (\phi_1, \psi_1)) = \sigma((\hat{F}_2, \hat{T}_2^{t_2}), (\rho_2, \tau_2), (\phi_2, \psi_2))$ implies that there exist an isomorphism $(\hat{\rho}, \hat{\tau})$ from $(\hat{F}_1, \hat{T}_1^{t_1})$ to $(\hat{F}_2, \hat{T}_2^{t_2})$ such that $\hat{\rho} \circ \rho_1 = \rho_2$, $\hat{\tau} \circ \tau_1 = \tau_2$, $\phi_1 = \phi_2 \circ \hat{\rho}$, $\psi_1 = \psi_2 \circ \hat{\tau}$.

1. We first prove that $\hat{F}_1 \cong \hat{F}_2$ and $\hat{T}_1^{t_1} \cong \hat{T}_2^{t_2}$. For any $u, v \in V_F$, if $\rho_1(u) \neq \rho_1(v)$, then $\phi_1 \circ \rho_1(u) \neq \phi_1 \circ \rho_1(v)$ since $\phi$ is an injection. Therefore, $\phi_2 \circ \rho_2(u) \neq \phi_2 \circ \rho_2(v)$, and thus $\rho_2(u) \neq \rho_2(v)$. By symmetry, we also have that $\rho_2(u) \neq \rho_2(v)$ implies $\rho_1(u) \neq \rho_1(v)$. This proves that $\rho_1(u) = \rho_1(v)$ iff $\rho_2(u) = \rho_2(v)$. For any $u, v \in V_F$, if $\{\rho_1(u), \rho_1(v)\} \in E_{\hat{F}_1}$, then $\{u, v\} \in E_F$ since $\rho_1$ is a surjection. Therefore, $\{\rho_2(u), \rho_2(v)\} \in E_{\hat{F}_2}$ since $\rho_2$ is a homomorphism.

   By symmetry, it follows that if $\{\rho_2(u), \rho_2(v)\} \in E_{\hat{F}_2}$, then $\{\rho_1(u), \rho_1(v)\} \in E_{\hat{F}_1}$. Therefore, we conclude that $\hat{F}_1 \cong \hat{F}_2$. Similarly, it follows that $\hat{T}_1^{t_1} \cong \hat{T}_2^{t_2}$.

2. Consequently, there exist isomorphism $\hat{\rho}$ and $\hat{\tau}$ such that $\hat{\rho} \circ \rho_1 = \rho_2$, $\hat{\tau} \circ \tau_1 = \tau_2$. For any node $q \in V_T$,
$$\hat{\rho}(\beta_{\hat{T}_1}(\tau_1(q))) = \hat{\rho} \circ \rho_1(\beta_T(q)) = \rho_2(\beta_T(q)) = \beta_{\hat{T}_2}(\tau_2(q)) = \beta_{\hat{T}_2}(\hat{\tau} \circ \tau_1(q)).$$
   Moreover, for any $\{q_1, q_2\} \in E_T$,
$$\hat{\rho}(\gamma_{\hat{T}_1}(\tau_1(q_1, q_2))) = \hat{\rho} \circ \rho_1(\gamma_T(q_1, q_2)) = \rho_2(\gamma_T(q_1, q_2)) = \gamma_{\hat{T}_2}(\tau_2(q_1, q_2)) = \gamma_{\hat{T}_2}(\hat{\tau} \circ \tau_1(q_1, q_2)).$$
   Since $\tau_1$ is surjective, $\tau_1(q)$ ranges over all nodes in $\hat{T}_1^{t_1}$ when $q$ ranges over $V_T$, and $\tau_1(q_1, q_2)$ ranges over all edges in $\hat{T}_1^{t_1}$ when $(q_1, q_2)$ ranges over $E_T$. We thus conclude that $(\rho, \tau)$ is an isomorphism from $(\hat{F}_1, \hat{T}_1^{t_1})$ to $(\hat{F}_2, \hat{T}_2^{t_2})$

3. We finally prove that $\phi_1 = \phi_2 \circ \hat{\rho}$ and $\psi_1 = \psi_2 \circ \hat{\tau}$. Pick any $u \in V_F$, we have $\phi_2 \circ \hat{\rho}\rho_1(u) = \phi_2 \circ \rho_2(u) = \phi_1 \circ \rho_1(u)$. Since $\rho_1$ is surjective, $\phi_1(u)$ ranges over all vertices in $\hat{F}_1$ when $u$ ranges over $V_F$. This proves that $\phi_1 = \phi_2 \circ \hat{\rho}$. Following the same procedure, we can prove that $\psi_1 = \psi_2 \circ \hat{\tau}$.

Combining the above three items concludes the proof. $\qquad\square$

From Lemma B.17, we can also obtain the finite-iteration version of Lemma B.17 as follows:

**Lemma B.18.** *For any parallel-tree decomposed graph $(F, T^r) \in \mathcal{S}_d^{pt}$ and $\left(\tilde{F}, \tilde{T}^s\right) \in \mathcal{S}_d^{pt}$, we have*

$$
\mathsf{strhom}\left((F, T^r), \left(\tilde{F}, \tilde{T}^s\right)\right) = \sum_{(\hat{F}, \hat{T}^t) \in \mathcal{S}_d^{\mathsf{pt}}} \frac{\mathsf{strsurj}\left((F, T^r), \left(\hat{F}, \hat{T}^t\right)\right) \cdot \mathsf{strinj}\left(\left(\hat{F}, \hat{T}^t\right), \left(\tilde{F}, \tilde{T}^s\right)\right)}{\mathsf{aut}\left(\hat{F}, \hat{T}^t\right)},
$$

*Proof.* According to the third condition in Definition B.13, for $(\hat{F}, \hat{T}^r) \in \mathcal{S}^{pt}$, if $\mathsf{strsurj}((F, T^r), (\hat{F}, \hat{T}^t)) \cdot \mathsf{strinj}((\hat{F}, \hat{T}^t), (\tilde{F}, \tilde{T}^s)) \neq 0$, it follows that $(\hat{F}, \hat{T}^r) \in \mathcal{S}_d^{pt}$. Therefore, the conclusion of the lemma is immediate. $\qquad\square$

**Definition B.19.** We can list all non-isomorhpic parallel-tree decomposed graphs into an infinite sequence $(F_1, T_1^{r_1}), (F_2, T_2^{r_2}), \ldots$ with the following order.

- The order requires $|V_{T_i}| \leq |V_{T^j}|$ for any $i < j$.
- If $|V_{T_i}| = |V_{T^j}|$ for any $i < j$, then $|F_{T_i}| \leq |F_{T_j}|$.

Then we define following function matrix and function vector based on the order defined above.

1. Let $f : \mathcal{S}^{\mathsf{pt}} \times \mathcal{S}^{\mathsf{pt}} \to \mathbb{N}$ be any mapping. Define the associated matrix $\boldsymbol{M}^f \in \mathbb{N}^{\mathbb{N}_+ \times \mathbb{N}_+}$, where $A_{i,j}^f = f\left((F_i, T_i^{r_i}), (F_j, T_j^{r_j})\right)$. Similarly, we consider the finite-iteration version. Let $f : \mathcal{S}_d^{\mathsf{pt}} \times \mathcal{S}_d^{\mathsf{pt}} \to \mathbb{N}$ be any mapping. Define the associated matrix $\boldsymbol{M}^f \in \mathbb{N}^{\mathbb{N}_+ \times \mathbb{N}_+}$, where $M_{i,j}^{f,(d)} = f\left((F_i, T_i^{r_i}), (F_j, T_j^{r_j})\right)$.

2. Let $g : \mathcal{S}^{\mathsf{pt}} \times \mathcal{G} \to \mathbb{N}$ be any mapping. Given a graph $G \in \mathcal{G}$, define the (infinite) vector $\boldsymbol{l}_G^g \in \mathbb{N}^{\mathbb{N}_+}$, where $l_{G,i}^g = g\left((F_i, T_i^{r_i}), G\right)$. For the finite-iteration version, let $g : \mathcal{S}_d^{\mathsf{pt}} \times \mathcal{G} \to \mathbb{N}$ be any mapping. Given a graph $G \in \mathcal{G}$, define the (infinite) vector $\boldsymbol{l}_G^{g,(d)} \in \mathbb{N}^{\mathbb{N}_+}$, where $l_{G,i}^{g,(d)} = g\left((F_i, T_i^{r_i}), G\right)$.

3. Let $h : \mathcal{G} \times \mathcal{G} \to \mathbb{N}$ be any mapping. Given a graph $G \in \mathcal{G}$, define the (infinite) vector $\boldsymbol{l}_G^h \in \mathbb{N}^{\mathbb{N}_+}$, where $l_{G,i}^h = h(F_i, G)$. In the finite-iteration setting, let $h : \mathcal{G} \times \mathcal{G} \to \mathbb{N}$ be any mapping. Given a graph $G \in \mathcal{G}$, define the (infinite) vector $\boldsymbol{l}_G^{h,(d)} \in \mathbb{N}^{\mathbb{N}_+}$, where $l_{G,i}^{h,(d)} = h(F_i, G)$.

**Theorem B.20.** *For any two graphs $G$ and $H$, we have $\mathsf{hom}((F, T^r), G) = \mathsf{hom}((F, T^r), H)$ for all parallel-tree decomposed graphs $(F, T^r)$ iff $\mathsf{treeCount}((F, T^r), G) = \mathsf{treeCount}((F, T^r), H)$ for all parallel-tree decomposed graphs. Similarly, in the finite-iteration setting, $\mathsf{hom}((F, T^r), G) = \mathsf{hom}((F, T^r), H)$ holds for all $(F, T^r) \in \mathcal{S}_d^{pt}$ iff $\mathsf{treeCount}((F, T^r), G) = \mathsf{treeCount}((F, T^r), H)$ for all $(F, T^r) \in \mathcal{S}_d^{pt}$.*

*Proof.* We consider each direction separately.

1. First, we prove that if $\mathsf{treeCount}((F, T^r), G) = \mathsf{treeCount}((F, T^r), H)$ for all parallel-tree decomposed graphs, then $\mathsf{hom}((F, T^r), G) = \mathsf{hom}((F, T^r), H)$ for all such graphs $(F, T^r)$. According to Theorem B.14, this result can be expressed in matrix form as $\boldsymbol{l}_G^{\mathsf{hom}} = \boldsymbol{M}^{\mathsf{strhom}} \cdot \boldsymbol{l}_G^{\mathsf{treeCount}}$ and $\boldsymbol{l}_H^{\mathsf{hom}} = \boldsymbol{M}^{\mathsf{strhom}} \cdot \boldsymbol{l}_H^{\mathsf{treeCount}}$ for all parallel trees $F$. This directly implies that $\boldsymbol{l}_G^{\mathsf{treeCount}} = \boldsymbol{l}_H^{\mathsf{treeCount}}$ leads to $\boldsymbol{l}_G^{\mathsf{hom}} = \boldsymbol{l}_H^{\mathsf{hom}}$. Similarly, in the finite-iteration setting, the result from Theorem B.15 can be rewritten in matrix form as $\boldsymbol{l}_G^{\mathsf{hom},(d)} = \boldsymbol{M}^{\mathsf{strhom},(d)} \cdot \boldsymbol{l}_G^{\mathsf{treeCount},(d)}$. Therefore, if $\boldsymbol{l}_G^{\mathsf{treeCount},(d)} = \boldsymbol{l}_H^{\mathsf{treeCount},(d)}$, it follows that $\boldsymbol{l}_G^{\mathsf{hom},(d)} = \boldsymbol{l}_H^{\mathsf{hom},(d)}$.

2. For the second direction of the lemma, it suffices to prove the finite-iteration setting, as the general case directly follows. According to Lemma B.18, we have the following equations:

$$l_G^{\mathsf{strhom},(d)} = M^{\mathsf{strsurj},(d)} \cdot M^{\mathsf{strinj},(d)} \cdot (M^{\mathsf{aut},(d)})^{-1} \cdot l_G^{\mathsf{treeCount},(d)},$$

$$l_H^{\mathsf{strhom},(d)} = M^{\mathsf{strsurj},(d)} \cdot M^{\mathsf{strinj},(d)} \cdot (M^{\mathsf{aut},(d)})^{-1} \cdot l_H^{\mathsf{treeCount},(d)}.$$

By simple observation, $M^{\mathsf{aut}}$ is a diagonal matrix where all diagonal elements are positive integers. Moreover, $M^{\mathsf{strinj}}$ is an upper triangular matrix with positive diagonal elements. This holds because $\mathsf{strinj}((F_i, T_i^{r_i}), (F_j, T_j^{r_j})) > 0$ only when $|V_{T_i}| \leq |V_{T_j}|$. Since $M^{\mathsf{strsurj},(d)}$ is a lower triangular matrix with positive diagonal elements, it is invertible. Thus,

$$M^{\mathsf{strinj},(d)} \cdot l_G^{\mathsf{treeCount},(d)} = M^{\mathsf{strinj},(d)} \cdot l_H^{\mathsf{treeCount},(d)}.$$

Additionally, by the definition of an unfolding tree, there are only finitely many non-zero elements in both $l_G^{\mathsf{treeCount},(d)}$ and $l_H^{\mathsf{treeCount},(d)}$, and the corresponding non-zero indices are restricted to a fixed (finite) set. In this case, the upper triangular matrix $M^{\mathsf{strinj},(d)}$ reduces to a finite-dimensional matrix, so we conclude that $l_G^{\mathsf{treeCount},(d)} = l_H^{\mathsf{treeCount},(d)}$. By enumerating over all $d \geq 0$, we obtain that $l_G^{\mathsf{treeCount}} = l_H^{\mathsf{treeCount}}$.

Combining item 1 and item 2, we finish the proof of the lemma. $\square$

## B.4 Step 3: Finding Pebble Game for Spectral Invariant GNN

In this section, we introduce the pebble game and demonstrate its equivalence to the expressive power of spectral invariant GNN.

### B.4.1 Pebble Game

We first formally define the rules of pebble game.

**Definition B.21 (Pebble game for spectral invariant GNN).** The pebbling game is conducted on two graphs $G = (V_G, E_G)$ and $H = (V_H, E_H)$. Initially, each graph is equipped with two distinct pebbles, denoted as $u$ and $v$, which start off outside the graphs. The game involves two players: the *Spoiler* and the *duplicator*. We now describe the procedure of the game as follows:

- *Initialization:* The Spoiler first selects a non-empty subset $V^S$ from either $V_G$ or $V_H$, and the duplicator responds with a subset $V^D$ from the other graph, ensuring that $|V^D| = |V^S|$. The duplicator loses the game if no feasible choice is available. The Spoiler places a pebble $u$ on a vertex in $V^D$, and the duplicator places a corresponding pebble $u$ in $V^S$. Similarly, the Spoiler and duplicator repeat the process to place two pebbles, $v$. Specifically, the Spoiler selects a non-empty subset $V^S$ from either $V_G$ or $V_H$, and the duplicator responds by selecting a subset $V^D$ from the other graph, maintaining $|V^S| = |V^D|$. The Spoiler then places $v$ on a vertex in $V^D$, while the duplicator places the corresponding $v$ in $V^S$.

- *Main Process:* The game iteratively repeats the following steps, where, in each iteration, the Spoiler may choose freely between the following two actions:

  1. Action 1 (moving pebble $v$): The Spoiler first selects a non-empty subset $V^S$ from either $V_G$ or $V_H$, and the duplicator responds with a subset $V^D$ from the other graph, ensuring that $|V^D| = |V^S|$. The Spoiler then moves pebble $v$ to a vertex in $V^D$, and the duplicator moves the corresponding pebble $v$ to a vertex in $V^S$.
  2. Action 2 (moving pebble $u$): The Spoiler first selects a non-empty subset $V^S$ from either $V_G$ or $V_H$, and the duplicator responds with a subset $V^D$ from the other graph, ensuring that $|V^D| = |V^S|$. The Spoiler then moves pebble $u$ to a vertex in $V^D$, and the duplicator moves the corresponding pebble $u$ to a vertex in $V^S$.

- *Termination:* The Spoiler wins if, after a certain number of rounds, $\omega_G^{\star}(u, v)$ for graph $G$ differs from $\omega_H^{\star}(u, v)$ for graph $H$. Conversely, the duplicator wins if the Spoiler is unable to achieve a win after any number of rounds.

### B.4.2 EQUIVALENCE BETWEEN SPECTRAL GNNS AND PEBBLING GAMES

**Lemma B.22.** *Let $l \in \mathbb{N}$ be any integer. For any vertices $u_G, v_G \in \mathcal{V}_G$ and $u_H, v_H \in \mathcal{V}_H$, if $\chi_G^{\mathsf{Walk},(l)}(u) \neq \chi_H^{\mathsf{Walk},(l)}(v)$, then the Spoiler can win the game in $l - 1$ rounds when the two pebbles $u$ are initially placed on vertices $u_G \in V_G$ and $u_H \in V_H$ in graphs $G$ and $H$, respectively.*

*Proof.* The proof proceeds by induction on $l$. First, consider the base case where $l = 0$. In this case, the statement is trivially true.

Now, assume that the lemma holds for all $l \leq L$, and consider the case where $l = L + 1$. Suppose $\chi_G^{\mathsf{Walk},(L+1)}(u_G) \neq \chi_H^{\mathsf{Walk},(L+1)}(u_H)$. If $\chi_G^{\mathsf{Walk},(L)}(u_G) \neq \chi_H^{\mathsf{Walk},(L)}(u_H)$, then by the inductive hypothesis, Spoiler wins. Otherwise, we have

$$\{\!\{(\omega_G^\star(u_G, v_G), \chi_G^{\mathsf{Walk},(L)}(v_G)) : v_G \in \mathcal{V}_G\}\!\} \neq \{\!\{(\omega_H^\star(u_H, v_H), \chi_H^{\mathsf{Walk},(L)}(v_H)) : v_H \in \mathcal{V}_H\}\!\}.$$

Therefore, there exists a color $c$ and $x \in \mathbb{R}^{|V_G|}$ such that $|\mathcal{C}_G(u_G, c, x)| \neq |\mathcal{C}_H(u_H, c, x)|$, where

$$\mathcal{C}_G(u_G, c, x) = \left\{ v_G \in \mathcal{V}_G : \chi_G^{\mathsf{Walk},(L)}(v_G) = c, \omega_G^\star(u_G, v_G) = x \right\}.$$

If $|\mathcal{C}_G(u_G, c, x)| > |\mathcal{C}_H(u_H, c, x)|$, the Spoiler can select the vertex subset $V^S = \mathcal{C}_G(u_G, c, x) \subset \mathcal{V}_G$. Regardless of how the Duplicator responds with a subset $V^D \subset \mathcal{V}_H$, there exists a vertex $v_H \in V^D$ such that $(\omega_H^\star(u_H, v_H), \chi_H^{\mathsf{Walk},(L)}(v_H)) \neq (x, c)$. The Spoiler then selects this vertex $x^S = v_H$, and no matter how the Duplicator responds with $x^D = v_G \in V^S$, we have either $\omega_G^\star(u_G, v_G) \neq \omega_H^\star(u_H, v_H)$ or $\chi_G^{\mathsf{Walk},(L)}(v_G) \neq \chi_H^{\mathsf{Walk},(L)}(v_H)$. If $\omega_G^\star(u_G, v_G) \neq \omega_H^\star(u_H, v_H)$, the Spoiler wins the game immediately. If $\chi_G^{\mathsf{Walk},(L)}(v_G) \neq \chi_H^{\mathsf{Walk},(L)}(v_H)$, the remainder of the game is equivalent to one where the two pebbles $u$ are initially placed on $v_G \in V_G$ and $v_H \in V_H$ in graphs $G$ and $H$ respectively. By the inductive hypothesis, the Spoiler wins the game.

If $|\mathcal{C}_G(u_G, c, x)| < |\mathcal{C}_H(u_H, c, x)|$, Spoiler can select the vertex subset $V^S = \mathcal{C}_H(u_H, c, x) \subset \mathcal{V}_H$, and the conclusion follows analogously. □

**Lemma B.23.** *For any vertices $u_G \in \mathcal{V}_G$ and $u_H \in \mathcal{V}_H$, if $\chi_G^{\mathsf{Walk},(l+1)}(u_G) = \chi_H^{\mathsf{Walk},(l+1)}(u_H)$, then the Spoiler cannot win the game within $l$ rounds when the two pebbles are initially placed on vertices $u_G \in V_G$ and $u_H \in V_H$ in graphs $G$ and $H$, respectively.*

*Proof.* The proof proceeds by induction on $l$. The base case $l = 0$ is trivially true. Now, assume the statement holds for $l \leq L$, and consider the case $l = L + 1$. Suppose $\chi_G^{\mathsf{Walk},(L+2)}(u_G) = \chi_H^{\mathsf{Walk},(L+2)}(u_H)$. Then,

$$\{\!\{\left(\omega_G^\star(u_G, v_G), \chi_G^{\mathsf{Walk},(L+1)}(v_G)\right) : v_G \in \mathcal{V}_G\}\!\} = \{\!\{\left(\omega_H^\star(u_H, v_H), \chi_H^{\mathsf{Walk},(L+1)}(v_H)\right) : v_H \in \mathcal{V}_H\}\!\}.$$

If Spoiler selects a subset $V^S$, and if $V^S \subset \mathcal{V}_G$, Duplicator can respond with a subset $V^D \subset \mathcal{V}_H$ such that

$$\{\!\{\left(\omega_G^\star(u_G, v_G), \chi_G^{\mathsf{Walk},(L+1)}(v_G)\right) : v_G \in V^S\}\!\} = \{\!\{\left(\omega_H^\star(u_H, v_H), \chi_H^{\mathsf{Walk},(L+1)}(v_H)\right) : v_H \in V^D\}\!\}.$$

Similarly, if $V^S \subset \mathcal{V}_H$, Duplicator can respond with a subset $V^D \subset \mathcal{V}_G$ such that

$$\{\!\{\left(\omega_G^\star(u_G, v_G), \chi_G^{\mathsf{Walk},(L+1)}(v_G)\right) : v_G \in V^D\}\!\} = \{\!\{\left(\omega_H^\star(u_H, v_H), \chi_H^{\mathsf{Walk},(L+1)}(v_H)\right) : v_H \in V^S\}\!\}.$$

In both cases, it is clear that $|V^S| = |V^D|$. Next, regardless of how Spoiler moves the pebble $v$ to a vertex $x^S \in V^D$, Duplicator can always respond by moving the corresponding pebble $v$ to a vertex $x^D \in V^S$, such that

$$\left(\omega_G^\star(u_G, \tilde{v}_G), \chi_G^{\mathsf{Walk},(L+1)}(\tilde{v}_G)\right) = \left(\omega_H^\star(u_H, \tilde{v}_H), \chi_H^{\mathsf{Walk},(L+1)}(\tilde{v}_H)\right),$$

where $(\tilde{v}_G, \tilde{v}_H)$ represents the new positions of the pebbles. The remaining game is then equivalent to a game in which the two pebbles are initially placed on vertices $\tilde{v}_G \in V_G$ and $\tilde{v}_H \in V_H$ in graphs $G$ and $H$, respectively.

□

Combining previous two lemmas, we have the following result:

**Lemma B.24.** *Given graph $G$ and $H$, Spoiler cannot wins the pebble game in $d$ steps iff* $\chi_G^{\mathsf{Spec},(d)}(G) = \chi_H^{\mathsf{Spec},(d)}(H)$.

Therefore, we have proven Lemma 3.19 in the main paper.

## B.5  Step 4: Introducing Fürer graphs

To continue, we draw introduce Fürer graphs, and we further prove that pebble games restricted on Fürer graphs can be greatly simplified.

### B.5.1  Properties of Fürer graphs

We first introduce the definition of Fürer graphs, introduced by Fürer (2001).

**Definition B.25 (Connected components).** Let $F = (V_F, E_F)$ be a connected graph, and let $U \subset V_F$ be a set of vertices, referred to as separation vertices. We define two edges $\{u, v\}, \{x, y\} \in E_F$ as belonging to the same connected component if there exists a simple path $\{\{y_0, y_1\}, \{y_1, y_2\}, \ldots, \{y_{k-1}, y_k\}\}$ such that $\{y_0, y_1\} = \{u, v\}$, $\{y_{k-1}, y_k\} = \{x, y\}$, and $y_i \notin U$ for all $i \in [1, k-1]$. It is straightforward to verify that this relation between edges induces an *equivalence relation*. Consequently, the edge set $E_F$ can be partitioned into disjoint subsets, denoted by $\mathsf{CC}_F(U) = \{P_i : i \in [m]\}$, where each $P_i \subset E_F$ represents a connected component for some $m$.

**Definition B.26 (Fürer graphs).** Given any connected graph $F = (V_F, E_F)$, the Fürer graph $G(F) = (V_{G(F)}, E_{G(F)})$ is constructed as follows:

$$V_{G(F)} = \{(x, X) : x \in V_F, X \subset N_F(x), |X| \bmod 2 = 0\},$$
$$E_{G(F)} = \{\{(x, X), (y, Y)\} \subset V_G : \{x, y\} \in E_F, (x \in Y \leftrightarrow y \in X)\}.$$

Here, $x \in Y \leftrightarrow y \in X$ holds when either ($x \in Y$ and $y \in X$) or ($x \notin Y$ and $y \notin X$) holds. For each $x \in V_F$, denote the set

$$\mathsf{Meta}_F(x) := \{(x, X) : X \subset N_F(x), |X| \bmod 2 = 0\}, \tag{4}$$

which is called the meta vertices of $G(F)$ associated to $x$. Note that $V_{G(F)} = \bigcup_{x \in V_F} \mathsf{Meta}_F(x)$.

We next define an operation called "twist":

**Definition B.27 (Twist).** Let $G(F) = (V_{G(F)}, E_{G(F)})$ be the Fürer graph of $F = (V_F, E_F, \ell_F)$, and let $\{x, y\} \in E_F$ be an edge of $F$. The *twisted* Fürer graph of $G(F)$ for edge $\{x, y\}$, is constructed as follows: $\mathsf{twist}(G(F), \{x, y\}) := (V_{G(F)}, E_{\mathsf{twist}(G(F), \{x, y\})})$, where

$$E_{\mathsf{twist}(G(F), \{x, y\})} := E_{G(F)} \triangle \{\{\xi, \eta\} : \xi \in \mathsf{Meta}_F(x), \eta \in \mathsf{Meta}_F(y)\},$$

and $\triangle$ is the symmetric difference operator, i.e., $A \triangle B = (A \backslash B) \cup (B \backslash A)$. For an edge set $S = \{e_1, \cdots, e_k\} \subset E_F$, we further define

$$\mathsf{twist}(G(F), S) := \mathsf{twist}(\cdots \mathsf{twist}(G(F), e_1) \cdots, e_k). \tag{5}$$

Note that Equation (5) is well-defined as the resulting graph does not depend on the order of edges $e_1, \cdots, e_k$ for twisting.

The following result is well-known (see e.g., Zhang et al., 2023a, Corollary I.5 and Lemma I.7)):

**Theorem B.28.** *For any connected graph $F$ and any set $S_1, S_2 \subset E_F$, $\mathsf{twist}(G(F), S_1) \simeq \mathsf{twist}(G(F), S_2)$ iff $|S_1| \equiv |S_2| \pmod 2$.*

We now present an essential property of Fürer graphs in terms of walk number:

**Theorem B.29.** *Let $G(F) = (V_G, E_G)$ be the Fürer graph of $F = (V_F, E_F)$, and let $H(F) = \mathsf{twist}(G(F), \mathcal{E})$ for some $\mathcal{E} \subset E_F$. Given $(x, \mathcal{X}), (y, \mathcal{Y}) \in V_G$, and a connected component $P \in \mathsf{CC}_F(\{x, y\})$ with $|P \cap \mathcal{E}| = 1$, the number of $n$-walks from $(x, \mathcal{X})$ to $(y, \mathcal{Y})$, passing through $\mathsf{Meta}(x_1), \mathsf{Meta}(x_2), \ldots, \mathsf{Meta}(x_n)$ sequentially in $G(F)$, is equal to the number of such walks in $H(F)$ for all $n \in \mathbb{N}^+$ and vertices $x_1, \ldots, x_n$ on $P$, iff $P$ is a path.*

*Proof.* If $P$ is a path, we denote $P = \{\{x_1, x_2\}, \ldots, \{x_{n-1}, x_n\}\}$ with $x_1 = x$ and $x_n = y$. It follows that the number of $n$-walks starting from $(x, \mathcal{X})$ and ending at $(y, \mathcal{Y})$, passing through $\mathsf{Meta}(x_1), \ldots, \mathsf{Meta}(x_n)$ sequentially on $G(F)$, is not equal to the number of such walks on $H(F)$. Thus, one direction of the lemma is established.

If $P$ is not a path, then there exists at least one vertex, besides $x$ and $y$, on $P$ whose degree is greater than 2. We define $\omega_n^{G(F)}((x, \mathcal{X}), \mathsf{Meta}(x_2), \ldots, \mathsf{Meta}(x_{n-1}), (y, \mathcal{Y}))$ and $\omega_n^{H(F)}((x, \mathcal{X}), \mathsf{Meta}(x_2), \ldots, \mathsf{Meta}(x_{n-1}), (y, \mathcal{Y}))$ as the number of $n$-walks starting from $(x, \mathcal{X})$, ending at $(y, \mathcal{Y})$, and passing through $\mathsf{Meta}_F(x_1), \ldots, \mathsf{Meta}_F(x_n)$ sequentially in $G(F)$ and $H(F)$, respectively. We use the notation $\deg_F(v)$ to denote the degree of a vertex $v$ in the graph $F$. We proceed by induction on $n$ to prove the following stronger statement: If the degrees of $x_2, \ldots, x_{n-1}$ are not all less than or equal to 2, then there exists a function $f_n^F : V_F^n \to \mathbb{N}$ such that

$$\omega_n^{G(F)}((x, \mathcal{X}), \mathsf{Meta}(x_2), \ldots, \mathsf{Meta}(x_{n-1}), (y, \mathcal{Y}))$$
$$= \omega_n^{H(F)}((x, \mathcal{X}), \mathsf{Meta}(x_2), \ldots, \mathsf{Meta}(x_{n-1}), (y, \mathcal{Y})) = f_n(x_1, \ldots, x_n)$$

for all $n \in \mathbb{N}$, $(x, \mathcal{X}) \in \mathsf{Meta}(x)$, and $(y, \mathcal{Y}) \in \mathsf{Meta}(y)$.

We first consider the case when $n = 2$. In this case, we can straightforwardly define the function $f_n^F$ as $f(x_1, x_2, x_3) = 2^{\deg(x_2)-3}$.

Next, assume that the statement holds for $n \leq N$. We now consider the case when $n = N + 1$, and analyze two separate cases:

1. Not all degrees of $x_3, x_4, \ldots, x_{n-1}$ are less than or equal to 2.
   The $n$-walk passing $\mathsf{Meta}_F(x_1), \ldots, \mathsf{Meta}_F(x_n)$ sequentially can be decomposed into a 1-walk from $(x, \mathcal{X})$ to $\mathsf{Meta}_F(x_2)$, followed by an $n - 1$-walk passing $\mathsf{Meta}_F(x_2), \ldots, \mathsf{Meta}_F(x_n)$ sequentially and ending at $(y, \mathcal{Y})$. According to the induction hypothesis, the number of $(n - 1)$-walks passing $\mathsf{Meta}_F(x_2), \ldots, \mathsf{Meta}_F(x_n)$ sequentially and ending at $(y, \mathcal{Y})$ equals $f_{n-1}^F(x_2, x_3, \ldots, x_n)$. Since the number of 1-walks from $(x, \mathcal{X})$ to $\mathsf{Meta}_F(x_2)$ equals $2^{\deg_F(x_2)-2}$, we can define the function $f_n^F$ as $f_n^F(x_1, x_2, \ldots, x_n) = 2^{\deg_F(x_2)-2} \cdot f_{n-1}^F(x_2, x_3, \ldots, x_n)$.

2. All degrees of $x_3, x_4, \ldots, x_{n-1}$ are less than or equal to 2. In this case, we have $\deg_F(x_2) \geq 3$. The number of $(n - 1)$-walks passing $\mathsf{Meta}_F(x_2), \ldots, \mathsf{Meta}_F(x_n)$ sequentially and ending at $(y, \mathcal{Y})$ is either 1 or 0. Therefore, we can define the function $f_n^F$ as $f_n^F = 2^{\deg_F(x_2)-3}$.

Combining the two cases, we conclude that if the degrees of $x_2, x_3, \ldots, x_{n-1}$ are not all equal to 2, then there exists a function $f_n^F : V(F)^n \to \mathbb{N}$ such that

$$\omega_n^{G(F)}((x, \mathcal{X}), \mathsf{Meta}(x_2), \ldots, \mathsf{Meta}(x_{n-1}), (y, \mathcal{Y}))$$
$$= \omega_n^{H(F)}((x, \mathcal{X}), \mathsf{Meta}(x_2), \ldots, \mathsf{Meta}(x_{n-1}), (y, \mathcal{Y})) = f_n(x_1, \ldots, x_n)$$

for all $\mathcal{X} \in \mathcal{N}_F(x), \mathcal{Y} \in \mathcal{N}_F(y)$, and any $\mathcal{E} \subset \mathcal{E}_F$.

By combining all previous analyses, we have proven the result of the theorem.

$\square$

### B.5.2 SIMPLIFIED PEBBLE GAME ON FÜRER GRAPHS

**Definition B.30 (Simplified Pebble Game).** The simplified pebble game is defined as follows. Let $F = (V_F, E_F)$ represent the base graph of a proper Fürer graph. The game is played on $F$ with two pebbles, $u$ and $v$, each of a different type. Initially, both pebbles are placed outside the graph $F$. The game begins with Spoiler placing pebble $u$ on any vertex of $F$, while pebble $v$ remains outside the graph. The game then proceeds in cycles, following these steps: Spoiler places pebble $v$ on any vertex of $F$, swaps the positions of $u$ and $v$, and then places pebble $v$ back outside the graph. Duplicator, on the other hand, maintains a subset $\mathcal{Q}$ of connected components, where $\mathcal{Q} \subset \mathsf{CC}_\mathcal{S}(F)$ and $\mathcal{S}$ is the set of vertices in $F$ where pebbles $u$ and $v$ are currently placed.

When Spoiler places a pebble on a vertex of $F$, one of two scenarios occurs. If $\mathsf{CC}_\mathcal{S}(F)$ remains unchanged, Duplicator takes no action. However, if the new pebble placement causes a connected

component to split into smaller regions, Duplicator updates $\mathcal{Q}$ by replacing any original component $\mathcal{P} \subset E_F$ that splits into $\mathcal{P}_1, \ldots, \mathcal{P}_k$ (where $\bigcup_{i=1}^{k} \mathcal{P}_i = \mathcal{P}$) with a subset of the newly formed components. That is, $\tilde{Q} = (\mathcal{Q} \setminus \mathcal{P}) \cup \{\mathcal{P}_{j_1}, \ldots, \mathcal{P}_{j_l}\}$ for some $j_1, \ldots, j_l \in [k]$, ensuring that $|\tilde{Q}| \equiv 1 \pmod 2$. In other words, Duplicator removes the old component $\mathcal{P}$ (if present) and adds some of the new components while preserving the parity of $|\mathcal{Q}|$. When Spoiler removes a pebble and places it outside the graph, two cases arise. If $\mathsf{CC}_{\mathcal{S}}(F)$ remains unchanged, Duplicator again takes no action. However, if the removal of the pebble causes multiple connected components $\mathcal{P}_1, \ldots, \mathcal{P}_k$ to merge into a larger component $\mathcal{P} = \bigcup_{i=1}^{k} \mathcal{P}_i$, Duplicator updates $\mathcal{Q}$ by either removing the smaller components, i.e., $\tilde{Q} = \mathcal{Q} \setminus \{\mathcal{P}_1, \ldots, \mathcal{P}_k\}$, or adding the merged component, i.e., $\tilde{Q} = (\mathcal{Q} \setminus \{\mathcal{P}_1, \ldots, \mathcal{P}_k\}) \cup \mathcal{P}$, depending on which option preserves $|\tilde{Q}| \equiv 1 \pmod 2$. When Spoiler swaps the positions of the two pebbles, the connected components $\mathsf{CC}_{\mathcal{S}}(F)$ do not change, so Duplicator does not modify $\mathcal{Q}$.

Spoiler wins the game if, after any round, $\mathcal{Q}$ contains a connected component that forms a path. Duplicator wins if Spoiler is unable to achieve this outcome after any number of rounds.

**Lemma B.31.** *Given a base graph F, Spoiler cannot win the simplified pebble game on F in d steps iff* $\chi_G^{\mathsf{Spec},(d+1)}(G(F)) = \chi_H^{\mathsf{Spec},(d+1)}(H(F))$.

The proof of Lemma B.31 follows a similar structure to the proof of Theorem 17 in Zhang et al. (2023a), and thus we omit the details here for the sake of simplicity. Notably, the main idea behind the proof is to show that the original pebble game is equivalent to the following 'half-simplified' version of the game:

Let $F = (V_F, E_F)$ be the base graph of a proper Fürer graph. This version of the pebble game is also played on $F$, with two pebbles $u$ and $v$. Initially, both pebbles are outside the graph $F$.

- First, we describe the rules for the Spoiler. Spoiler maintains a subset $\mathcal{Q}_1 \subset \mathsf{CC}_{\mathcal{S}}(F)$ of connected components, where the set $\mathcal{S}$ consists of the vertices in $F$ currently occupied by the pebbles $u$ and $v$. (If pebble $v$ is outside $F$, then $\mathcal{S}$ contains only the vertex where $u$ is placed.) Initially, Spoiler places $u$ on any vertex of $F$ and leaves $v$ outside the graph, maintaining $\mathcal{Q}_1 = \{E_F\}$. Then, the game proceeds cyclically as follows:
  - Spoiler places $v$ on any vertex of $F$. Two cases arise for maintaining $\mathcal{Q}_1$: if $\mathsf{CC}_{\mathcal{S}}(F)$ does not change, Spoiler leaves $\mathcal{Q}_1$ unchanged. Otherwise, the new pebble may split some connected components into smaller regions. For each original component $\mathcal{P} \subset \mathcal{E}_F$ that splits into $\mathcal{P}_1, \ldots, \mathcal{P}_k$ with $\bigcup_{i=1}^{k} \mathcal{P}_i = \mathcal{P}$, Spoiler updates $\mathcal{Q}_1$ to $\tilde{\mathcal{Q}}_1 = (\mathcal{Q}_1 \setminus \mathcal{P}) \cup \{\mathcal{P}_{j_1}, \ldots, \mathcal{P}_{j_l}\}$, where $j_1, \ldots, j_l \in [k]$ and $|\tilde{\mathcal{Q}}_1| \equiv 0 \pmod 2$. This ensures that the parity of $|\mathcal{Q}_1|$ remains unchanged.
  - Spoiler swaps the positions of $u$ and $v$, leaving $\mathcal{Q}_1$ unchanged.
  - Spoiler removes $v$ from the graph, leaving it outside $F$. Again, two cases arise for maintaining $\mathcal{Q}_1$: if $\mathsf{CC}_{\mathcal{S}}(F)$ does not change, Spoiler does nothing. Otherwise, several connected components $\mathcal{P}_1, \ldots, \mathcal{P}_k$ may merge into a larger component $\mathcal{P} = \bigcup_{i=1}^{k} \mathcal{P}_i$. Spoiler then updates $\mathcal{Q}_1$ to either $\tilde{\mathcal{Q}}_1 = \mathcal{Q}_1 \setminus \{\mathcal{P}_1, \ldots, \mathcal{P}_k\}$ or $\tilde{\mathcal{Q}}_1 = (\mathcal{Q}_1 \setminus \{\mathcal{P}_1, \ldots, \mathcal{P}_k\}) \cup \mathcal{P}$, whichever satisfies $|\tilde{\mathcal{Q}}_1| \equiv 0 \pmod 2$.
- Next, we describe the rules for the Duplicator, which are analogous to the Spoiler's rules but with a key difference: Duplicator maintains a subset $\mathcal{Q}_2 \subset \mathsf{CC}_{\mathcal{S}}(F)$ where the parity of $|\mathcal{Q}_2|$ is always odd. Initially, $\mathcal{Q}_2 = \{E_F\}$, and throughout the game, Duplicator performs the following updates:
  - When Spoiler places a pebble, Duplicator updates $\mathcal{Q}_2$ in the same manner as $\mathcal{Q}_1$, but ensuring $|\tilde{\mathcal{Q}}_2| \equiv 1 \pmod 2$.
  - When Spoiler removes a pebble, Duplicator updates $\mathcal{Q}_2$ as in the previous case, ensuring that the parity of $|\mathcal{Q}_2|$ remains odd.
  - When Spoiler swaps the pebbles, Duplicator does nothing, as $\mathsf{CC}_{\mathcal{S}}(F)$ remains unchanged.

The result of the game is determined as follows: Suppose that pebbles $u$ and $v$ are placed on vertices of $F$. Spoiler maintains the subset $\mathcal{Q}_1$, and Duplicator maintains $\mathcal{Q}_2$. We then construct two twisted Fürer graphs: $\tilde{G}(F) = \mathsf{twist}(G(F), \tilde{\mathcal{E}})$ and $\hat{G}(F) = \mathsf{twist}(G(F), \hat{\mathcal{E}})$, where $|\tilde{\mathcal{E}}| = |\mathcal{Q}_1|$ and, for each $\mathcal{P} \in \mathcal{Q}_1$, we select a single edge $\tilde{\mathcal{E}} \cap \mathcal{P} = 1$. Similarly, $|\hat{\mathcal{E}}| = |\mathcal{Q}_2|$, and for each $\mathcal{P} \in \mathcal{Q}_2$,

we select a single edge $\hat{\mathcal{E}} \cap \mathcal{P} = 1$. Spoiler wins if the walk vector satisfies $\omega^{\star}_{\tilde{G}(F)}((u, \emptyset), (v, \emptyset)) \neq \omega^{\star}_{\hat{G}(F)}((u, \emptyset), (v, \emptyset))$, meaning that there exists an $n$-walk in $\tilde{G}(F)$ from $(u, \emptyset)$ to $(v, \emptyset)$ that differs from the corresponding $n$-walk in $\hat{G}(F)$. By following a similar analysis to that in Theorem 17 of Zhang et al. (2023a), we can demonstrate that the 'half-simplified' pebble game is equivalent to the original pebble game. Specifically, the Spoiler can win in $d$ steps in the original pebble game if and only if they can win in $d$ steps in the half-simplified pebble game. Furthermore, since it is clear that the 'half-simplified' game is equivalent to the simplified game, we can conclude that the original game and the simplified game are equivalent on Fürer graphs.

### B.6   Step 5: Proving the Maximality of Homomorphism Expressivity

Before presenting the proof, we redefine the concept of the game state graph for clarity in the technical exposition. Notably, there is a slight difference between the definition of the game state graph here and the one in the previous section: we only consider game states with a single pebble in the game state graph.

**Definition B.32.** We define the game state $(u, \mathcal{Q})$ as in the previous section, where $u \in V_F$ represents the position of the pebble, and $\mathcal{Q}$ is the connected component maintained by the Duplicator. The game state graph is formed by all game states $(u, \mathcal{Q})$. There is an edge from $(u, \mathcal{Q})$ to $(\tilde{u}, \tilde{\mathcal{Q}})$ if there exists a game transition from $(u, \mathcal{Q})$ to $((\hat{u}, \hat{v}), \hat{\mathcal{Q}})$, followed by a transition from $((\hat{u}, \hat{v}), \hat{\mathcal{Q}})$ to $(\tilde{u}, \tilde{\mathcal{Q}})$, for some connected component set $\hat{\mathcal{Q}} \subset E_F$ and vertex $\hat{v} \in V_F$.

**Definition B.33.** A game state $(u, \mathcal{Q})$ is called a terminal game state if there is a transition from $(u, \mathcal{Q})$ to a game state $((u, \tilde{v}), \tilde{\mathcal{Q}})$ for some connected component set $\tilde{\mathcal{Q}} \subset E_F$ and vertex $\tilde{v} \in V_F$, such that $\tilde{\mathcal{Q}}$ consists only of a single path. In this case, the game state $(u, \mathcal{Q})$ is called a terminal game state. It is straightforward to see that the Spoiler can win in the terminal state.

**Definition B.34.** Given a game state graph $G^{\mathsf{S}}$, a state $(u, \mathcal{Q})$ is termed "contracted" if, for any transition $(u, \mathcal{Q}) \to (u', \mathcal{Q}') \in E_{G^{\mathsf{s}}}$, it holds that $\mathcal{Q}' \subset \mathcal{Q}$. The state is called "strictly contracted" if, for any transition $(u, \mathcal{Q}) \to (u', \mathcal{Q}') \in E_{G^{\mathsf{s}}}$, it holds that $\mathcal{Q}' \subsetneq \mathcal{Q}$.

**Definition B.35.** A game state $(u, \mathcal{Q})$ is defined as "unreachable" if any path starting from the initial state $(\emptyset, E_F)$ and ending at $(u, \mathcal{Q})$ passes through a terminal state.

We do not need to consider unreachable states since the Spoiler always wins before reaching them.

**Lemma B.36.** *For any graph $F$, if the Spoiler can win the pebble game on $F$, then there exists a game state graph $G^{\mathsf{S}}$ corresponding to a winning strategy for the Spoiler such that all reachable and non-terminal states are strictly contracted.*

*Proof.*  1. First, we prove that there exists a strategy for the Spoiler such that every reachable and non-terminal state is contracted. Since the Spoiler can win the pebble game, they can win at any reachable state $(u, \mathcal{Q})$. Consider any strategy where $(u, \mathcal{Q})$ is not contracted. Note that the game state graph induced by all reachable states is a Directed Acyclic Graph (DAG), so we can choose a state $(u, \mathcal{Q})$ such that no path from the initial state $(\emptyset, \{E_F\})$ to $(u, \mathcal{Q})$ passes through any intermediate state that is not contracted. Next, we construct a new strategy to make the state $(u, \mathcal{Q})$ unreachable. We clearly have $u \neq \emptyset$. Without loss of generality, assume there is a transition $(u, \mathcal{Q}) \to (u', \mathcal{Q}')$ such that $\mathcal{Q}' \not\subset \mathcal{Q}$. Let $(u_0, \mathcal{Q}_0), (u_1, \mathcal{Q}_1), \ldots, (u_T, \mathcal{Q}_T)$ be any path from the initial state $(\emptyset, \{E_F\})$ to $(u, \mathcal{Q})$. We modify the strategy as follows: at state $(u_{T-1}, \mathcal{Q}_{T-1})$, the Spoiler places the pebble $\mathsf{p}_v$ on $u'$, swaps the pebbles at $u$ and $v$, and then removes $\mathsf{p}_v$ from the graph. This process can be repeated for every path from the initial state $(\emptyset, E_F)$ to the state $(u, \mathcal{Q})$. In the new strategy, $(u, \mathcal{Q})$ will become unreachable. However, the state $(u_{T-1}, \mathcal{Q}_{T-1})$ may now violate the contraction condition. In this case, we recursively apply the above procedure to $(u_{T-1}, \mathcal{Q}_{T-1})$. Note that this process will terminate after a finite number of steps, as the length of the path from the initial state to $(u_{T-1}, \mathcal{Q}_{T-1})$ is strictly shorter than the path to $(u, \mathcal{Q})$.

2. Next, we prove that every reachable and non-terminal state can be strictly contracted. Suppose, for contradiction, that $(u, \mathcal{Q})$ is reachable and non-terminal, but not strictly contracted. Then there exists a transition $((u, \mathcal{Q}) \to (u', \mathcal{Q}')) \in E_{G^{\mathsf{s}}}$. Since $u$ is at the boundary of all connected

components, we have $u = u'$. This implies that the game state graph is not acyclic, which contradicts the assumption that it is a DAG.

Combining the above two points, we conclude that for any given graph $F$, if the Spoiler can win the pebble game on $F$, then there exists a game state graph $\tilde{G}^{\mathsf{S}}$ corresponding to a winning strategy for the Spoiler such that every reachable and non-terminal state is strictly contracted. □

**Lemma B.37.** *Given any connected graph $F$, if the Spoiler can win the pebble game on $F$, then $F$ is a parallel tree. Specifically, there exists a tree skeleton $T^r = (V_{T^r}, E_{T^r}, \beta_{T^r}, \gamma_{T^r})$ such that $(F, T^r) \in \mathcal{S}^{\mathsf{pt}}$.*

*Proof.* Let $G^{\mathsf{S}}$ be the game state graph satisfying Lemma B.36. For each game state $s$, denote $\mathsf{next}_{G^{\mathsf{S}}}(s)$ as the set of states $s'$ such that $(s, s')$ is a transition in $G^{\mathsf{S}}$ and $s'$ contains only a single component, i.e., $s'$ has the form $(u, \{P\})$. By definition, $\mathsf{next}_{G^{\mathsf{S}}}(\emptyset, \{E_F\}) = \{(u, \mathsf{Q}_1), \ldots, (u, \mathsf{Q}_m)\}$ for some $u \in V_F$, where $Q_1, \ldots, Q_m$ is the finest partition of $\mathsf{CC}_F(\{u\})$.

The tree $T^r$ will be recursively constructed as follows. First, create the tree root $r$ with $\beta_T(r) = u$. As will be explained later, the root node will be associated with the set of states $S(r) := \mathsf{next}_{G^{\mathsf{S}}}(\emptyset, \{E_F\})$. We then proceed with the following procedure:

Let $t$ be a leaf node in the current tree associated with a non-empty set of game states $S(t)$ such that $|\cup_{(u, \{P\}) \in S(t)} P| > 1$. For each state $(u, \{P\}) \in S(t)$, create a new node $\tilde{t}$ and set its parent to be $t$. Pick any state $(v, \{P'\}) \in \mathsf{next}_{G^{\mathsf{S}}}(u, \{P\})$, and set $\beta_T(\tilde{t}) = v$. Then, node $\tilde{t}$ will be associated with the set of states $S(\tilde{t}) = \{(v, \{\tilde{P}\}) : (v, \{\tilde{P}\}) \in \mathsf{next}_{G^{\mathsf{S}}}(u, \{P\})\}$.

We now prove that $T^r$ is indeed a valid tree skeleton for $F$. By definition of a parallel tree, when constructing $T^r$ and defining the label function $\beta_T : V_{T^r} \to V_F$, we can naturally define the label function for edges $\gamma_T : E_{T^r} \to 2^{E_F}$. For any edge $(t_1, t_2) \in E_{T^r}$, there exist only paths connecting $\beta_T(t_1)$ and $\beta_T(t_2)$ in $F$. Therefore, the image of $(t_1, t_2)$ is naturally defined as the set of paths connecting $\beta_T(t_1)$ and $\beta_T(t_2)$. Since $\beta_T$ is already defined for the nodes of $T^r$, it remains to prove that for every edge $(t_1, t_2) \in E_{T^r}$, there exist only paths connecting $\beta_T(t_1)$ and $\beta_T(t_2)$ in $F$.

We revisit the construction of $T^r$. Let $t$ be a leaf node associated with the game states $S(t)$. For each game state $(u, \{P\}) \in S(t)$, create a new node $\tilde{t}$ and set its parent to $t$. Pick any state $(v, \{P'\}) \in \mathsf{next}_{G^{\mathsf{S}}}(u, \{P\})$, and set $\beta_T(\tilde{t}) = v$. Since $(v, \{P'\}) \in \mathsf{next}_{G^{\mathsf{S}}}(u, \{P\})$, the transition $((u, \{P\}), (v, \{P'\}))$ is a legal move in the pebble game.

Moreover, since we assume that $G^{\mathsf{S}}$ satisfies Lemma B.36, we can conclude that the game state $(u, \{P\})$ is strictly contracted. In other words, $P' \subset P$. This implies that when the Spoiler places the pebble $\mathsf{p}_v$ on vertex $v \in V_F$, the Duplicator can only choose a strictly contracted connected component set. Hence, we deduce that there are only paths connecting $u$ and $v$. Consequently, there exist only paths connecting $\beta_T(t)$ and $\beta_T(\tilde{t})$.

By recursively applying this analysis throughout the construction of $T^r$, we conclude that for every edge $(t_1, t_2) \in E_{T^r}$, there exist only paths connecting $\beta_T(t_1)$ and $\beta_T(t_2)$ in graph $F$. □

We now prove finite-iteration version of Lemma B.37 as follows:

**Lemma B.38.** *Given any base graph $F$, Spoiler can win the simplified pebble game on $F$ in $d$ steps iff there exsits a parallel tree skeleton $T^r$ of $F$ such that $T^r$ has depth at most $d + 1$.*

*Proof.* Initially, it is evident that if $F$ is a parallel tree with a tree skeleton of depth at most $d + 1$, then the Spoiler has a winning strategy in $d$ steps. Therefore, we are left to consider the converse direction of the lemma. Now, consider the case where, for a base graph $F$, the Spoiler has a winning strategy in $d$ steps. According to the analysis in Lemma B.36, if the Spoiler has a winning strategy in $d$ steps, then he can guarantee that all reachable non-terminal states in the game state graph $\tilde{G}^{\mathsf{S}}$ are strictly contracted. We will prove this statement by induction. The statement trivially holds for $d = 1$. Assume that if the Spoiler has a winning strategy in $d - 1$ steps, then the base graph is a

parallel tree with a tree skeleton of depth at most $d$. Now, we consider the case where the Spoiler can win in $d$ steps.

By Lemma B.37, the Spoiler can win the game on $F$, implying that $F$ is a parallel tree. Let $T^r$ be the tree skeleton of $F$. At the beginning of the game, we first consider the case where the Spoiler places a pebble on a vertex $u$ such that $u \notin \{v : \exists t \in V_T, \beta_T(t) = v\}$. We assume that the Duplicator selects connected component $P$ (since $F$ is a parallel tree, the Duplicator can only select one connected component in this case). Assume further that there exist $t, t' \in V_T$ such that $u$ is on a path connecting $\beta_T(t)$ and $\beta_T(t')$. We now consider two separate cases:

- If there is more than one path connecting $\beta_T(t)$ and $\beta_T(t')$ in $F$, i.e., $|\gamma_T(t,t')| > 1$, then placing the pebble on $u$ does not split $F$, and it remains as one connected component. In this case, we can directly eliminate $(u, P)$ from the game state graph, and the remaining game state graph still represents a winning strategy for the Spoiler.

- If there is only one path connecting $\beta_T(t)$ and $\beta_T(t')$ in $F$, i.e., $|\gamma_T(t,t')| = 1$, then placing the pebble on $u$ splits the base graph $F$ into two connected components. In this case, we replace the game state $(u, \{P\})$ in the game state graph with $\{u', \{P'\}\}$, where $u' \in \{\beta_T(t), \beta_T(t')\}$ and $P \subset P'$.

Following this discussion, we only need to consider the case where, at the beginning of the game, the Spoiler places the pebble on a vertex $u \in V_F$ such that there exists $t \in V_T$ and $u = \beta_T(t)$. Without loss of generality, assume $u = \beta_T(r)$, and the children of $r$ are $\{t_1, \ldots, t_n\}$. Further, assume that among all subtrees induced by $t_1, t_2, \ldots, t_n$, the subtree induced by $t_1 \in V_T$ has the greatest depth. We now consider the case where the Duplicator picks the connected component formed by the subtree induced by $t_1$ and the path in $\gamma_T(r, t_1)$. If the Spoiler must ensure that the subsequent game state is strictly contracted, he must place the pebble on $t_1$. The remaining game now reduces to a game played on the graph induced by the subtree formed by all descendants of $t_1$. By the induction hypothesis, the subtree induced by $t_1$ has depth at most $d$. Thus, $T^r$ has depth at most $d + 1$. □

We now prove Lemma 3.21 from the main paper.

**Lemma B.39.** *For any $F \notin \mathcal{F}^{\mathsf{Spec},(d)}$, the Spoiler cannot win the simplified pebble game on $F$ in $d - 1$ steps. Consequently, $\chi_G^{\mathsf{Spec},(d)}(G(F)) = \chi_H^{\mathsf{Spec},(d)}(H(F))$.*

*Proof.* By Lemma B.38, since $F \notin \mathcal{F}^{\mathsf{Spec},(d)}$, the Spoiler cannot win the simplified pebble game on the base graph $F$. Thus, by Lemma 3.20, we conclude that $\chi_G^{\mathsf{Spec},(d)}(G(F)) = \chi_H^{\mathsf{Spec},(d)}(H(F))$. □

Combining all the results from steps 1 through 5, we now conclude the proof of our main theorem.

**Theorem B.40.** *The homomorphism expressivity of spectral invariant GNNs with $d$ iterations can be characterized as follows:*

$$\mathcal{F}^{\mathsf{Spec},(d)} = \{F \mid F \text{ has parallel tree depth at most } d\}.$$

*Specifically, the following properties hold:*

- *For graphs $G$ and $H$, $\chi_G^{\mathsf{Spec},(d)}(G) = \chi_H^{\mathsf{Spec},(d)}(H)$ if and only if, for all graphs $F$ with parallel tree depth at most $d$, $\mathrm{hom}(F, G) = \mathrm{hom}(F, H)$.*

- *$\mathcal{F}^{\mathsf{Spec},(d)}$ is maximal; that is, for any graph $F \notin \mathcal{F}^{\mathsf{Spec},(d)}$, there exist graphs $G$ and $H$ such that $\chi_G^{\mathsf{Spec},(d)}(G) = \chi_H^{\mathsf{Spec},(d)}(H)$ and $\mathrm{hom}(F, G) \neq \mathrm{hom}(F, H)$.*

*Proof.* By Theorem B.20 and Corollary B.10, we obtain that for graphs $G$ and $H$, $\chi_G^{\mathsf{Spec},(d)}(G) = \chi_H^{\mathsf{Spec},(d)}(H)$ if and only if, for all graphs $F$ with parallel tree depth at most $d$, $\mathrm{hom}(F, G) = \mathrm{hom}(F, H)$. Furthermore, by Lemma 3.21, there exist counterexamples $G$ and $H$ for any $F \notin \mathcal{F}^{\mathsf{Spec},(d)}$ such that $\chi_G^{\mathsf{Spec},(d)}(G) = \chi_H^{\mathsf{Spec},(d)}(H)$ and $\mathrm{hom}(F, G) \neq \mathrm{hom}(F, H)$. Thus, we conclude the proof of the main theorem. □

## C   Proof of Theorem 3.10

In this section, we provide the proof of Theorem 3.10 from the main paper.

**Theorem C.1.** *The homomorphism expressivity of graph spectra is the set of all cycles $C_n$ ($n \geq 3$) plus paths $P_1$ and $P_2$, i.e., $\{C_n | n \geq 3\} \cup \{P_1, P_2\}$.*

*Proof.* We separately prove that the set of all cycles satisfies the two conditions of homomorphism expressivity. For a graph $G$, we denote $\boldsymbol{A}_G \in \mathbb{R}^{|V_G| \times |V_G|}$ as the adjacency matrix of $G$, and $\mathsf{Spec}(G) = \{\lambda_{G,1}, \lambda_{G,2}, \ldots, \lambda_{G,|V_G|}\}$ as the spectrum of $G$.

- We first prove that for any two graphs $G$ and $H$, their spectra are identical if and only if for every $F \in \{C_n | n \geq 3\} \cup \{P_1, P_2\}$, $\mathsf{hom}(F, G) = \mathsf{hom}(F, H)$. Let $\mathcal{C}_n$ denote a cycle with $n$ vertices. For any graph $G$, we have $\mathsf{hom}(\mathcal{C}_n, G) = \mathsf{tr}(\boldsymbol{A}_G^n)$ for all $n \in \mathbb{N}_{\geq 3}$, and for $n = 2$, we denote $\mathcal{C}_2 = P_2$. Moreover, by a basic result from linear algebra, we further obtain:

$$\mathsf{hom}(\mathcal{C}_n, G) = \mathsf{tr}(\boldsymbol{A}_G^n) = \lambda_{G,1}^n + \lambda_{G,2}^n + \cdots + \lambda_{G,|V_G|}^n.$$

  Therefore, if $\mathsf{hom}(\mathcal{C}_n, G) = \mathsf{hom}(\mathcal{C}_n, H)$ for all $n \in \mathbb{N}^+$, then we have:

$$\lambda_{G,1}^n + \lambda_{G,2}^n + \cdots + \lambda_{G,|V_G|}^n = \lambda_{H,1}^n + \lambda_{H,2}^n + \cdots + \lambda_{H,|V_H|}^n, \quad \text{for all } n \in \mathbb{N}^+.$$

  Thus, $\mathsf{Spec}(G) = \mathsf{Spec}(H)$. Conversely, if we are given that $\mathsf{Spec}(G) = \mathsf{Spec}(H)$, then:

$$\mathsf{hom}(\mathcal{C}_n, G) = \mathsf{tr}(\boldsymbol{A}_G^n) = \mathsf{tr}(\boldsymbol{A}_H^n) = \mathsf{hom}(\mathcal{C}_n, H), \quad \text{for all } n \in \mathbb{N}^+.$$

  Therefore, we have proven that for any two graphs $G$ and $H$, their spectra are identical if and only if for every $F \in \{C_n | n \geq 3\} \cup \{P_1, P_2\}$, $\mathsf{hom}(F, G) = \mathsf{hom}(F, H)$.

- We now prove that for any graph $F$ that is not a cycle nor a path, there exists a pair of graphs $G$ and $H$ such that their spectra are identical, but $\mathsf{hom}(F, G) \neq \mathsf{hom}(F, H)$. Specifically, we show that for any graph $F$ that is not a cycle, $\mathsf{Spec}(G(F)) = \mathsf{Spec}(H(F))$ holds, where $G(F)$ and $H(F)$ denote the pair of Fürer graphs constructed with $F$ as the base graph.

If $F$ is not nor a path, then there exist vertices $x, y \in V_F$ such that the degree of $x$ is greater 2. We then consider the Fürer graph $G(F)$ and the twisted Fürer graph $H(F) = \mathsf{twist}(G(F), \{x, y\})$. According to **??**, for vertices $v, x_2, \ldots, x_n \in V_F$ and $\mathcal{V} \subset V_F$, the number of $n$-walks passing through $(v, \mathcal{V}), \mathsf{Meta}(x_2), \ldots, \mathsf{Meta}(x_n), (v, \mathcal{V})$ sequentially in $G(F)$ and $H(F)$ is unequal. Specifically, $x_2, \ldots, x_n$ satisfy the following properties:

1. $(v, x_2), (x_2, x_3), \ldots, (x_{n-1}, x_n), (x_n, v) \in E_F$.
2. The degree of $x_2, \ldots, x_n$ is 2 in the base graph $F$.
3. Let $x_1 = x_{n+1} = v$, then:

$$|\{\{x_i, x_{i+1}\}, i = 1, 2, \ldots, n\} \cap \{\{x, y\}\}| \equiv 1 \pmod 2.$$

From this, we deduce that $v = x$, and we have:

$$\sum_{(v,\mathcal{V}') \in \mathsf{Meta}(v)} c_{G(F)}^n((v, \mathcal{V}'), x_2, \ldots, x_n) = \sum_{(v,\mathcal{V}') \in \mathsf{Meta}(v)} c_{H(F)}^n((v, \mathcal{V}'), x_2, \ldots, x_n), \quad (6)$$

where for any vertex $(v, \mathcal{V}') \in \mathsf{Meta}(v)$, we use notations $c_{G(F)}^n((v, \mathcal{V}'), x_2, \ldots, x_n)$ and $c_{H(F)}^n((v, \mathcal{V}'), x_2, \ldots, x_n)$ to denote the number of $n$-walks passing through $(v, \mathcal{V}'), \mathsf{Meta}(x_2), \ldots, \mathsf{Meta}(x_n), (v, \mathcal{V}')$ in $G(F)$ and $H(F)$, respectively. If $x_2, \ldots, x_n$ do not satisfy the above properties, then for all $(v, \mathcal{V}') \in \mathsf{Meta}(v)$, the number of $n$-walks passing through $(v, \mathcal{V}'), \mathsf{Meta}(x_2), \ldots, \mathsf{Meta}(x_n), (v, \mathcal{V}')$ in $G(F)$ and $H(F)$ is equal. Thus, equation 6 holds for all $v, x_2, \ldots, x_n \in V_F$. Consequently, we observe the following property in terms of walk counts:

$$\sum_{(v,\mathcal{V}') \in \mathsf{Meta}(v)} \omega_{G(F)}^n((v, \mathcal{V}'), (v, \mathcal{V}')) = \sum_{(v,\mathcal{V}') \in \mathsf{Meta}(v)} \omega_{H(F)}^n((v, \mathcal{V}'), (v, \mathcal{V}')),$$

where $\omega_{G(F)}^n((v, \mathcal{V}'), (v, \mathcal{V}'))$ and $\omega_{H(F)}^n((v, \mathcal{V}'), (v, \mathcal{V}'))$ denote the number of $n$-walks starting and ending at $(v, \mathcal{V}')$ in $G(F)$ and $H(F)$, respectively. This holds for all $n \in \mathbb{N}^+$ and

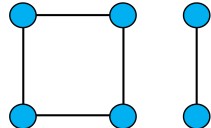 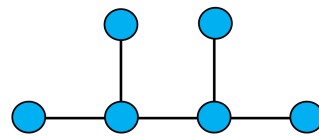

(a) Counterexample for Theorem 3.10 (Graph $G$)    (b) Counterexample for Theorem 3.10 (Graph $H$)

Figure 5: Counterexample for Theorem 3.10

all $(v, \mathcal{V}) \in \mathsf{Meta}(v)$. Thus, we conclude that $\mathsf{tr}(\boldsymbol{A}^n_{G(F)}) = \mathsf{tr}(\boldsymbol{A}^n_{H(F)})$ for all $n \in \mathbb{N}$. By a basic result from linear algebra, this implies that $\mathsf{Spec}(G(F)) = \mathsf{Spec}(H(F))$. However, since $\mathsf{hom}(F, G(F)) \neq \mathsf{hom}(F, H(F))$, we have proven that for any graph $F$ that is not a cycle, there exists a pair of graphs $G$ and $H$ such that their spectra are identical, but $\mathsf{hom}(F, G) \neq \mathsf{hom}(F, H)$.

- We now prove that for any path $F$ of length at least 2, there exist graphs $G$ and $H$ such that $\mathsf{hom}(F, G) = \mathsf{hom}(F, H)$. A pair of counterexamples is provided in Figure 5. Initially, we observe that the two graphs are cospectral. Furthermore, for any path $P$ of length $k$ ($k \geq 2$), $\mathsf{hom}(F, G) = 4 \cdot 2^k + 2$. For the graph $H$, let the number of $k$-walks starting from the vertex with degree 3 be denoted as $a_k$. We then have the following recurrence relation:

$$a_k = a_{k-1} + 2 \cdot a_{k-2}, \quad a_0 = 1, \quad a_1 = 3.$$

From this relationship, we can deduce that:

$$a_{2k+1} = 2^{2k+1} - 2^{2k+1} + \cdots + 2^2 - a_0 = 1 + 2 \cdot (1 + 4 + \cdots + 2^{2k}) = \frac{1}{3}\left(2^{2k+3} + 1\right),$$

$$a_{2k} = 2^{2k+2} - \frac{1}{3}\left(2^{2k+3} + 1\right) = \frac{1}{3}\left(2^{2k+2} - 1\right).$$

Therefore, the total number of homomorphisms from a path of length $2k+1$ to $H$ is given by:

$$\begin{aligned}
\mathsf{hom}(P_{2k+2}, H) &= 4 \cdot a_{2k} + 2 \cdot a_{2k+1} \\
&= \frac{1}{3}\left(2 \cdot 2^{2k+3} - 4\right) + \frac{1}{3}\left(2 \cdot 2^{2k+3} + 2\right) + 3 \\
&= \frac{1}{3}\left(4 \cdot 2^{2k+3} - 2\right).
\end{aligned}$$

Similarly, the total number of homomorphisms from a path of length $2k+2$ to $H$ is:

$$\begin{aligned}
\mathsf{hom}(P_{2k+2}, H) &= 4 \cdot a_{2k+1} + 2 \cdot a_{2k+2} \\
&= \frac{4}{3}\left(2^{2k+3} + 1\right) + \frac{2}{3}\left(4 \cdot 2^{2k+5} - 2\right) \\
&= 3 \cdot 2^{2k+5}.
\end{aligned}$$

Thus, for all $k \geq 3$, we conclude that:

$$\mathsf{hom}(P_k, G) \neq \mathsf{hom}(P_k, H), \quad \text{for all } k \geq 3.$$

Combining the previous three results, we have proven that homomorphic expressivity is $\{C_n | n \geq 3\} \cup \{P_1, P_2\}.$. $\qquad\square$

# D   EXPERIMENTAL DETAILS

In this section, we provide details on the experiments in Section 4. For dataset setup and training parameters, we follow Zhang et al. (2024a). We also use exactly the same model architecture for MPNN, subgraph GNN, and local 2-GNN as Zhang et al. (2024a) did.

**Model architecture of spectral invariant GNN.**   For spectral invariant GNN, we use the same feature initialization and final pooling layer as other models. The feature propogation in each layer is implemented to incorporate the eigenvalues and their projection matrices of the graph. Specifically, suppose $\{\!\{(\lambda, \boldsymbol{P}_\lambda(u, v))\}\!\}$ are all eigenvalues and eigenvectors of the input graph, $h^l(u) \in \mathbb{R}^d$ is the

feature vector of node $u$ in layer $l$. Then, the feature in next layer $l+1$ is updated according to the following rule:

$$
\begin{aligned}
h^{(l+1)}(u) &= \mathsf{ReLU}(\mathsf{BN}^{(l)}(\mathsf{MLP}_1^{(l)}((1+\epsilon^{(l)})h^{(l)}(u) + f^{(l)}(u)))), \\
f^{(l)}(u) &= \sum_{v \in \mathcal{V}} \mathsf{ReLU}(h^{(l)}(v) + \sum_{\lambda} \mathsf{MLP}_2^{(l)}(\lambda)P_\lambda(u,v)),
\end{aligned}
\tag{7}
$$

where $\mathsf{MLP}_{1,2}$ are two-layer feed-forward networks with batch normalization in the hidden layer.

Similar to Zhang et al. (2024a), for graphs with edge features, we maintain a learnable edge embedding, $g^{(l)}(u,v)$, for each type of edges, and add them to the aggregation rule $f^{(l)}(u)$. The number of layers and hidden dimensions is set to match MPNN, such that all four models have roughly the same, and obey the 500K parameter budget in ZINC, as Zhang et al. (2024a) did.

## E  HIGHER ORDER SPECTRAL INVARIANT GNN

### E.1  UPDATE RULE OF HIGHER-ORDER SPECTRAL INVARIANT GNN

A natural update rule for higher-order spectral invariant GNNs is as follows:

**Definition E.1 (Higher-Order Spectral Invariant GNN).** For any $k \in \mathbb{N}_+$, the $k$-order spectral invariant GNN maintains a color $\chi_G^{k\text{-Spec}}(\boldsymbol{u})$ for each vertex $k$-tuple $\boldsymbol{u} = (u_1, \ldots, u_k) \in V_G^k$. Initially, $\chi_G^{k\text{-Spec},(0)}(\boldsymbol{u}) = (\mathcal{P}(u_1, u_2), \ldots, \mathcal{P}(u_1, u_k), \ldots, \mathcal{P}(u_{k-1}, u_k))$. In each iteration $t+1$, the color is updated as follows:

$$
\begin{aligned}
\chi_G^{k\text{-Spec},(t+1)}(\boldsymbol{u}) = \mathsf{hash}(\chi_G^{k\text{-Spec},(t)}(\boldsymbol{u}), \{\!\!\{ (\chi_G^{k\text{-Spec},(t)}(v, u_2, \ldots, u_k), \mathcal{P}(u_1, v)) : v \in V_G \}\!\!\}, \cdots, \\
\{\!\!\{ (\chi_G^{k\text{-Spec},(t)}(u_1, u_2, \ldots, u_{k-1}, v), \mathcal{P}(u_k, v)) : v \in V_G \}\!\!\}).
\end{aligned}
$$

Denote the stable color of vertex tuple $\boldsymbol{u} \in V_G^k$ as $\chi_G^{k\text{-Spec}}(\boldsymbol{u})$. The graph representation is defined as $\chi_G^{k\text{-Spec}}(G) := \{\!\!\{ \chi_G^{k\text{-Spec}}(\boldsymbol{u}) : \boldsymbol{u} \in V_G^k \}\!\!\}$.

### E.2  HOMOMORPHISM EXPRESSIVITY OF HIGHER-ORDER SPECTRAL INVARIANT GNN

To describe the homomorphism expressivity of higher-order spectral invariant GNNs, we draw inspiration from the concept of "strong nested ear decomposition" from Zhang et al. (2024a). For the reader's convenience, we restate the relevant definitions here:

**Definition E.2 ($k$-order Ear).** A $k$-order ear is a graph $G$ formed by the union of $k$ paths $P_1, \cdots, P_k$ (possibly of zero length), along with an edge set $Q$, satisfying the following conditions:

- For each path $P_i$, let its two endpoints be $u_i$ (outer endpoint) and $v_i$ (inner endpoint). All edges in $Q$ are between inner endpoints, i.e., $Q \subset \{\{v_i, v_j\} : 1 \le i, j \le k, v_i \ne v_j\}$.

- Any two distinct paths $P_i$ and $P_j$ intersect only at their inner endpoints (if $v_i = v_j$).

- $G$ is a connected graph.

The endpoints of the $k$-order ear are the outer endpoints $u_1, \cdots, u_k$.

**Definition E.3 (Nested Interval).** Let $G$ and $H$ be two $k$-order ears with $\mathsf{inner}(G) = \{v_1, \cdots, v_k\}$, $\mathsf{outer}(G) = \{u_1, \cdots, u_k\}$, and $\mathsf{outer}(H) = \{w_1, \cdots, w_k\}$, where each $\{u_i, v_i\}$ corresponds to the endpoints of a path $P_i \in \mathsf{path}(G)$. We say $H$ is nested on $G$ if at least one endpoint $w_i$ of $H$ ($i \in [k]$) lies on the path $P_i$, and all other vertices of $H$ are not part of $G$. The nested interval is defined as the union of the subpaths $\mathsf{subpath}_{P_i}(w_i, v_i)$ for all $i \in [k]$ such that $w_i$ lies on $P_i$.

**Definition E.4 ($k$-Order Strong Nested Ear Decomposition (NED)).** A $k$-order strong NED $\mathcal{P}$ of a graph $G$ is a partition of the edge set $E_G$ into a sequence of edge sets $Q_1, \cdots, Q_m$, satisfying the following conditions:

- Each $Q_i$ is a $k$-order ear.

- Any two ears $Q_i$ and $Q_j$ with indices $1 \le i < j \le c$ do not intersect, where $c$ is the number of connected components of $G$.

- For each $Q_j$ with index $j > c$, it is nested on some $k$-order ear $Q_i$ with index $1 \leq i < j$. Moreover, except for the endpoints of $Q_j$ on $Q_i$, no other vertices in $Q_j$ belong to any previous ear $Q_k$ for $1 \leq k < i$.

- Denote by $I(Q_j) \subset Q_i$ the *nested interval* of $Q_j$ in $Q_i$. For all $Q_j$ and $Q_k$ with $c < j < k \leq m$, if $Q_j$ and $Q_k$ are nested on the same ear, then $I(Q_j) \subset I(Q_k)$.

**Definition E.5** (**Parallel $k$-Order Strong NED**). A graph $F$ is said to have a parallel $k$-order strong nested ear decomposition (NED) if there exists a graph $G$ such that $F$ can be obtained from $G$ by replacing each edge $(u, v) \in E_G$ with a parallel edge that has endpoints $(u, v)$.

With the definition of parallel $k$-order strong NED, we now state the homomorphism expressivity of $k$-spectral invariant GNN as follows:

**Theorem E.6.** *The homomorphism expressivity of a $k$-spectral invariant GNN is characterized by the set of all graphs that possess a parallel $k$-order strong NED.*

### E.3 PROOF OF THEOREM E.6

The proof of Theorem E.6 follows a similar structure to the analysis of Theorem 3.3 and Theorem 3.4 in Zhang et al. (2024a). Therefore, we provide only a brief sketch, emphasizing the key differences between the proof of Theorem E.6 and the previous analyses.

**Lemma E.7.** *For any given graphs $G$ and $H$, we have $\chi_G^{k-\mathsf{Spec}}(G) = \chi_H^{k-\mathsf{Spec}}(H)$ if and only if, for every graph $F$ that has a parallel $k$-order strong NED, $\mathsf{hom}(F, G) = \mathsf{hom}(F, H)$.*

*Proof.* We first define a parallel tree decomposition, which is a variant of the standard tree decomposition. Given a graph $G = (V_G, E_G)$, its tree decomposition is represented as a tree $T^r = (V_T, E_T, \beta_T, \gamma_T)$. The label functions $\beta_T : V_T \to 2^{V_G}$ and $\gamma_T : V_T \to 2^{P_G}$ are defined, where $P_G$ denotes the set of paths in $G$. The tree $T = (V_T, E_T, \beta_T, \gamma_T)$ satisfies the following conditions:

1. Each tree node $t \in V_T$ is associated with a non-empty subset of vertices $\beta_T(t) \subset V_G$ in $G$, referred to as a bag. Each node $t \in V_T$ is also associated with a set of paths $\gamma_T(t)$, called a sub-bag, which includes paths in $G$ that begin and end with vertices in $\beta_T(t)$. We say that a tree node $t$ contains a vertex $u$ if $u \in \beta_T(t)$, and contains a path $p$ if $p \in \gamma_T(t)$.

2. For each path $(u_1, u_2, \ldots, u_n)$ with $u_i \in V_G$ for $i \in [n]$, there exists a tree node $t \in V_T$ that contains the path, i.e., $(u_1, \ldots, u_n) \in \gamma_T(t)$.

3. For each vertex $u \in V_G$, the set of tree nodes $t$ that contain $u$, denoted by $B_T(u) = \{t \in V_T : u \in \beta_T(t)\}$, forms a non-empty connected subtree of $T$.

4. The depth of $T$ is even, i.e., $\max_{t \in V_T} \mathrm{depth}_{T^r}(t)$ is an even number.

5. $|\beta_T(t)| = k$ if $\mathrm{depth}_{T^r}(t)$ is even, and $|\beta_T(t)| = k + 1$ if $\mathrm{depth}_{T^r}(t)$ is odd.

6. For all tree edges $\{s, t\} \in E_T$, where $\mathrm{depth}_{T^r}(s)$ is even and $\mathrm{depth}_{T^r}(t)$ is odd, we have $\beta_T(s) \subset \beta_T(t)$.

We refer to $(G, T^r)$ as a parallel tree-decomposed graph and $k$ as the width of $G$'s parallel tree decomposition. The set of parallel tree-decomposed graphs with width at most $k$ is denoted as $\mathcal{S}^{k-\mathsf{Spec}}$.

Similar to the low-dimensional case, we define the unfolding tree of a $k$-spectral invariant graph neural network as follows. Given a graph $G$, a vertex $k$-tuple $\mathbf{u} = (u_1, \ldots, u_k) \in V_G^k$, and a non-negative integer $D$, the depth-$2D$ spectral $k$-spectral invariant tree of $G$ at $\mathbf{u}$, denoted $(F_G^{k-\mathsf{Spec},(D)}(\mathbf{u}), T_G^{k-\mathsf{Spec},(D)}(\mathbf{u}))$, is a parallel tree-decomposed graph $(F, T^r) \in \mathcal{S}^{k-\mathsf{Spec}}$ constructed as follows:

1. *Initialization.* Initialize $F = G[\mathbf{u}]$, and $T$ with a root node $r$ such that $\beta_T(r) = \{u_1, \ldots, u_k\}$. Define a mapping $\pi : V_F \to V_G$ by setting $\pi(\mathbf{u}) = \mathbf{u}$. For all $i, j \in [k]$ with $i \neq j$ and $r \in [n]$, if there exists an $r$-length walk $(v_1, \ldots, v_r)$ with $v_1 = u_i$ and $v_r = u_j$, we add a path $(w_1, \ldots, w_r)$ with $w_1 = u_i$ and $w_r = u_j$ to $F$, and include $(w_1, \ldots, w_r)$ in the sub-bag $\gamma_T(r)$. Moreover, we extend $\pi$ by setting $\pi(w_i) = v_i$ for all $i \in [r]$.

2. *Iterate for $D$ rounds.* For each leaf node $t \in T^r$, execute the following for each $j \in [n]$:

    (a) If $w \notin \{\pi(u_1), \ldots, \pi(u_k)\}$, add a new vertex $z$ to $F$ and extend $\pi$ by setting $\pi(z) = w$. Set $\beta_T(t_w) = \beta_T(t) \cup \{z\}$.

        Initialize $\gamma_T(t_w) = \gamma_T(t)$. For all $i \in [k]$ and $r \in [n]$, if there exists a path of length $r$, $(v_1, \ldots, v_r)$, where $v_1 = w$ and $v_r = \pi(w_i)$, we construct a corresponding path $(w_1, \ldots, w_r)$, with $w_1 = z$ and $w_r = u_i$, and include $(w_1, \ldots, w_r)$ in the sub-bag $\gamma_T(t_w)$.

    (b) If $w = \pi(u_r)$ for some $r \in [k]$, set $\beta_T(t_w) = \beta_T(t) \cup \{u_r\}$ without modifying $F$.

    For each $t_w$, add a child node $t'_w$ to $T^r$, designate $t_w$ as its parent, and update $\beta_T(t'_w)$ based on the following cases:

    (a) If $w \notin \{\pi(u_1), \ldots, \pi(u_k)\}$, set $\beta_T(t'_w) = \{u_1, \ldots, u_{j-1}, w, u_{j+1}, \ldots, u_k\}$.

    (b) If $w = \pi(u_r)$ for some $r \in [k]$, set $\beta_T(t'_w) = \{u_1, \ldots, u_{j-1}, u_r, u_{j+1}, \ldots, u_k\}$.

    Finally, set $\gamma_T(t'_w)$ as the set of all paths in $F$ that connect pairs of vertices in $\beta_T(t'_w)$.

Following a similar analysis as in the low-dimensional setting, we can first prove that for any two graphs $G$ and $H$, $\chi_G^{k-\mathsf{Spec}}(G) = \chi_H^{k-\mathsf{Spec}}(H)$ if and only if $\mathsf{treeCount}^{k-\mathsf{Spec}}((F, T^r), G) = \mathsf{treeCount}^{k-\mathsf{Spec}}((F, T^r), H)$ holds for all $(F, T^r) \in \mathcal{S}^{k-\mathsf{Spec}}$. We define

$$\mathsf{treeCount}^{\mathsf{Spec}}((F, T^r), G)$$
$$:= \left| \left\{ \mathbf{u} \in V_G^k : \exists D \in \mathbb{N}_+ \text{ such that } \left( F_G^{k-\mathsf{Spec},(D)}(\mathbf{u}), T_G^{k-\mathsf{Spec},(D)}(\mathbf{u}) \right) \cong (F, T^r) \right\} \right|.$$

With similar arguments as in Theorem 3.4 in Zhang et al. (2024a), we can further prove that for any two graphs $G$ and $H$, $\mathsf{treeCount}((F, T^r), G) = \mathsf{treeCount}((F, T^r), H)$ holds for all tree-decomposed graphs $(F, T^r)$ if and only if $\hom(F, G) = \hom(F, H)$ holds. We now prove that a graph $F$ has a parallel tree decomposition with width at most $k$ if and only if $F$ admits a parallel $k$-order strong NED. We prove each direction separately. First, we use induction on the number of vertices in $F$ to show that for any $(F, T^r) \in \mathcal{S}^{k-\mathsf{Spec}}$ with $\beta_T(r) = \{u_1, u_2, \ldots, u_k\}$, there exists a graph $\tilde{F}$ with a strong NED such that $\{u_1, \ldots, u_k\}$ are the endpoints of the first ear. We can construct $F$ by replacing edges in $\tilde{F}$ with parallel edges. For the converse direction, assume that $F$ admits a parallel $k$-order strong NED. We aim to prove that there exists a parallel tree decomposition $T^r$ of $F$ such that $(F, T^r) \in \mathcal{S}^{k-\mathsf{Spec}}$. We proceed by induction on the number of vertices and prove a stronger statement. For any connected graph $F$, if $F$ can be constructed by replacing edges in a graph $\tilde{F}$ with parallel edges, where $\tilde{F}$ has a $k$-order strong NED and the endpoints of the first ear are $\{u_1, u_2, \ldots, u_k\}$, then there exists a tree decomposition $T^r$ of $F$. This decomposition satisfies $(F, T^r) \in \mathcal{S}^{k-\mathsf{Spec}}$, and $\beta_T(r) = \{u_1, u_2, \ldots, u_k\}$. By combining the proofs for both directions, we conclude the proof of the lemma. $\qquad \square$

We then prove the maximality of homomorphism expressivity as follows.

**Lemma E.8.** *For any connected graph $F \notin \mathcal{F}^{k-\mathsf{Spec}}$, there exist graphs $G$ and $H$ such that $\hom(F, G) \neq \hom(F, H)$ and $\chi_G^{k-\mathsf{Spec}}(G) = \chi_H^{k-\mathsf{Spec}}(H)$.*

*Proof.* As in the low-dimensional case, we consider a pebble game between two players, the Spoiler and the Duplicator. The game involves a graph $F$ and several pebbles. Initially, all pebbles are placed outside the graph. During the course of the game, some pebbles are placed on the vertices of $F$, which divides the edges $E_F$ into connected components. In each round, the Spoiler updates the position of the pebbles, while the Duplicator manages a subset of connected components, ensuring that the number of selected components is *odd*. There are three main types of operations:

1. *Adding a pebble* p: the Spoiler places a pebble p (which was previously outside the graph) on some vertex of $F$. If adding this pebble does not change the connected components, the Duplicator does nothing. Otherwise, some connected component $P$ is divided into several components $P = \bigcup_{i \in [m]} P_i$ for some $m$. the Duplicator updates his selection as follows: if $P$ was selected, he removes $P$ and adds a subset of $\{P_1, \ldots, P_m\}$, while ensuring that the total number of selected components remains odd.

2. *Removing a pebble* p: the Spoiler removes a pebble p from a vertex. If this action does not alter the connected components, the Duplicator again does nothing. Otherwise, several connected

components $P_1, \ldots, P_m$ merge into a single component $P = \bigcup_{i \in [m]} P_i$. the Duplicator updates his selection by removing all selected $P_i$ and optionally adding $P$, while ensuring the total number of selected components is odd.

3. *Swapping two pebbles* $\mathsf{p}$ *and* $\mathsf{p}'$: the Spoiler swaps the positions of two pebbles, which does not affect the connected components, and therefore the Duplicator does nothing.

the Spoiler wins the game if, at any point, there exists a path $p$ such that both of its endpoints are covered by pebbles and the connected component containing $\{p\}$ is selected by the Duplicator. If the Spoiler cannot achieve this throughout the game, the Duplicator wins. In the case of the $k$-spectral invariant GNN, there are $k + 1$ pebbles, denoted $\mathsf{p}_{u_1}, \ldots, \mathsf{p}_{u_k}, \mathsf{p}_v$. Initially, all pebbles are placed outside the graph. the Spoiler first sequentially adds the pebbles $\mathsf{p}_{u_1}, \ldots, \mathsf{p}_{u_k}$ (using operation 1). The game proceeds in a cyclical manner. In each round, Spoiler selects an $r \in [k]$ and freely chooses one of the following two actions:

- For $r = 1, 2, \ldots, k$, Spoiler removes pebble $\mathsf{p}_{u_r}$ (operation 2), and then re-adds it (operation 1).

- For $r = 1, 2, \ldots, k$, Spoiler adds pebble $\mathsf{p}_w$ (operation 1) adjacent to $\mathsf{p}_{u_r}$, swaps $\mathsf{p}_{u_r}$ with $\mathsf{p}_w$ (operation 3), and then removes $\mathsf{p}_w$ (operation 2).

For a given graph $F$, let $G(F)$ and $H(F)$ denote the Fürer graph and the twisted Fürer graph with respect to $F$. Using similar reasoning as in the low-dimensional case, we can show that if the Spoiler cannot win the pebble game on $F$, then $\chi_{G(F)}^{k-\mathsf{Spec}}(G(F)) = \chi_{H(F)}^{k-\mathsf{Spec}}(H(F))$. Furthermore, analogous to the analysis of Lemma B.37, we can conclude that if the Spoiler wins the pebble game on $F$, then there exists a parallel tree decomposition $T^r$ of $F$ such that $(F, T^r) \in \mathcal{S}^{k-\mathsf{Spec}}$. Thus, for any connected graph $F \notin \mathcal{F}^{k-\mathsf{Spec}}$, there exist graphs $G(F)$ and $H(F)$ such that $\mathsf{hom}(F, G(F)) \neq \mathsf{hom}(F, H(F))$ and $\chi_G^{k-\mathsf{Spec}}(G(F)) = \chi_H^{k-\mathsf{Spec}}(H(F))$. This completes the proof of the lemma. $\square$

Finally, the proof of Theorem E.6 is completed by combining the results from Lemma E.7 and Lemma E.8.

# F PROOF FOR SYMMETRIC POWER

## F.1 PROPERTIES OF LOCAL $k-$GNN

In this section, we review key properties of the local $k$-GNN as presented in previous works. We begin by formally introducing the update rule for the local $k$-GNN.

**Definition F.1.** Local $k$-GNN maintains a color $\chi_G^{\mathsf{L}(k)}(\boldsymbol{u})$ for each vertex $k$-tuple $\boldsymbol{u} \in V_G^k$. Initially, $\chi_G^{\mathsf{L}(k),(0)}(\boldsymbol{u}) = \mathsf{atp}_G(\boldsymbol{u})$, called the isomorphism type of vertex $k$-tuple $\boldsymbol{u}$, where $\mathsf{atp}_G(\boldsymbol{u})$ is the *atomic type* of $\boldsymbol{u}$. Then, in each iteration $t + 1$,

$$\chi_G^{\mathsf{L}(k),(t+1)}(\boldsymbol{u}) = \mathsf{hash}\left(\chi_G^{\mathsf{L}(k),(t)}(\boldsymbol{u}), \{\!\!\{\chi_G^{\mathsf{L}(k),(t)}(\boldsymbol{v}) : \boldsymbol{v} \in N_G^{(1)}(\boldsymbol{u})\}\!\!\}, \cdots, \{\!\!\{\chi_G^{\mathsf{L}(k),(t)}(\boldsymbol{v}) : \boldsymbol{v} \in N_G^{(k)}(\boldsymbol{u})\}\!\!\}\right),$$
(8)

where $N_G^{(j)}(\boldsymbol{u}) = \{(u_1, \cdots, u_{j-1}, w, u_{j+1}, \cdots, u_k) : w \in N_G(u_j)\}$. Denote the stable color as $\chi_G^{\mathsf{L}(k)}(\boldsymbol{u})$. The representation of graph $G$ is defined as $\chi_G^{\mathsf{L}(k)}(G) := \{\!\!\{\chi_G^{\mathsf{L}(k)}(\boldsymbol{u}) : \boldsymbol{u} \in V_G^k\}\!\!\}$.

**Definition F.2 (Canonical Tree Decomposition).** Given a graph $G = (V_G, E_G)$, a canonical tree decomposition of width $k$ is a rooted tree $T^r = (V_T, E_T, \beta_T)$ satisfying the following conditions:

1. The depth of $T$ is even, i.e., $\max_{t \in V_T} \mathsf{dep}_{T^r}(t)$ is even;

2. Each tree node $t \in V_T$ is associated to a multiset of vertices $\beta_T(t) \subset V_G$, called a *bag*. Moreover, $|\beta_T(t)| = k$ if $\mathsf{dep}_{T^r}(t)$ is *even* and $|\beta_T(t)| = k + 1$ if $\mathsf{dep}_{T^r}(t)$ is *odd*;

3. For all tree edges $\{s, t\} \in E_T$ where $\mathsf{dep}_{T^r}(s)$ is even and $\mathsf{dep}_{T^r}(t)$ is odd, $\beta_T(s) \subset \beta_T(t)$ (where "$\subset$" denotes the multiset inclusion relation);

4. For each edge $\{u, v\} \in V_G$, there exists at least one tree node $t \in V_T$ that contains the edge, i.e., $\{u, v\} \subset \beta_T(t)$;

5. For each vertex $u \in V_G$, all tree nodes $t$ whose bag contains $u$ form a (non-empty) collection.

We further define set $\mathcal{S}^{\mathsf{L}(k)}$ as follows:

**Definition F.3.** $(F, T^r) \in \mathcal{S}^{\mathsf{L}(k)}$ iff $(F, T^r)$ satisfies F.2 with width $k$, and any tree node $t$ of odd depth has only one child. Moreover, all vertex of $F$ is contained in at least one node of $t$.

Then, we can obtain the following theorem of the homomorphic expressivity of Local $k$-GNN.

**Theorem F.4.** *Any graph $G$ and $H$ have the same representation under Local $k-$GNN (i.e.,* $\chi_G^{\mathsf{L}(k)}(G) = \chi_H^{\mathsf{L}(k)}(H)$) *iff* $\mathsf{hom}(F, G) = \mathsf{hom}(F, H)$ *for all* $(F, T^r) \in \mathcal{S}^{\mathsf{L}(k)}$.

### F.2 MAIN RESULT

Since $2k$-local GNN is strictly weaker than $2k$-WL, we aim to extend previous result by showing that $2k$-local GNN can encode $k$-symmetric power of a graph. We state our main result as follows:

**Theorem F.5.** *The Local $2k$-GNN defined in Morris et al. (2020); Zhang et al. (2024a) can encode the symmetric $k$-th power. Specifically, for given graphs $G$ and $H$, if $G$ and $H$ have the same representation under Local $2k$-GNN, then $G^{\{k\}}$ and $H^{\{k\}}$ have the same representation under the spectral invariant GNN defined in Section 2.1.*

### F.3 PROOF OF THEOREM F.5

**Definition F.6.** Let $\mu_1 < \mu_2 < \cdots < \mu_m$ represent the distinct eigenvalues of the $k$-th order symmetric power matrix of a graph $G$. Let $E_i$ denote the eigenspace corresponding to $\mu_i$, and $P_i^S$ the orthogonal projection matrix from $\mathbb{R}^{C_n^k}$ onto $E_i$. For $u_1, u_2, \ldots, u_{2k} \in V_G$, if both $\{\!\{u_1, u_2, \ldots, u_k\}\!\}$ and $\{\!\{u_{k+1}, u_{k+2}, \ldots, u_{2k}\}\!\}$ are multisets of $k$ distinct vertices, then we define

$$P_*^S(S_1, S_2) = (P_1^S(S_1, S_2), \ldots, P_m^S(S_1, S_2)),$$

where $S_1 = \{\!\{u_1, u_2, \ldots, u_k\}\!\}$ and $S_2 = \{\!\{u_{k+1}, u_{k+2}, \ldots, u_{2k}\}\!\}$. Otherwise, we define

$$P_*^S(\{\!\{u_1, u_2, \ldots, u_k\}\!\}, \{\!\{u_{k+1}, u_{k+2}, \ldots, u_{2k}\}\!\}) = \mathbf{0}.$$

We encode the spectral information of the symmetric power into the aggregation of local $2k$-GNN, resulting in a variant of the local $2k$-GNN, defined as follows:

**Definition F.7.** A local $2k$-GNN with symmetric power maintains a color $\chi_G^{\mathsf{SL}(2k)}(\boldsymbol{u})$ for each vertex $2k$-tuple $\boldsymbol{u} \in V_G^{2k}$. Initially, the color is defined as

$$\chi_G^{\mathsf{SL}(2k),(0)}(\boldsymbol{u}) = \left( P_*^S(\{\!\{u_1, \cdots, u_k\}\!\}, \{\!\{u_{k+1}, \cdots, u_{2k}\}\!\}), \mathsf{atp}_G(\boldsymbol{u}) \right).$$

Then, at each iteration $t + 1$, the update rule is given by:

$$\chi_G^{\mathsf{SL}(2k),(t+1)}(\boldsymbol{u}) = \mathsf{hash}\left( \chi_G^{\mathsf{SL}(2k),(t)}(\boldsymbol{u}), \{\!\{\chi_G^{\mathsf{SL}(2k),(t)}(\boldsymbol{v}) : \boldsymbol{v} \in N_G^{(1)}(\boldsymbol{u})\}\!\}, \ldots, \right.$$
$$\left. \{\!\{\chi_G^{\mathsf{L}(2k),(t)}(\boldsymbol{v}) : \boldsymbol{v} \in N_G^{(k)}(\boldsymbol{u})\}\!\} \right), \tag{9}$$

where $N_G^{(j)}(\boldsymbol{u}) = \{(u_1, \cdots, u_{j-1}, w, u_{j+1}, \cdots, u_k) : w \in N_G(u_j)\}$.

The stable color is denoted as $\chi_G^{\mathsf{SL}(k)}(\boldsymbol{u})$. The graph representation is then defined as

$$\chi_G^{\mathsf{SL}(k)}(G) := \{\!\{\chi_G^{\mathsf{SL}(k)}(\boldsymbol{u}) : \boldsymbol{u} \in V_G^{2k}\}\!\}.$$

Next, we define the concept of a $k$-dimensional path as follows:

**Definition F.8.** For a graph $G$ and vertices $u_1, \ldots, u_k \in V_G$, we define the neighboring multiset of $\{\!\{u_1, u_2, \ldots, u_k\}\!\}$ as:

$$N_G\left(\{\!\{u_1, u_2, \ldots, u_k\}\!\}\right) = \bigcup_{r=1}^{k} \{\{\!\{u_1, \cdots, u_{r-1}, v, u_{r+1}, \cdots, u_k\}\!\} \mid v \in N_G(u_r)\}.$$

A $k$-dimensional walk of length $n$ is defined as a sequence $(S_1, S_2, \ldots, S_n)$, where each $S_1, \ldots, S_n$ is a multiset of $k$ elements, and for all $r \in [n-1]$, $S_r \in N_G(S_{r+1})$. If the path further satisfies the condition that for all $u \in S_r$ with $r \in \{2, 3, \ldots, n-1\}$ and $v \in V_G$, $u \in N_G(v)$ implies $v \in S_i$ for some $i \in \{r-1, r, r+1\}$, then we denote $(S_1, \ldots, S_n)$ as a $k$-dimensional path of length $n$.

We then define set $\mathcal{S}^{\mathsf{SL}(k)}$ base on the definition of set $\mathcal{S}^{\mathsf{L}(k)}$.

**Definition F.9.** $(F, T^r) \in \mathcal{S}^{\mathsf{SL}(2k)}$ iff $(F, T^r)$ satisfies definition F.2 with width $2k$, and any tree node $t$ of odd depth has only one child. Furthermore, for tree node $t \in V_T$ if $\mathsf{dep}_{T^r}(t)$ is even, we further associate it with a set of $k-$dimensional path $\gamma_T(t)$, called sub-bag. Specifically, for node $t \in V_T$, let $\beta_T(t) = \{\!\{u_1, \ldots, u_{2k}\}\!\}$, then $\gamma_T(t)$ contains $k$-dimensional path linking $\{\!\{u_1, \ldots, u_k\}\!\}$ and $\{\!\{u_{k+1}, \ldots, u_{2k}\}\!\}$. Each vertex of $F$ is contained in at least one node of $T^r$, either in bags or sub-bags.

**Lemma F.10.** *Any graph $G$ and $H$ have the same representation under Local $2k$-GNN with symmetric power if* $\mathsf{hom}(F, G) = \mathsf{hom}(F, H)$ *for all* $(F, T^r) \in \mathcal{S}^{\mathsf{SL}(2k)}$.

*Proof.* To prove the theorem, we first define unfolding tree of local $2k$-GNN with symmetric power. Given a graph $G$, $2k$-tuple $\boldsymbol{u} \in V_G^{2k}$ and a non-negative integer $D$, the depth-$D$ unfolding tree of graph $G$ at tuple $\boldsymbol{u}$, denoted as $(F_G^{\mathsf{SL}(D)}(\boldsymbol{u}), T_G^{\mathsf{SL}(D)}(\boldsymbol{u}))$ is constructed as follows:

1. *Initialization.* We assume multiset $\boldsymbol{u} = \{\!\{u_1, u_2, \cdots, u_{2k}\}\!\}$. At the beginning, $F = G[\{\!\{u_1, u_2, \cdots, u_{2k}\}\!\}]$, and $T$ only has a root node $r$ with $\beta_T(r) = \{\!\{u_1, u_2, \cdots, u_{2k}\}\!\}$. Define a mapping $\pi : V_F \to V_G$ as $\pi(u_i) = u_i, \forall i \in [2k]$. For every $k$-dimensional walk $\{\!\{u_1, \ldots, u_k\}\!\} = S_1, \ldots, S_n = \{\!\{u_{k+1}, \ldots, u_{2k}\}\!\}$ with $n \leq |V_G|^k$, we introduce a $k$-dimensional path $\{\!\{u_1, \ldots, u_k\}\!\} = S_1', S_2', \ldots, S_n' = \{\!\{u_{k+1}, \ldots, u_{2k}\}\!\}$, and we add $(S_1', S_2', \ldots, S_n')$ to sub-bag $\gamma_T(r)$.

2. *Loop for $D$ rounds.* For each leaf node $t$ in $T^r$, do the following procedure for all $i \in [2k]$: Let $\beta_T(t) = \{\!\{u_1, \ldots, u_{2k}\}\!\}$. For each $w \in V_G$, add a fresh child node $t_w$ to $T$ and designate $t$ as its parent. Then, consider the following two cases:

   (a) If $w \notin \{\!\{\pi(u_1), \ldots, \pi(u_{2k})\}\!\}$, then add a fresh vertex $z$ to $F$ and extend $\pi$ with $\pi(z) = w$. Define $\beta_T(t_w) = \beta_T(t) \cup \{\!\{z\}\!\}$. Then, add edges between $z$ and $\beta_T(t)$, so that $\pi$ is an isomorphism from $F[\beta_T(t_w)]$ to $G[\pi(\beta_T(t_w))]$.

   (b) If $w \in \{\!\{\pi(u_1), \ldots, \pi(u_{2k})\}\!\}$, let $w = \pi(u_r)$. Then, we simply set $\beta_T(t_w) = \beta_T(t) \cup \{\!\{u_r\}\!\}$ without modifying graph $F$.

   Next, add a fresh child node $t_w'$ in $T^r$, designate $t_w$ as its parent, and set $\beta_T(t_w')$ and $\gamma_T(t_w')$ based on the following two cases:

   (a) If $w \notin \{\!\{\pi(u_1), \ldots, \pi(u_{2k})\}\!\}$, then $\beta_T(t_w') = \{\!\{u_1, \ldots, u_{i-1}, z, u_{i+1}, \ldots, u_{2k}\}\!\}$. For every $k$-dimensional walk linking $\pi(S_1)$ and $\pi(S_n)$ of length $n$ $(n \leq |V_G|^k)$, we introduce $k-$dimensional path $S_1 = S_1', S_2', \ldots, S_n' = S_n$. If $i < k$, then $S_1 = \{\!\{u_1, \ldots, u_{i-1}, z, \ldots, u_k\}\!\}$. If $i = k$, then $S_1 = \{\!\{u_1, \ldots, u_{i-1}, z\}\!\}$, while if $i > k$, then $S_1 = \{\!\{u_1, \ldots, u_k\}\!\}$. We denote $S_2 = \{\!\{u_1, \ldots, u_{i-1}, z, u_{i+1}, \ldots, u_{2k}\}\!\} \setminus S_1$. We add $(S_1', S_2', \ldots, S_n')$ into sub-bag $\gamma_T(t_w')$.

   (b) Conversely, if $w \in \{\!\{\pi(u_1), \ldots, \pi(u_{2k})\}\!\}$, we assume that $w = \pi(u_r)$. Then $\beta_T(t_w') = \{\!\{u_1, \ldots, u_{i-1}, u_r, u_{i+1}, \ldots, u_{2k}\}\!\}$. For every $k$-dimensional walk linking $\pi(S_1)$ and $\pi(S_n)$ of length $n$ $(n \leq |V_G|^k)$, we introduce $k-$dimensional path $S_1 = S_1', S_2', \ldots, S_n' = S_n$. If $i < k$, then $S_1 = \{\!\{u_1, \ldots, u_{i-1}, u_r, \ldots, u_k\}\!\}$. If $i = k$, then $S_1 = \{\!\{u_1, \ldots, u_{i-1}, u_r\}\!\}$, while if $i > k$, then $S_1 = \{\!\{u_1, \ldots, u_k\}\!\}$. We denote $S_2 = \{\!\{u_1, \ldots, u_{i-1}, u_r, u_{i+1}, \ldots, u_{2k}\}\!\} \setminus S_1$. We add $(S_1', S_2', \ldots, S_n')$ into sub-bag $\gamma_T(t_w')$.

We can see from the construction of unfolding tree that for all $k$-tuple $\mathbf{u} \in V_G^{2k}$ and $D > 0$, $(F_G^{\mathsf{SL}(D)}(\mathbf{u}), T_G^{\mathsf{SL}(D)}(\mathbf{u})) \in \mathcal{S}^{\mathsf{SL}(2k)}$. Given $(F, T^r), (\tilde{F}, \tilde{T}^r) \in \mathcal{S}^{\mathsf{SL}(2k)}$, we define a pair of mapping $(\rho, \tau)$ as an isomorphism from $(F, T^r)$ to $(\tilde{F}, \tilde{T}^r)$, denoted by $(F, T) \cong (\tilde{F}, \tilde{T}^r)$, if the following hold:

1. $\rho$ is an isomorphism from $F$ to $\tilde{F}$.

2. $\tau$ is an isomorphism from $T^r$ to $\tilde{T}^r$ (ignoring $\beta$ and $\gamma$).

3. For any $t \in V_{T^r}$, $\rho(\beta_{T^r}(t)) = \beta_{\tilde{T}^r}(\tau(t))$, and $\rho(\gamma_{T^r}(t)) = \gamma_{\tilde{T}^r}(\tau(t))$.

With similar analysis as Theorem B.8 we obtain that for any $k$-tuple $\mathbf{u} \in V_G^k$ and $\mathbf{v} \in V_H^k$, $\chi_G^{\mathsf{SL}(2k)(D)}(\mathbf{u}) = \chi_H^{\mathsf{SL}(2k)(D)}(\mathbf{v})$ if there exists an isomorphism $(\rho, \tau)$ from $(F_G^{\mathsf{SL}(D)}(\mathbf{u}), T_G^{\mathsf{SL}(D)}(\mathbf{u}))$

to $(F_H^{\mathsf{SL}(D)}(\mathbf{v}), T_H^{\mathsf{SL}(D)}(\mathbf{v}))$. Given a graph $G$ and $(F, T^r) \in \mathcal{S}^{\mathsf{SL}(2k)}$, we define

$$\mathsf{treeCount}^{\mathsf{SL}(2k)}((F, T^r), G) := \left| \left\{ \mathbf{u} \in V_G^{2k} : \exists D \in \mathbb{N}_+ s.t. \left( F_G^{\mathsf{SL}(D)}(\mathbf{u}), T_G^{\mathsf{SL}(D)}(\mathbf{u}) \right) \cong (F, T^r) \right\} \right|.$$

Therefore, we can obtain that if $\mathsf{treeCount}^{\mathsf{SL}(2k)}((F, T^r), G) = \mathsf{treeCount}^{\mathsf{SL}(2k)}((F, T^r), H)$ for all $(F, T^r) \in \mathcal{S}^{\mathsf{SL}(2k)}$, then $\chi_G^{\mathsf{SL}(2k)}(G) = \chi_H^{\mathsf{SL}(2k)}(H)$. Additionally, with similar analysis as Theorem B.20 and Theorem B.14, we can obtain that $\mathsf{treeCount}^{\mathsf{SL}(2k)}((F, T^r), G) = \mathsf{treeCount}^{\mathsf{SL}(2k)}((F, T^r), H)$ for all $(F, T^r) \in \mathcal{S}^{\mathsf{SL}(2k)}$ if and only if $\mathsf{hom}((F, T^r), G) = \mathsf{hom}((F, T^r), H)$ for all $(F, T^r) \in \mathcal{S}^{\mathsf{SL}(2k)}$. Therefore, we can obtain that if $\mathsf{hom}((F, T^r), G) = \mathsf{hom}((F, T^r), H)$ for all $(F, T^r) \in \mathcal{S}^{\mathsf{SL}(2k)}$, then $\chi_G^{\mathsf{SL}(2k)}(G) = \chi_H^{\mathsf{SL}(2k)}(H)$. Thus, we finish the proof of the lemma. □

**Lemma F.11.** *For all $k \geq 1$, we have $\mathcal{S}^{\mathsf{L}(2k)} = \mathcal{S}^{\mathsf{SL}(2k)}$.*

*Proof.* We can directly see that $\mathcal{S}^{\mathsf{L}(2k)} \subset \mathcal{S}^{\mathsf{SL}(2k)}$, so it is sufficed to prove that for all $(F, T^r) \in \mathcal{S}^{\mathsf{SL}(2k)}$, there exists an alternative tree decomposition $\tilde{T}^r$ such that $(F, \tilde{T}^r) \in \mathcal{S}^{\mathsf{SL}(2k)}$. We will prove that for $(F, T^r) \in \mathcal{S}^{\mathsf{SL}(2k)}$, if $\max_{t \in V_{T^r}} |\gamma_T(t)| \geq 1$, then we can construct $\tilde{T}^r$ such that $\max_{t \in V_{\tilde{T}^r}} |\gamma_{\tilde{T}}(t)| < \max_{t \in V_{T^r}} |\gamma_T(t)|$. For $(F, T^r) \in \mathcal{S}^{\mathsf{SL}(2k)}$, let $t = \arg \max_{\tilde{t} \in V_{T^r}} |\gamma_T(\tilde{t})|$ and suppose $k$-dimensional path $(S_1, S_2, \ldots, S_n) \in \gamma_T(t)$. We apply the following modification to $T^r$ to construct $\tilde{T}^r$:

1. We construct tree node $t_1, t_2, \ldots, t_{n-1}$ and $\hat{t}_1, \hat{t}_2, \ldots, \hat{t}_{n-1}$ such that $\beta_{\tilde{T}}(t_r) = S_{r+1} \cup S_r \cup S_n$ and $\beta_{\tilde{T}}(\hat{t}_r) = S_{r+1} \cup S_n$ for all $r \in [n-1]$.

2. We add $\hat{t}_r$ as the child node of $t_r$ for all $r \in [n-1]$ and add $t_r$ as the child node of $\hat{t}_{r-1}$ for all $r \in \{2, 3, \ldots, n-1\}$. Eventually, we add $t_1$ as the child node of $t$.

3. We delete $k$-dimensional path from $\gamma_T(t)$ and keep the bags of all $t \in V_T$ and sub-bags of vertices in $V_T \setminus \{t\}$ unchanged. Namely, we assume that $\gamma_{\tilde{T}}(t) = \gamma_T(t) \setminus \{(S_1, \ldots, S_n)\}$. Moreover, $\beta_T(t) = \beta_{\tilde{T}}(t)$ for all $t \in V_T$ and $\gamma_T(t) = \gamma_{\tilde{T}}(t)$ for all $V_T \setminus \{t\}$.

With the procedure above we can obtain $(F, \tilde{T}^r) \in \mathcal{S}^{\mathsf{SL}(2k)}$ such that $\max_{t \in \tilde{T}^r} |\gamma_{\tilde{T}}(t)| < \max_{t \in T^r} |\gamma_T(t)|$. If we recursively apply this procedure to modify $\tilde{T}^r$, we can eventually obtain $\hat{T}^r$ such that $\max_{t \in \hat{T}^r} |\gamma_{\hat{T}}(t)| = 0$. Therefore, $(F, \hat{T}^r) \in \mathcal{S}^{\mathsf{L}(2k)}$. Eventually, we have proven that for all $(F, T^r) \in \mathcal{S}^{\mathsf{SL}(2k)}$, there exists an alternative decomposition $\tilde{T}^r$ such that $(F, \tilde{T}^r) \in \mathcal{S}^{\mathsf{L}(2k)}$. Thus, for all $k \geq 1$, $\mathcal{S}^{\mathsf{L}(2k)} = \mathcal{S}^{\mathsf{SL}(2k)}$. □

Finally, we can finish the proof of Theorem F.5.

**Theorem F.12.** *The Local $2k$-GNN defined in Morris et al. (2020); Zhang et al. (2024a) can encode the symmetric $k$-th power. Specifically, for given graphs $G$ and $H$, if $G$ and $H$ have the same representation under Local $2k$-GNN, then $G^{\{k\}}$ and $H^{\{k\}}$ have the same representation under the spectral invariant GNN defined in Section 2.1.*

*Proof.* According to Lemma F.11, the homomorphism expressivity of the vanilla Local $2k$-GNN is equivalent to that of the Local $2k$-GNN with symmetric power. Hence, the expressive power of the Local $2k$-GNN is the same as that of the Local $2k$-GNN with symmetric power. If there exist graphs $G$ and $H$ such that $\chi_G^{\mathsf{L}(k)}(G) = \chi_H^{\mathsf{L}(k)}(H)$, then it must follow that $\chi_G^{\mathsf{SL}(k)}(G) = \chi_H^{\mathsf{SL}(k)}(H)$. Therefore, we also have $\chi_G^{\mathsf{SL}(k),(0)}(G) = \chi_H^{\mathsf{SL}(k),(0)}(H)$, meaning that the symmetric $k$-th powers of $G$ and $H$ are cospectral. Moreover, it is straightforward that if graphs $G$ and $H$ have the same representation under a Local $2k$-GNN with symmetric power, then $G^{\{k\}}$ and $H^{\{k\}}$ also have the same representation under the spectral invariant GNN. Thus, the proof of the theorem is complete. □

