# OpenReview forum: "Homomorphism Expressivity of Spectral Invariant Graph Neural Networks"
_ICLR.cc/2025/Conference — ICLR 2025 Oral_

### Official Review · Reviewer_B3EZ · 2024-10-28

**Soundness:** 4
**Presentation:** 4
**Contribution:** 4
**Rating:** 10
**Confidence:** 5

**Summary:**

The present manuscript tackles the theoretical understanding of the power of spectral invariants over graphs, in particular linking such power to the expressive power of GNNs built accordingly. The paper provides a comprehensive analysis of the expressive power of such invariants by exploiting the concept of homomorphism expressivity, which characterizes class of graphs in term of pseudo-isomorphism test algorithms, such as coloring algorithms. An experimental framework is also provided to validate the theoretical findings.

**Strengths:**

The manuscript presents an impressive comprehensive overview over the expressive power of spectral invariants, bridging together many elements from the literature and providing answers to open problems on the topic.
The presentation of the paper is well structured and made very clear to the reader.

**Weaknesses:**

I do not find any particular flaw in the manuscript. The argumentations seem solid and the scientific analysis is carried on in a proper way.

**Questions:**

Corollaries 3.11 and 3.13 state that increasing the number of iterations would make the spectral invariant GNNs more powerful. How would the authors relate such a theoretical statement with the known issues of over smoothing and gradient vanishing in GNNs? A comment in this regard would be insightful.

I wonder how the datasets used for the experimental validation were selected. Is there a particular criterion?

---

> ### Author Response · Authors · 2024-11-24
> **Response to Reviewer B3EZ**
>
> We sincerely thank reviewer B3EZ for the careful reading, proof checking, and positive feedback on our work. We now address your questions below:
>
> **Regarding oversmoothing and gradient vanishing in GNNs.** Thanks for raising this insightful question. Indeed, there is a trade-off between the expressive power and optimization difficulty when increasing the number of layers. If the model depth is too large, then the expressiveness improvement will be marginal, but the oversmoothing problem will become much severe. Although our theory implies that spectral invariant GNNs requires a linear number of model depth in order to achieve maximal expressive power, such a theoretically maximal expressivity corresponds to a **worst-case** analysis and may not be necessary in practice. Instead, our theory also suggests that a constant number of layers already suffices to achieve some practical expressivity such as counting cycles. Generally, we believe that for most practical tasks, the optimal model depth should not depend on the input graph size (when taking both optimization and efficiency into consideration).
>
> **Regarding dataset selection.** Our datasets are generally selected following prior work [1], with several slight modifications.
>
> - **Real-world tasks.** We adopt the ZINC Subset and ZINC Full datasets for real-world tasks, as they are widely used in prior literature regarding the expressive power of GNNs (e.g., [1,2,3,4]).
>
> - **Synthetic dataset for homomorphism counting.** We follow a similar setup to [1], but with different substructures. Here, we carefully select five substructures to highlight the gaps in homomorphism expressivity among various GNN models, thereby validating our theoretical findings. Specifically:
>
>    - All five substructures are non-trees and, therefore, fall outside the homomorphism expressivity of MPNNs.
>    - The first two graphs are parallel trees and lie within the homomorphism expressivity of Spectral Invariant GNNs, whereas the other three graphs do not.
>    - The first three graphs admit an endpoint-shared nested ear decomposition (NED), while the other two do not. Consequently, the first three graphs are within the homomorphism expressivity of Subgraph GNNs (according to [1]), while the remaining two are not.
>    - Finally, only the last graph does not admit a strong NED. Thus, according to [1], the first four graphs fall within the homomorphism expressivity of Local 2-GNNs, while the last one does not.
>
>    In summary, by selecting these five substructures, we validate our theoretical results by demonstrating that the performance of different GNNs in homomorphism counting aligns with their respective homomorphism expressivity.
>
> - **Synthetic dataset for subgraph counting.** We adopt a similar setup to [1], with a primary focus on cycle counting, as explored in [1, 2, 3]. Specifically, we evaluate the performance of various GNN models in counting 3-, 4-, 5-, and 6-cycles. The empirical results align with our theoretical findings, demonstrating that Spectral Invariant GNNs can accurately count these cycles.
>
> [1] Zhang et al. Beyond weisfeiler-lehman: A quantitative framework for GNN expressiveness. ICLR 2024.
>
> [2] Frasca et al. Understanding and extending subgraph gnns by rethinking their symmetries. NeurIPS 2022.
>
> [3] Huang et al. Boosting the cycle counting power of graph neural networks with I$^2$-GNNs. ICLR 2023.
>
> [4] Lim et al. Sign and basis invariant networks for spectral graph representation learning. ICLR 2023.

---

> > ### Comment · Reviewer_B3EZ · 2024-11-25
> >
> > The authors addressed my (few) concerns in an exhaustive manner, therefore I recommend this paper for acceptance at ICLR 2025.

---

### Official Review · Reviewer_i3cu · 2024-11-01

**Soundness:** 3
**Presentation:** 3
**Contribution:** 3
**Rating:** 6
**Confidence:** 1

**Summary:**

This paper is a theoretical work, which studies the homorphism expressivity of spectral invariant GNNs, and proves the fundamental conjecture that 'spectral invariant GNNs converge within constant iterations'. Specifically, the authors show that the homomorphism expressivity of spectral invariant GNNs can be characterized by the set of parallel trees, which bridges between homomorphism and spectral invariants. The authors proves the strict order of the expressiveness of MPNN, spectral GIN, subgraph GNN, and Local 2-GNN with both theories and experiments.

**Strengths:**

1. The authors develop new theoretical tools, e.g. the parallel tree decomposition (Def. A.3) and pebble game for spectral IGN (Def. 3.18), to identify the quantifiable expressiveness of spectral IGNs.
2. The authors conduct empirical experiments in validating the expressiveness inequality / order of mainstream GNN structures (Table 1 and Section 3.2).

**Weaknesses:**

The weakness listed below does not compromise the contribution of this paper.

Although the authors have characterized the homomorphism expressivity of spectral IGNs, there still exists a gap between such expressiveness and the approximation error and generation error of practical spectral IGNs (or other GNN architectures) in machine learning practice (e.g. graph / node / edge prediction). 'Whether we can predict or bound the training / generalization error based on the proven homorphsim expressivity' seems to be a challenging open question (which is out of the scope of this paper). From my perspective, this requires to consider the homomorphism complexity of the data distribution, while this paper only focus on the expressiveness of the hypothesis set / GNN backbone.

**Questions:**

I wonder whether the proposed homomorphism expressiveness can help discriminating between different GNN models (w.r.t both model weights and architectures), so that one can design learning algorithms to improve the training and architecture design of GNN. Or whether this expressiveness can guide GNN designing, e.g. finding GNN paradigm that 'enjoys a larger set of  homomorphism expressiveness'.

May the authors add some illustrations on how can the proposed quantifiable expressiveness benefit GNN learning and design? This can help audiences that do not have a learning theory / GNN expressiveness background to better feel the impact of the series of this work.

---

> ### Author Response · Authors · 2024-11-24
> **Response to Reviewer i3cu**
>
> We thank reviewer i3cu for the positive feedback and for raising these very interesting questions. Here is our response to your comments and questions.
>
> **Regarding generalization.** Thanks for the good question. Generalization is undoubtedly an important aspect of graph learning, though it is not the primary focus of this paper. Nonetheless, our established concepts and tools, in particular, the homomorphism expressivity and tree decomposition, has a profound relationship with generalization. Specifically, for any substructure $F$ in the homomorphism expressivity set, there exists a **fixed** GNN architecture (i.e., model depth and hidden dimension), such that the model can count $F$ for **all** input graphs. In other words, our results essentially reflect the *generalization* ability of GNNs models in homomorphism counting tasks. This is in stark contrast to prior expressivity measures such as the Weisfeiler Lehman test, for which the separation power does not uniformly hold given a fixed model, and the model size must **grow** with input graph size.
>
> Although we did not quantitatively establish a generalization bound, based on the above argument, we believe a generalization bound could be derived using standard analysis (e.g., margin theory or Radamacher complexity). We believe this is a very interesting research question and are happy to leave it as future work.
>
> **Can homomorphism expressiveness help discriminate between different GNN models?** Our work offers deep insights into the expressive power of a variery of practical GNN models, ranging from the classic ChebNet [1] to recently proposed BasisNet [2] and SPE [3], allowing for a dedicated comparison of different models. For example, out results readily implies that ChebNet is strictly less powerful than BasisNet, while BasisNet is strictly less powerful than SPE. Finally, all of these three GNN models are bounded by spectral invatiant GNNs studied in this paper.
>
> Moreover, our results offer insights for the power of these practical GNN architectures in **real-world** tasks. For instance, it is known that the ability to detect and count benzene rings is crucial for the performance on molecular property prediction tasks [7], which is related to the 6-cycle counting power. We can easily prove that SPE can count benzene rings, but the proof does not hold for weaker architectures like BasisNet. This offer clear theoretical evidence to the empirical observation in [2] that SPE performs significantly better than BasisNet on molecular property prediction tasks.
>
> **Practical impact and guidance for GNN design.** Our framework can provide guidance for the design of new GNN models in a more effective and efficient way. Here we provide an illustrative example. It is known that classic message-passing GNNs intrinsically lose expressive power, and most recently proposed GNNs adds preprocessed graph structural information into the GNN models to enhance the expressiveness, such as spectral information, substructure count, distance, etc. However, naively adding all types of structural information will make the resulting GNN too complicated and inefficient. How to select and combine different features in a way that removes redundancy, preserves architectural simplicity, while maintaining a high expressive power is still an active research direction. In this paper, we establish deep connections between three types of graph structural features including graph spectra, homomorphism, and subgraph counting, offering a clear guidance on how to select and combine these features. For example:
>
> - If one adds raw graph spectra (i.e. eigenvalues), there is no need to add the triangle/4-cycle counting as input features, as graph spectra itself can express 4-cycles.
> - If one further adds projection information, 5-cycle counting and 6-cycle counting will also become redundant.
> - However, 4-clique is always essential even if all spectral information is added to GNN design.
>
> We hope our response can address the reviewer's concerns. We are happy to go into details regarding any of them, and we look forward to your reply.
>
> [1] Defferrard et al. Convolutional Neural Networks on Graphs with Fast Localized Spectral Filtering. NeurIPS 2016.
>
> [2] Lim et al. Sign and basis invariant networks for spectral graph representation learning. ICLR 2023.
>
> [3] Huang et al. On the stability of expressive positional encodings for graph neural networks. ICLR 2024.

---

> > ### Comment · Reviewer_i3cu · 2024-11-25
> >
> > Dear authors,
> >
> > Thanks for your insightful response.
> >
> > Regarding the our discussions on "*Can homomorphism expressiveness help discriminate between different GNN models*", while I understood that the framework enables us to rank the expressiveness of different GNN architectures explicitly and elegantly, I was initially pondering "*whether homomorphism expressiveness help discriminating between two GNN models with the same architecture but different parameter weights*". I was thinking about whether this tool can provide us with some computable regularization towards better generalization or approximation performance.
> >
> > I would like to clarify that this is not a further concern, as this is out of the scope of this work. I support this paper towards acceptance as long as no substantial issue is found by other reviewers with expertise.
> >
> > However, since I lack expertise in learning theory, we would like to keep my score as (rating = 6, conf = 1). I keep my confidence score as 1 to avoid affecting the final assessment of this paper.

---

### Official Review · Reviewer_58aL · 2024-11-01

**Soundness:** 3
**Presentation:** 3
**Contribution:** 2
**Rating:** 8
**Confidence:** 4

**Summary:**

This paper explores the expressive power of spectral invariant GNNs through the concept of homomorphism expressivity. The authors provide a framework for characterizing the homomorphism expressivity of spectral invariant GNNs and prove that such GNNs can accurately count homomorphisms for parallel trees. They establish a hierarchy comparing the expressive power of spectral invariant GNNs with other GNN architectures and demonstrate that, while these GNNs have greater expressivity than 1-WL but are bounded by 2-FWL. The paper also extends these results to understand the subgraph counting capabilities of spectral invariant GNNs.

**Strengths:**

The paper provides a rigorous theoretical framework for understanding the expressive power of spectral invariant GNNs, using the novel concept of homomorphism expressivity. By introducing the concept of parallel trees, authors show how spectral invariant GNNs can count homomorphisms for such structures. The authors position spectral invariant GNNs between 1-WL and 2-FWL in terms of expressive power.

**Weaknesses:**

* The sec 2.1, the definition of spectral invariant GNN uses a not so standard eigen-decomposition involves projection matrices. The projection matries need to be idempotent which is an extra constraint. I would expect a more detailed description of this somewhere in the paper.
* As the spectral invariant GNNs is defined differently to existing spectral GNNs such as ChebNet, GCN, etc. The connection to these classical spectral GNNs are not discussed in the paper. If they are not related, the contribution of this paper could be further limited as the results are only applicable to this GNN which can be expensive to run in practice (as it requires full eigendecomposition).
* Dicussion on related work is limited and scattered across multiple places, making it harder to understand the position of this paper in literature.

**Questions:**

* How does the eigen-decomposition described in sec 2.1 work? What are the methods to compute it? Complexity and is it guranteed to find these projection matrices?
* How is spectral invariant GNN relate to other spectral GNNs? Does the results in this paper apply to these GNNs?
(happy to raise score if these concerns are addressed)

---

> ### Author Response · Authors · 2024-11-24
> **Response to Reviewer 58aL (Part 1)**
>
> We thank reviewer 58aL for the careful reading, insightful questions, and positive feedback. Below, we would like give detailed responses to each of the reviewer's concerns and questions.
>
> **Regarding eigen-decomposition.** Thanks for the careful reading and for raising this detailed comment! We would like to clarify that our definition of eigen-decomposition is actually standard. It is well-defined, always exists, and is unique. From linear algebra, we know for any symmetric matrix $\mathbf{A}$, it can be diagonalized as $\mathbf A=\mathbf Q\mathbf \Sigma \mathbf Q^\top$, where $\mathbf \Sigma$ is a diagonal matrix and $\mathbf Q$ is an orthogonal matrix satisfying $\mathbf Q \mathbf Q^\top=\mathbf Q^\top \mathbf Q=\mathbf I$. Denote $\mathbf Q=[\mathbf q_1,\cdots,\mathbf q_n]$ and $\mathbf \Sigma=\operatorname{diag}(\sigma_1,\cdots,\sigma_n)$. Then $\mathbf A=\sum_{i\in[n]}\sigma_i \mathbf q_i \mathbf q_i^\top$, where $\mathbf q_1,\cdots,\mathbf q_n$ satisfies $\sum_i\mathbf q_i\mathbf q_i^\top=\mathbf I$, $\\\|\mathbf q_i\\\|=1$, and $\mathbf q_i^\top \mathbf q_j=0$ for $i\neq j$.
>
> Let $m$ be the number of different eigenvalues ($m \le n$), and denote $\lambda_1,\cdots,\lambda_m$ be the set of different eigenvalues $\sigma_1,\cdots,\sigma_n$ after deduplication. Define $\mathbf P_i=\sum_{j:\sigma_j=\lambda_i}\mathbf q_j\mathbf q_j^\top$. It follows that
>
> - $\sum_{i} \mathbf{P}_i =\sum_j\mathbf q_j\mathbf q_j^\top=\mathbf I$;
> - $\mathbf{P}_i \mathbf{P}_j = \mathbf O$ for $i \neq j$ (since $\mathbf q_k^\top \mathbf q_l=0$ for $k\neq l$);
> - $\mathbf P_i \mathbf P_i =\sum_{j:\sigma_j=\lambda_i}\sum_{k:\sigma_k=\lambda_i} \mathbf q_j\mathbf q_j^\top\mathbf q_k\mathbf q_k^\top=\sum_{j:\sigma_j=\lambda_i}\mathbf q_j\mathbf q_j^\top=\mathbf{P}_i $;
> - $\sum_{i}\lambda_i \mathbf{P}_i=\sum_j\sigma_j \mathbf q_j\mathbf q_j^\top=\mathbf{A}$.
>
> This proves that our definition of $\mathbf P_1,\cdots,\mathbf P_m$ always exists. Note that the third item is the idempotent property. Moreover, the decomposition $\mathbf A=\sum_{i\in[m]}\lambda_i \mathbf P_i$ is unique (unlike the decomposition $\mathbf A=\mathbf Q\mathbf \Sigma \mathbf Q^\top$), which is a classic result in linear algebra. We apologize for the lack of explanation in the initial version of our paper. Thank you for pointing out this valuable suggestion; we have added further clarification in the revised version.
>
> **Regarding computation method of eigen-decomposition.** Based on the previous formulas, to compute eigen-decomposition, it suffices to compute all eigenvalues and eigenvectors of $\mathbf A$, namely, $\sigma_i$ and $\mathbf q_i$ for $i\in[n]$. This is a standard operation supported in modern scientific libraries, such as numpy or pytorch. The worst-case computational complexity is $O(n^3)$, but there are many efficient (randomized/approximation) algorithms to reduce the complexity. Note that the preprocessing step can only be performed once and is negligible compared with training a neural network. Also note that the complexity is smaller than preprocessing some common graph structural information like the number of 4-cliques.

---

> ### Author Response · Authors · 2024-11-24
> **Response to Reviewer 58aL (Part 2)**
>
> **Related to other spectral GNNs.** Thank you for raising this great question. Our results can indeed be applied to analyze other spectral GNNs, such as ChebNet and recently proposed architectures like BasisNet [1] and SPE [2].
>
> - **Regarding GCN.** Analyzing the homomorphism expressivity of GCNs is straightforward as it has been proved to be bounded by 1-WL. Therefore, their homomorphism expressiveness is at most the set of all forests.
>
> - **Regarding ChebNet.** Our results can offer insights into the homomorphism expressivity of ChebNet. It is known that ChebNet is not more powerful than an MPNN in distinguishing non-isomorphic graphs if the input graphs have the same maximal eigenvalue [3]. Based on this observation, the expressive power of ChebNet is bounded by 1-WL plus the set of eigenvalues, for which the homomorphism expressivity has been given in our paper (Theorem 3.5). So we can conclude that the homomorphism expressiveness of ChebNet is at most the set of all forests augmented by the set of all cycles. As a corollay, it is strictly less powerful than spectral invariant GNNs, since the set of all forests augmented by the set of all cycles is still a proper subset of the set of parallel trees.
>
> - **Recent architectures.** In addition to GCN and ChebNet, recent spectral GNNs, such as BasisNet and SPE, also possess expressive power no greater than that of spectral invariant GNNs. Hence, our results are applicable to these networks as well, and all negative results (expressivity limitations) holds for these architectures. Moreover, the expressive power of SPE has been proved to be equivalent to that of spectral invariant GNNs. Therefore, our positive results can be applied to SPE as well. For example, as indicated in Corollary 3.14, spectral invariant GNNs can count cycles and paths with up to 7 vertices, implying that SPE can also count paths and cycles with up to 7 vertices.
>
> **Regarding related works.** Thank you for pointing out this weakness. To further improve clarity,  in the revised version we have added a dedicated section in the Appendix to discuss related work, which we believe will help readers better understand the positioning of our paper within the GNN literature.
>
> We hope our response can address the reviewer's concerns. We are happy to go into details regarding any of them, and we look forward to your reply.
>
> [1] Lim et al. Sign and basis invariant networks for spectral graph representation learning. ICLR 2023.
>
> [2] Huang et al. On the stability of expressive positional encodings for graph neural networks. ICLR 2024.
>
> [3] Geerts & Reutter. Expressiveness and approximation properties of graph neural networks. ICLR 2022.

---

> > ### Comment · Reviewer_58aL · 2024-11-29
> >
> > Thank you for the reply. My concerns has been address, I have raised the score.

---

### Official Review · Reviewer_9NdD · 2024-11-04

**Soundness:** 2
**Presentation:** 3
**Contribution:** 2
**Rating:** 6
**Confidence:** 4

**Summary:**

The paper investigates the expressive power of spectral invariants in GNNs by employing a homomorphism-based approach. The authors theoretically demonstrate that spectral invariant GNNs can homomorphism-count certain tree-like structures, thus providing a quantitative hierarchy of expressiveness across different GNN architectures. This hierarchy assesses the influence of spectral invariants on the capability of counting substructures, such as small cycles. Empirical results test how homomorphism expressivity is related with performance in downstream tasks.

**Strengths:**

The paper is well-written and organized. The key concepts, such as spectral invariants, homomorphism expressivity, and the hierarchy of expressiveness, are presented in a manner that facilitates readability.

This analysis introduces a novel perspective, providing an alternative theoretical framework for examining the expressive power and function approximation capabilities of GNNs.

**Weaknesses:**

I am concerned about the significance of the paper's contributions. The work presents a limited set of novel results and falls short in advancing our understanding of the expressive power of message-passing and spectral invariant GNNs. While the paper is well-written and accessible, it does not substantially enhance our knowledge of GNN architecture or provide actionable insights for designing more efficient models.

Some important related work is missing:

Kanatsoulis, C. and Ribeiro, A., Counting Graph Substructures with Graph Neural Networks. In The Twelfth International Conference on Learning Representations.

Black, M., Wan, Z., Mishne, G., Nayyeri, A. and Wang, Y., Comparing Graph Transformers via Positional Encodings. In Forty-first International Conference on Machine Learning.

How does the submitted paper compare to these previous works?

**Questions:**

Please see weaknesses

---

> ### Author Response · Authors · 2024-11-24
> **Response to Reviewer 9NdD (Part 1)**
>
> We thank reviewer 9NdD for the valuable comments, particularly for highlighting two highly relevant works. Below, we would like to address each of the reviewer's concerns and questions.
>
> **Regarding the significance of our results.**
> We would like to clarify that our work provides a unified framework for various previously proposed spectral GNN architectures. This  not only leads to a deep understanding of the expressive power of many architectures in a principled way, but also provides practical guidelines for designing efficient models that encode spectral and substructure information. We elaborate it in the following aspects:
>
> - **Insights on various prior GNN models.** This paper significantly enhances our understanding of a wide range of practical spectral GNN architectures, from the classic ChebNet [5] to the more recent BasisNet [1] and SPE [2]. Specifically, all these architectures are bounded in expressive power by the Spectral Invariant GNN. Consequently, the implications of our work can be directly applied to these models, revealing their *practical limitations*. For example, our results can be used to demonstrate that ChebNet, BasisNet, and SPE cannot count substructures like 4-cliques. Furthermore, since the expressive power of SPE is equivalent to that of the Spectral Invariant GNN [6], all positive results in this paper are also applicable to SPE. For instance, we prove that the Spectral Invariant GNN can count 6-cycles, a result that extends to SPE.
>
> - **Relevance to real applications.** Our findings provide theoretical insights into the capabilities of these practical GNN architectures in **real-world tasks**. For example, detecting and counting benzene rings is crucial for molecular property prediction tasks [7], which relates to the ability to count 6-cycles. Our theoretical results can be used to show that SPE can detect and count benzene rings, while it may not hold for weaker architectures like BasisNet. This offers clear evidence for the empirical observation in [2] that SPE significantly outperforms BasisNet in molecular property prediction tasks.
>
> - **Guidance for new GNN design.** Our framework can offer guidance for designing new GNN models in a more effective and efficient manner. We give the following concrete example. It is known that classic message-passing GNNs inherently lose expressive power, and many recent GNNs attempt to address this by incorporating preprocessed graph structural information, such as graph spectra, substructure counts, or distance metrics. However, simply adding all types of structural information will make the resulting GNN overly complex and inefficient. Selecting and combining features in a way that removes redundancy, preserves simplicity, and maintains high expressiveness remains an active research area.
>
>    In this paper, we establish deep connections among three types of graph structural features: graph spectra, homomorphism, and subgraph counting, offering clear guidance for feature selection and combination. For example:
>
>    - If raw graph spectra (e.g., eigenvalues) are added into GNN design, triangle and 4-cycle counts are unnecessary as spectra alone can express 4-cycles.
>    - Adding projection information further renders 5-cycle and 6-cycle counts redundant.
>    - However, 4-clique counts remain essential, even with all spectral information included in the GNN design.

---

> ### Author Response · Authors · 2024-11-24
> **Response to Reviewer 9NdD (Part 2)**
>
> **Discussions on related work.** Thank you for pointing out these valuable related works! In the revised version of the paper, we have cited these references and provided a detailed discussion in the related work section. Below, we summarize our key points:
>
> - **Discussions on [4].** [4] introduced several new WL algorithms based on absolute and relative positional encodings (PE). The authors further established a bunch of equivalence relationships among these algorithms. Notably, we observe a strong connection between the proposed "stack of power of matrices" PE and Spectral Invariant GNNs.
>     - We can prove that the proposed $(I,L,\cdots,L^{2n-1})$-WL (see Theorem 4.6 in [4]) is **as expressive as** spectral invariant GNNs with matrix $L$, and similarly, $(I,A,\cdots,A^{2n-1})$-WL is **as expressive as** spectral invariant GNNs with the ordinary adjacency matrix. Therefore, all results in our paper can be used to understand the power of these WL variants. For example, we can readily show that $(I,A,\cdots,A^{2n-1})$-WL is bounded by Local 2-WL in expressive power, which is further strictly bounded by 2-FWL.
>
>     - Since [6] has shown that the expressive power of RD-WL is bounded by Spectral Invariant GNNs, it follows that the proposed $L^\dagger$-WL (see Theorem 4.6 in [4]) is also bounded in expressive power by Spectral Invariant GNNs. This conclusion reproduces their key result (Theorem 4.4 in [4]).
>
>    Besides the similarities, we would like to highlight that our paper goes beyond **stable** WL algorithms and further studies how model depth plays a role in the expressive power. This is highly challenging and we regard it as a major contribution of our work.
>
> - **Discussion on [3].** [3] studies subgraph counting power for a novel GNN framework, where classic message-passing GNNs are enhanced with random node features, and the GNN output is computed by taking the expectation over the introduced randomness. The paper demonstrates that such GNNs can learn to count various substructures, including cycles and cliques. These findings share similarities with our work, as both studies characterize the cycle-counting power of certain GNN models. Notably, the GNN framework proposed in [3] can count more complex substructures, such as 4-cliques and 8-cycles, which exceed the expressive power of 2-FWL.
>
>    However, unlike Spectral Invariant GNNs, the GNNs in [3] cannot compute outputs deterministically due to the reliance on randomness. As a result, accurately computing outputs requires sampling a large number of input variables, which poses additional computational challenges.
>
> We hope our response can address the reviewer's concerns. We are happy to go into details regarding any of them, and we look forward to your reply.
>
> [1] Lim et al. Sign and basis invariant networks for spectral graph representation learning. ICLR 2023.
>
> [2] Huang et al. On the stability of expressive positional encodings for graph neural networks. ICLR 2024.
>
> [3] Kanatsoulis and Ribeiro. Counting Graph Substructures with Graph Neural Networks. ICLR 2024.
>
> [4] Black et al. Comparing Graph Transformers via Positional Encodings. ICML 2023.
>
> [5] Defferrard et al. Convolutional Neural Networks on Graphs with Fast Localized Spectral Filtering. NeurIPS 2016.
>
> [6] Zhang et al. On the expressive power of spectral invariant graph neural networks. ICML 2024.
>
> [7] Huang et al. Boosting the Cycle Counting Power of Graph Neural Networks with I$^2$-GNNs. ICLR 2023.

---

> > ### Comment · Reviewer_9NdD · 2024-11-26
> >
> > I thank the authors for their detailed response. I have increased my score and now recommend the paper for acceptance.

---

### Official Review · Reviewer_MN8L · 2024-11-04

**Soundness:** 4
**Presentation:** 3
**Contribution:** 4
**Rating:** 8
**Confidence:** 4

**Summary:**

The paper investigates the expressive power of spectral-invariant GNNs through the homomorphism expressivity perspective. The authors prove the followings: (1) projection-based spectral invariants and corresponding GNNs are strictly bounded by 2-FWL, and establish a quantitative hierarchy among spectral invariants and a family of WL-test variants; (2) increasing numbers of iterations can improve expressive power with an upper-bound; (3) substructure counting capabilities of certain spectral invariant GNNs. Experiments on synthetic and real-world datasets are conducted.

**Strengths:**

The paper is well-organized. The theoretical contributions are significant, addressing several remaining open questions including existence of homomorphism expressivity of certain GNN architectures. The paper quantitatively characterizes the expressivity of spectral invariant GNNs through parallel tree, establishes connections with k-FWL hierarchy, proves counting abilities of several substructures, and extends to higher-order spectral invariant GNNs. I check with proofs and they are solid.

**Weaknesses:**

While the paper is theoretically grounded, the experimental part could be strengthened. (1) The expressivity of spectral-invariant GNNs can be further validated on classical benchmarks of GNN expressivity, e.g., EXP, CSL, SR25, and BREC [1]. (2) The study on real-world datasets is insufficient. In many circumstances, expressive power does not necessarily determine the empirical performance on real-world datasets, so the authors should carry more extensive experiments on more benchmarks with careful tuning.

[1] Wang, Y., & Zhang, M. (2023). Towards better evaluation of gnn expressiveness with brec dataset. arXiv preprint arXiv:2304.07702.

**Questions:**

N/A

---

> ### Author Response · Authors · 2024-11-24
> **Response to Reviewer MN8L**
>
> We thank reviewer MN8L for the valuable suggestions and positive feedback on our work. Following your advice, we conducted additional experiments on the recent benchmarks of GNN expressivity, specifically using the BREC dataset. Here, we reimplemented MPNN, vanilla Subgraph GNN, Local 2-GNN, and spectral invariant GNN in a unified codebase. To ensure a fair comparison, the model depths, architectural components (e.g., activation functions, aggregation types, pooling functions), and the number of parameters are chosen to be the same across all models. We adopted the configuration provided in [1], training all models for 20 epochs and evaluating them using RPC. The results are presented in the table below.
>
> | Model | Basic | Regular | Extension | CFI | Total |
> |--|--|--|--|--|--|
> | MPNN | 0 (0%) | 0 (0%)  | 0 (0%) | 0 (0%) | 0 (0%) |
> | Subgraph GNN | 60 (100%) | 50 (35.7%) | 98 (90%) | 11 (11%) | 219 (54.8%) |
> | Local 2-GNN | 60 (100%) | 50 (35.7%) | 100 (100%) | 12 (12%) | 222 (55%) |
> | Spectral invariant GNN | 60 (100%) | 50 (35.7%) |100 (100%) | 9 (9%) | 219 (54.8%) |
>
> In the table above, each number represents the number of distinguishable graph pairs for each of the four graph classes (Basic, Regular, Extension, and CFI), along with their corresponding percentages. The final column summarizes the overall performance of different models. One can see that spectral invariant GNNs has a similar performance to that of Subgraph GNN and Local 2-GNN. Such observations actually highlight the challenges of our theoretical results, as revealing the expressivity gaps between spectral invariant GNNs and Subgraph/Local GNNs requires a much more fine-grained analysis, which may not be easily achieved using toy example graphs.
>
> Beside the BREC dataset, we also tried additional experiments on the Alchemy dataset. However, due to the tight schedule, the results have not yet been finalized. We will include these results in the final version of the paper.
>
> [1] Wang \& Zhang. Towards better evaluation of GNN expressiveness with BREC dataset. arXiv preprint arXiv:2304.07702.
>
> [2] Zhang et al. Beyond Weisfeiler-Lehman: A quantitative framework for GNN expressiveness. ICLR 2024.

---

> > ### Comment · Reviewer_MN8L · 2024-11-25
> >
> > I thank the authors for their response. I will keep my positive rating.

---

### Author Response · Authors · 2024-11-24
**General Response on Paper Updates**

We sincerely thank all the reviewers and the area chair for their great efforts in reviewing our paper. We have followed the reviewers' suggestions and updated our paper accordingly. We highlight our major updates as follows:
- We added a comprehensive discussion of related work in Appendix A, as suggested by Reviewer 58aL.
- We also add discussions to several recent works as pointed out by Reviewer 9NdD.
- We slightly modify the eigen-decomposition part in Section 2 to make it more readable (as suggested by Reviewer 58aL).

---

### Meta-Review · Area_Chair_fhmN · 2024-12-11

**Metareview:**

This paper, on the expressive power of spectral invariants in GNNs via homomorphism counts, received all-around positive reviews.
For example, some reviewers said that the analysis introduces a novel perspective, providing an alternative theoretical framework for examining the expressive power and function approximation capabilities of GNNs. Other reviewers say that the theoretical contributions are significant, and that the paper addresses several open questions.
The recommendation is to accept (spotlight).

**Additional Comments On Reviewer Discussion:**

Some reviewers raised concerns about the literature review. They increased their scores after the discussion period.

---

### Decision · Program_Chairs · 2025-01-22

Accept (Oral)